# BCC-CSM2-HR: A High-Resolution Version of the Beijing Climate Center Climate System Model

Tongwen Wu[1*], Rucong Yu[1], Yixiong Lu[1], Weihua Jie[1], Yongjie Fang[1], Jie Zhang[1],

Li Zhang[1], Xiaoge Xin[1], Laurent Li[1,2], Zaizhi Wang[1], Yiming Liu[1], Fang Zhang[1],

Fanghua Wu[1], Min Chu[1], Jianglong Li[1], Weiping Li[1], Yanwu Zhang[1],

Xueli Shi[1], Wenyan Zhou[1], Junchen Yao[1], Xiangwen Liu[1], He Zhao[1], Jinghui Yan[1],

Min Wei[3], Wei Xue[4], Anning Huang[5], Yaocun Zhang[5], Yu Zhang[6], Qi Shu[7], Aixue Hu[8]

1.*Beijing Climate Center, China Meteorological Administration, Beijing, China*

2.*Laboratoire de Mééorologie Dynamique, IPSL, CNRS, Sorbonne Université, Ecole Normale Supérieure, Ecole Polytechnique, Paris 75005, France*

3.*National Meteorological Information Center, China Meteorological Administration, Beijing 100081, China.*

4.*Tsinghua University, Beijing 100084, China*

5.*Nanjing University, Nanjing 210023, China*

6.*Chengdu University of Information Technology, Chengdu 610225, China*

7.*The First Institute of Oceanography of the Ministry of Natural Resources, Qingdao 266061, China*

8.*National Center for Atmospheric Research, PO Box 3000, Boulder, Colorado 80307-3000, USA*

*Corresponding author: Tongwen Wu (twwu@cma.gov.cn)

First version on August 23, 2020

Revised on March 27, 2021

**Abstract**

BCC-CSM2-HR is a high-resolution version of the Beijing Climate Center (BCC) Climate System Model (T266 in the atmosphere and 1/4 ˚lat.×1/4 ˚lon. in the ocean). Its development is on the basis of the medium-resolution version BCC-CSM2-MR (T106 in the atmosphere and 1 ˚lat.×1 ˚lon. in the ocean) which is the baseline for BCC participation in the Coupled Model Intercomparison Project Phase 6 (CMIP6). This study documents the high-resolution model, highlights major improvements in the representation of atmospheric dynamical core and physical processes. BCC-CSM2-HR is evaluated for historical climate simulations from 1950 to 2014, performed under CMIP6-prescribed historical forcing, in comparison with its previous medium-resolution version BCC-CSM2-MR. Observed global warming trend of surface air temperature from 1950 to 2014 are well captured by both BCC-CSM2-MR and BCC-CSM2-HR. Present-day basic atmospheric mean states during the period from 1995 to 2014 are then evaluated at global scale, followed by an assessment on climate variabilities in the tropics including the tropical cyclones (TCs), the El Niño–Southern Oscillation (ENSO), the Madden-Julian Oscillation (MJO), and the quasi-biennial oscillation (QBO) in the stratosphere. It is shown that BCC-CSM2-HR keeps well the global energy balance and can realistically reproduce main patterns of atmosphere temperature and wind, precipitation, land surface air temperature and sea surface temperature (SST). It also improves the spatial patterns of sea ice and associated seasonal variations in both hemispheres. The bias of double intertropical convergence zone (ITCZ), obvious in BCC-CSM2-MR, almost disappears in BCC-CSM2-HR. TC activity in the tropics is increased with resolution enhanced. The cycle of ENSO, the eastward propagative feature and convection intensity of MJO, the downward propagation of QBO in BCC-CSM2-HR are all in a better agreement with observation than their counterparts in BCC-CSM2-MR. Some imperfections are however noted in BCC-CSM2-HR, such as the excessive cloudiness in the eastern basin of the tropical Pacific with cold SST biases and the insufficient number of tropical cyclones in the North Atlantic.

## 1. Introduction.

Accurately modeling climate and weather is a major challenge for the scientific community and needs high spatial resolution. However, many climate models, such as those involved in the Coupled Model Intercomparison Project Phase 5 (CMIP5, Taylor et al., 2012) and the more recent CMIP6 (Eyring et al., 2016), still use a spatial resolution of hundreds of kilometers (Flato et al., 2013). This nominal resolution is suitable for global-scale applications that run simulations for centuries into the future, but fails to capture small-scale phenomena and features that influence local or regional weather and climate events. This resolution is fine enough to simulate mid-latitude weather systems which evolve in thousands of kilometers, but insufficient to describe convective cloud systems that rarely extend beyond a few tens of kilometers. The study of Strachan et al. (2013) showed that while the average tropical cyclone number can be well simulated at a resolution of around 130 km, grids finer than 60 km are needed to properly simulate the inter-annual variability of cyclone counts. Higher horizontal resolutions (e.g. 50 km) can further improve the simulated climatology of tropical cyclones (e.g., Oouchi et al., 2006; Zhao et al., 2009; Murakami et al., 2012; Manganello et al., 2012; Bacmeister et al., 2014; Wehner et al., 2015; Reed et al., 2015; Zarzycki et al., 2016). Growing evidence showed that high-resolution models (50 km or finer in the atmosphere) can reproduce the observed intensity of extreme precipitation (Wehner et al., 2010; Endo et al., 2012; Sakamoto et al., 2012). Some phenomena are sensitive to increasing resolution such as ocean mixing (Small et al., 2015), diurnal cycle of precipitation (Sato et al., 2009; Birch et al., 2014; Vellinga et al., 2016), the QBO (Hertwig et al., 2015), the MJO (Peatman et al., 2015), and monsoons (Sperber et al., 1994; Lal et al., 1997; Martin et al., 1999; Yao et al., 2017; Zhang et al. 2018). Some small-scale processes associated with mid-latitude storms and tropical cyclones, and ocean eddies also feedback on the simulated large-scale circulation, climate variability and extremes (Smith et al., 2000; Masumoto et al., 2004; Mizuta et al., 2006; Shaffrey et al., 2009; Masson et al., 2012; Doi et al., 2012; Rackow et al., 2016). Many studies (e.g. Ohfuchi et al., 2004; Zhao et al., 2009; Walsh et al., 2012; Bell et al., 2013; Strachan et al., 2013; Kinter et al.

2013; Demory et al., 2014; Schiemann et al., 2014; Small et al. 2014; Shaevitz et al., 2014; Hertwig et al., 2015; Murakami et al., 2015; Hertwig et al., 2015; Roberts et al. 2016; Hewitt et al. 2016; Roberts C.D. et al, 2018; Roberts M.J. et al., 2019) show that enhanced horizontal resolution in atmospheric and ocean models has many beneficial impacts on model performance and helps to reduce model systematic biases.

High-resolution climate system modelling becomes a key activity within the climate research community, although increasing model resolution needs considerable computational resources. In 2004, the first high-resolution global climate model produced its first simulations using the Japanese Earth Simulator (Ohfuchi et al., 2004; Masumoto et al., 2004). At present day, performing high-resolution climate simulations with model grid smaller than 50 km in the atmosphere and $0.25°$ in the ocean is still a very costly effort but a growing number of research centers can exercise it (e.g. Shaffrey et al., 2009; Delworth et al., 2012; Mizielinski et al., 2014; Bacmeister et al., 2014; Satoh et al., 2014; Roberts et al., 2018; Zhou et al., 2020). The High Resolution Model Intercomparison Project (HighResMIP, Haarsma et al., 2016) is a CMIP6-endorsed MIP (Model Intercomparison Project), which aimed to investigate the impact of model resolution on climate simulation fidelity and systematic model biases.

As a main climate modelling center in China (Wu et al., 2010, 2013, 2014, 2019, 2020; Xin et al., 2013, 2019; Li et al., 2019; Lu et al., 2020a,b), Beijing Climate Center (BCC), China Meteorological Administration, also put important efforts in developing high-resolution fully-coupled Beijing Climate Center Climate System Model (BCC-CSM-HR) (Yu et al., 2014). The currently released version (BCC-CSM2-HR, Table 1) is one of the three BCC model versions (Wu et al., 2019) involved in CMIP6 to run HighResMIP experiment. It is now in its pre-operational phase to become the next generation Beijing Climate Center Climate Prediction System to produce forecasts at leading times of two weeks to one year. The purpose of this paper is to evaluate its performance by comparing it with the medium resolution version (BCC-CSM2-MR, Wu et al., 2019). In particular, we assess their

performance to simulate large-scale mean climate and some important phenomena
such as the ITCZ, tropical cyclones (TCs), MJO, and QBO which are expected to be
improved with enhanced resolution. A relevant description of BCC-CSM2-HR is
shown in Section 2, and the experiment design is shown in Section 3. Main results of
model performance are presented in Section 4.
**2. Model description at high-resolution configuration**

Due to the diversity of research and operational needs in BCC, a basic rule that

we imposed to ourselves in the development of BCC-CSMs (Wu et al., 2019) is the
construction of a traceable hierarchy of model versions running from a coarse grid
(T42, approximately 280km), to a medium grid (T106, approximately 110×110 km),
and to a fine grid (T266, around 45×45 km). Actually, we fulfilled our target with an
achievement to deliver all of these model versions. All of them are fully-coupled
models with four components □ atmosphere, ocean, land and sea-ice □ interacting
with each other (Wu et al., 2013, 2019, 2020). They are physically coupled through
fluxes of momentum, energy, water at their interfaces. The ocean‐atmosphere
coupling frequency is 30 minutes, which is sufficient to account for the diurnal cycle.
As shown in Table 1, the medium resolution of BCC-CSM2-MR is at T106 for the
atmosphere and has 46 layers with its model lid at 1.459 hPa. The resolution of the
global ocean is of 1 ˚lat.×1 ˚lon. on average, but 1/3 °lat.×1 ˚lon. for the tropical oceans.
BCC-CSM2-MR was described in detail in Wu et al. (2019). The atmosphere
resolution of BCC-CSM2-HR is T266 on the globe and 56 layers with the top layer at
0.156 hPa (Figure 1) and model lid at 0.092 hPa (Table 1). The ocean and sea ice
resolution in BCC-CSM2-HR is 1/4 ˚lat.×1/4 ˚lon. and 40 layers in depth. Compared to
BCC-CSM2-MR, BCC-CSM2-HR is updated for its dynamical core and model
physics in the atmospheric component (Table 1). The ocean and sea ice components
are also updated from Modular Ocean Model version 4 (MOM4) and Sea Ice
Simulator version 4 (SIS4) in BCC-CSM2-MR to their version 5 (MOM5 and SIS5),
respectively. The land component in the two versions of BCC-CSMs is the Beijing
Climate Center Atmosphere-Vegetation Interaction Model version 2 (BCC-AVIM2,
Li et al., 2019).
**2.1 Atmosphere Model**
The atmospheric component of BCC-CSM2-MR is Beijing Climate Center
Atmospheric General Circulation Model version 3 (BCC-AGCM3) at medium
resolution (BCC-AGCM3-MR), with details being described in Wu et al. (2019) and
in a series of relevant publications (Wu et al., 2008, 2010; Wu, 2012; Wu et al., 2013;
Lu et al., 2013; Wu et al., 2019; Lu et al., 2020a; Wu et al., 2020). The dynamical
core in BCC-AGCM3-MR uses the spectral framework as described in Wu et al.
(2008), in which explicit time difference scheme is applied to vorticity equation,
semi-implicit time difference scheme for divergence, temperature, and surface
pressure equations, and semi-Lagrangian tracer transport scheme is used for water
vapor, liquid cloud water and ice cloud water. The main model physics in
BCC-AGCM3-MR was described in Wu et al. (2019), which includes the modified
scheme of deep convection suggested by Wu (2012), a new diagnostic scheme of
cloud amount (Wu et al, 2019), the shallow convection transport scheme of Hack
(1994), the stratiform cloud microphysics following the framework of non-convective
cloud processes in NCAR Community Atmosphere Model version 3 (CAM3, Collins
et al., 2004) but with a different treatment for indirect effects of aerosols affecting
clouds and precipitation, the radiative transfer parameterization that was originally
implemented in CAM3, a modified boundary layer turbulence parameterization based
on the eddy diffusivity approach (Holtslag and Boville, 1993), and a treatments of
gravity waves that are generated by a variety of sources related to orography and
convection (Lu et al., 2020a).
The atmospheric component in BCC-CSM2-HR is the newly-developed version
of BCC-AGCM3 with high-resolution (BCC-AGCM3-HR). Main differences
between BCC-AGCM3-HR and BCC-AGCM3-MR are listed in Table 1 and detailed
in the following sub-sections. Actually, the high-resolution atmospheric component
has incorporated a spatially-varying divergence damping scheme, amelioration of
Wu's deep convective scheme (Wu, 2012), and an integrated consideration for
shallow convection and boundary layer processes.
**a.  Spatially-varying divergence damping**
The performance of a climate model is largely determined by complex motions
at different spatiotemporal scales and interactions between them. Subgrid-scale
motions are generally caused by high-frequency waves, and they can exert impacts on
the computational stability especially for a high-resolution model. Horizontal
divergence damping is often needed to control numerical noise in weather forecast
models and for numerical stability reasons (Dey, 1978; Bates et al., 1993; Whitehead
et al., 2011).
In BCC-AGCM3-HR, a second-order and a fourth-order horizontal Laplacians
($\nabla^2$ and $\nabla^4$) are used to realize the damping operation on the divergence field D:
$$\frac{\partial D}{\partial t} = \cdots + k_2 \nabla^2 D, \tag{1}$$

and
$$\frac{\partial D}{\partial t} = \cdots - k_4 \nabla^4 D, \tag{2}$$

where $k_2$ and $k_4$ express the damping coefficients for the second-order and
fourth-order dissipation operators, respectively. They are generally set as constant
parameters. The second-order damping is used for the top three layers and the
fourth-order damping for other layers.
Whitehead et al. (2011) proposed a horizontal divergence damping scheme that
works on a latitude–longitude grid by using a linear von Neumann analysis. Here, we
extended their idea to the spectral dynamical core in our high-resolution model
BCC-AGCM3-HR, and we use a second-order horizontal damping operator with
spatially-varying damping coefficient. In order to express the grid spacing
dependence of the dissipation, an additional term is introduced in Eqs. (1) and (2) as:
$$\frac{\partial D}{\partial t} = \cdots + k_2 \nabla^2 D + k_v \nabla^2 D \tag{3}$$

and
$$\frac{\partial D}{\partial t} = \cdots - k_4 \nabla^4 D + k_v \nabla^2 D. \tag{4}$$

where
$$\qquad k_v = C_s \frac{[A_E \Delta\emptyset] \cdot [A_E \Delta\lambda]}{\Delta t} \qquad\qquad (5)$$
$k_v$ is dependent on the time-step $\Delta t$ and grid spacing. $A_E$ in Eq. (5) is the radius of
the earth. $\Delta\emptyset$ and $\Delta\lambda$ stand for the latitudinal and longitudinal mesh sizes,
respectively. The parameter $C_s$ is designed to depend on vertical position as,
$$\qquad C_s = C_{s0} \max \langle 1, 8\left\{1 + \tanh\left[\ln(\frac{p_{\text{top}}}{p_k})\right]\right\} \rangle, \qquad\qquad (6)$$
where $C_{s0}$ is a constant and related to model resolution, $p_{\text{top}}$ and $p_k$ are the
pressures at the top and the kth layers of the model, respectively. The expression (6)
provides a rather flat vertical profile until the final two to three model levels, where
the damping coefficient is increased rapidly by up to a factor of 8 (Whitehead et al.,
2011). This dependence introduces a diffusive sponge layer near the model top to
absorb rather than reflect outgoing gravity waves (Whitehead et al., 2011). The
expression (5) implies the damping coefficient increase with latitude for
BCC-AGCMs spectral grid. This spatially-varying damping scheme can improve the
atmospheric temperature simulation in the stratosphere, especially at polar areas of
both hemispheres, which is possibly due to the more efficient damping of small-scale
meridional waves as Whitehead et al. (2011) pointed out.
**b. Deep convection**
In BCC-AGCM3-MR, as well as in BCC-AGCM3-HR, a modified scheme of the
deep cumulus convection developed by Wu (2012) is used (Wu et al., 2019). It is
characterized by the following points:
(1) Deep convection is initiated at the level of maximum moist static energy
above the boundary layer, and convection is triggered only when the boundary layer is
unstable or there exists updraft velocity in the environment at the lifting level of
convective cloud, and simultaneously there is positive convective available potential
energy (CAPE).
(2) A bulk cloud model is used to calculate the convective updraft with
consideration of budgets for mass, dry static energy, moisture, cloud liquid water, and
momentum, and the entrainment/detrainment amount for the updraft cloud parcel is
determined according to the increase/decrease of updraft parcel mass with altitude.
(3) The convective downdraft is assumed to be saturated and originated from the
level of minimum environmental saturated equivalent potential temperature within the
updraft cloud.
(4) The closure scheme determines the mass flux at the base of convective cloud,
and depends on the decrease/increase of CAPE resulting from large-scale processes.
Along with increasing resolution in BCC-AGCM3-HR, the detrained cloud water
can be transported to its adjacent grid boxes, which is accomplished in the dynamical
core. Part of the horizontally-transported cloud water is permitted to be transferred
downward to lower troposphere and the amount of downward transferred water vapor
is determined by the horizontally-transported convective cloud water increment with
time. These modifications of the deep convection scheme only in BCC-CSM2-HR are
found in favor of improving the simulation of eastward propagation of MJO in the
tropics, and their details will be presented in another paper.
**c.  Boundary layer turbulence**
BCC-AGCM3-HR employs the University of Washington Moist Turbulence
(UWMT) scheme as proposed in Bretherton and Park (2009) to replace the dry
turbulence scheme of Holtslag and Boville (1993). The latter was used in
BCC-AGCM3-MR. In UWMT, the first-order K diffusion is used to represent all
turbulences, by which the turbulent fluxes of a variable $\chi$ are written as

$$\overline{w'\chi'} = -K_\chi \frac{\partial \chi}{\partial z}$$

(6)

The eddy diffusivity, $K_\chi$, is calculated based on the turbulent kinetic energy $e$ and
proportional to the stability-corrected length scale $L$, given by

$$K_\chi = L\sqrt{e}.$$

(7)

In the case of an entrainment layer at the top of convective boundary layers (BLs), the
diffusivity is parameterized with

$$K_\chi = w_e \Delta z_e,$$

(8)

where $\Delta z_e$ is the thickness of the entrainment layer, and $w_e$ is the entrainment rate
which uses the expression in Nicholls and Turton (1986):

$$w_e = A \frac{w_*^3}{\left( g \Delta^E s_{vl} / s_{vl} \right) \left( z_t - z_b \right)}$$

. (9)

Here, $w_*$ is the convective velocity, $z_t$ and $z_b$ are the top and bottom heights of
the entrainment layer, $\Delta^E$ denotes a jump across the entrainment layer, and $s_{vl}$ is
the liquid virtual static energy. *A* is a nondimensional entrainment efficiency, which is
affected by evaporative cooling of mixtures of cloud-top and above-inversion air.
Compared to dry convective BLs over land which is mainly forced by the surface
heating, the structure of marine stratocumulus-topped BLs depends strongly on
dominant turbulence generating mechanism resulting from both evaporative and
radiative cooling at cloud top. The UWMT scheme aims to provide a more physical
and realistic treatment of marine stratocumulus-topped BLs and it has been
demonstrated that the observed patterns of low-cloud amount with maxima in the
subtropical stratocumulus decks can be well reproduced by UWMT in the Community
Atmosphere Model (Park and Bretherton, 2009). The implementation of the UWMT
scheme in BCC-AGCM3-HR is aimed to improve the simulation of the low-level
clouds over subtropical eastern oceans and these improvements are found critical to
reduce the double-ITCZ bias of precipitation (Lu et al., 2020b).
d. **Shallow convection**
BCC-AGCM3-HR basically inherits the shallow convection parameterization
used in BCC-AGCM3-MR, which is a stability-dependent mass-flux representation of
moist convective processes with the use of a simple bulk three-level cloud model, as
in Hack (1994). Specifically, in a vertically discrete model atmosphere where the
level index *k* decreases upward and considering the case where layers *k* and *k*+1 are
moist adiabatically unstable, the Hack scheme assumes the existence of a
non-entraining convective element with roots in level *k*+1, condensation and rain out
processes in level *k*, and limited detrainment in level *k*-1. By repeated application of
this procedure from the bottom of the model to the top, the thermodynamic structure
is locally stabilized.
The Hack shallow cumulus scheme can also be active in moist turbulent mixing,
such as stratocumulus entrainment, which has different physical characteristics than
cumulus convection. Shallow cumulus is usually regarded as a decoupled BL regime
in which the vertical mixing processes do not achieve a single well-mixed layer, while
the stratocumulus regime represents a well-mixed BL up to cloud top. The decoupling
criterion to distinguish between the two regimes is of great importance for simulating
the stratocumulus-to-cumulus transition (Bretherton and Wyant, 1997; Wood and
Bretherton, 2004). A number of these decoupling criteria have been explored, such as
static stability (Klein and Hartmann, 1993) and buoyancy flux integral ratio (Turton
and Nicholls, 1987). In the light of its robustness, the stability criterion with a
threshold of 17.5 K is introduced into the Hack scheme. The lower tropospheric
stability ($LTS$) is defined as

$$LTS = \theta_{700hPa} - \theta_{sfc},$$  (10)


where $\theta_{700hPa}$ and $\theta_{sfc}$ are potential temperatures at 700 hPa and at surface,
respectively. In BCC-CSM2-HR, the modified Hack scheme is activated only in the
decoupled BL regimes with $LTS < 17.5$ K below 700 hPa to remove adiabatically
moist instability, and the original Hack scheme (Hack, 1993) is still retained above
700 hPa to remove any local instability as long as the two adjacent model layers are
moist adiabatically unstable. This modification to the triggering of shallow convection
is found very useful to improve the simulation of the ITCZ precipitation (Lu et al.,
2020b).
**2.2 Land surface model**

BCC-AVIM2 is a comprehensive land surface model developed and maintained

in BCC. Its previous   version BCC-AVIM1 was used as the land component in
BCC-CSM1.1m participating in CMIP5 (Wu et al., 2013), which includes major land
surface biophysical processes treated similarly as in the Community Land Model
version 3.0 (CLM3, Oleson et al., 2004) developed at the National Center for
Atmospheric Research (NCAR), and plant physiological processes (Ji, 1995; Ji et al.,
2008), with 10 layers for soil and up to five layers for snow. The land component in

BCC-CSM2-MR and BCC-CSM2-HR is BCC-AVIM version 2 (Li et al., 2019). Updates in BCC-AVIM2 from its precedent version BCC-AVIM1 include a replacement of the water-only lake module by the common land model lake module (CoLM-lake) with a more realistic snow–ice–water–soil framework, a parameterization scheme for rice paddies added in the vegetation module, renewed parameterizations of snow cover fraction and snow surface albedo to accommodate the varied snow aging effect during different stages of a snow season, a revised parameterization to calculate the threshold temperature to initiate freeze (thaw) of soil water (ice) rather than being fixed at 0 ℃ in BCC-AVIM1, a prognostic phenology scheme for vegetation growth instead of empirically prescribed dates for leaf onset/fall, and a renewed scheme to depict solar radiation transfer through the vegetation canopy. Details of the updating are given in Li et al. (2019). BCC-AVIM2 implemented in BCC-CSM2-MR and is identical to what implemented in BCC-CSM2-HR, except horizontal resolution (same as in their atmosphere component) and the corresponding sub-grid surface classification.

**2.3 Ocean and Sea Ice Models**

The ocean component of BCC-CSM2-MR is MOM4-L40, developed by the Geophysical Fluid Dynamics Laboratory (GFDL, Griffies et al., 2005). It has a nominal resolution of $1°\times 1°$ with a tri-pole grid, and the actual resolution is from $1/3°$ latitude between 10 ℃ and 10 ℃ to $1°$ at $60°$ latitude. There are 40 levels in the vertical. More details of its implementation can be found in Wu et al. (2019).

The ocean component of BCC-CSM2-HR is MOM5, also developed by GFDL (Griffies, 2012). The model is based on the hydrostatic primitive equations and uses the Boussinesq approximation. The model uses Arakawa B-grid in the horizontal, with a globally uniform $0.25°$ resolution. The quasi-horizontal rescaled height coordinate, namely, z* vertical coordinate is employed to enhance flexibility of model applications, which allows for the free surface to fluctuate to values as large as the local ocean depth. There are 50 levels in the vertical, with a resolution of 10 m in the upper ocean and 367 m at the bottom. The tracer advection scheme used in both the

horizontal and vertical is the multi-dimensional piecewise parabolic method (MDPPM,
Marshall et al., 1997), which is of higher order and more accurate (less dissipative).
MOM5 has a complete set of physical processes with advanced parameterization
schemes. Effect of mesoscale eddies through the neutral diffusion scheme of Griffies
et al. (1998) is not included in this work. The K-profile parameterization (KPP) is
used to parameterize ocean surface boundary layer processes (Large et al., 1994).
MOM5 uses the optical scheme of Manizza et al. (2005) to define the light attenuation
exponentials. SeaWiFS chlorophyll-a monthly climatology is used in the calculation
of the attenuation of shortwave radiation entering the ocean layers with a maximum
depth set at 200m. The re-stratification effect of sub-mesoscale eddies in the ocean
surface mixed layer are parameterized with the sub-mesoscale scheme of Fox-Kemper
et al. (2008) and Fox-Kemper et al. (2011).
The sea-ice component of BCC-CSM2-HR and BCC-CSM2-MR is SIS4
(Winton, 2000) and SIS5 (Delworth et al., 2006) that developed by GFDL,
respectively. Both SIS4 and SIS5 are the sea ice component of MOM4 and MOM5,
respectively, and have three vertical layers including one snow cover and two ice
layers of equal thickness. They operate on the same oceanic grid of MOM4 in
BCC-CSM2-MR and MOM5 in BCC-CSM2-HR, respectively. There are up to five
categories of sea ice on each model grid for SIS4 and SI5 according to the thickness
of sea ice, and the mutual transformation from one category to another are taken into
account under thermodynamic conditions. Both SIS4 and SIS5 employ the scheme of
Semtner (1976) for the vertical thermodynamics and contains full dynamics with
internal ice forces calculated using an elastic-viscous-plastic rheology.
**3. Experimental design and data used**
**3.1 Historical simulation**
The principal simulation to be analyzed is the CMIP6 historical run (hereafter
referred to as historical) with prescribed forcings from 1850 to 2014 for
BCC-CSM2-MR and from 1950 to 2014 for BCC-CSM2-HR. All historical forcings
are from the CMIP6-recommended data (https://esgf-node.llnl.gov/search/input4mips/)

including: (1) Greenhouse gases concentrations such as $CO_2$, $N_2O$, $CH_4$, CFC11 and CFC12 with zonal-mean values and updated monthly; (2) Annual means of total solar irradiance derived from the CMIP6 solar forcing; (3) Stratospheric aerosols from volcanoes; (4) CMIP6-recommended tropospheric aerosol optical properties due to anthropogenic emissions that are formulated in terms of nine spatial plumes associated with different major anthropogenic source regions using version 2 of the Max Planck Institute Aerosol Climatology Simple Plume model (MACv2-SP, Stevens et al., 2017); (5) Time-varying gridded ozone concentrations; (6) Yearly global gridded land-use forcing. In addition, aerosol masses based on CMIP5 (Taylor et al., 2012) are also used for the on-line calculation of cloud droplet effective radius in our models.

The historical simulation of BCC-CSM2-MR follows the requirement of CMIP6, with the preindustrial initial state obtained after a 500-year piControl simulation. It covers the whole period from 1850 to 2014 (Wu et al., 2019). The simulation of BCC-CSM2-HR covers a shorter historical period from 1950 to 2014. Its initial state is the final state from a 50-year control simulation with fixed historical forcing of the year 1950, following the HighResMIP protocol. The control run itself is initiated from the states of individual components with their uncoupled mode. That is, the state of atmosphere and land are obtained from a 10-year AMIP run forced with monthly climatology of sea surface temperature (SST) and sea ice concentration, while the states of ocean (MOM5) and sea ice (SIS5) are derived from a 1000-year forced run with a repeating annual cycle of monthly climatology of atmospheric state from the Coordinated Ocean-Ice Reference Experiment (CORE) dataset version 2 (Danabasoglu et al., 2014).

**3.2 Data used for evaluations**

We choose the same period of 1950-2014 from both BCC-CSM2-MR and BCC-CSM2-HR historical simulations to evaluate their performance against observation-based or reanalysis data.

The 1950-2014 monthly global $1°\times1°$ gridded surface temperature from the Hadley Centre–Climatic Research Unit (HadCRUT version 4.6.0.0, available at

https://www.metoffice.gov.uk/hadobs/ hadcrut4/) is used to evaluate the global warming trend from BCC-CSM2-MR and BCC-CSM2-HR. HadCRUT (Morice et al., 2012) is a dataset combining land surface air temperature from the Climatic Research Unit (CRUTEM) and Hadley Centre Sea Ice and Sea Surface Temperature (HadISST). CRUTEM is derived from air temperatures near the land surface recorded at weather stations across the globe (Harris et al., 2013). HadISST contains global $1°\times1°$ sea ice concentration and SST, including in-situ measurements from ships and buoys (Rayner et al., 2003).

For the evaluation of present-day mean climate over the globe and major climate variabilities in the tropics, we choose the recent past 20 years of 1995-2014 as our reference period which will be observed as close as possible for observation-based or reanalysis data, described as follows.

(a) The 2001-2014 monthly global $1°\times1°$ gridded net radiations at top-of-atmosphere (TOA) from CERES-EBAF version 4.1 products (Loeb et al., 2018, available at https://asdc.larc.nasa.gov/project/CERES/CERES_EBAF_Edition4.1) are used to evaluate the global energy budget in models. CERES-EBAF data are derived on the basis of satellite observation from CERES (Clouds and Earth's Radiant Energy System) and synthesized with EBAF (Energy Balanced and Filled) data. Satellite observation is a direct monitoring of the net radiation at TOA, and a primary source of data for estimating Earth's energy balance (Wielicki et al, 1996).

(b) The 1995-2014 monthly global $0.25°\times0.25°$ gridded atmospheric temperature and wind from the fifth generation of ECMWF (the European Centre for Medium-Range Weather Forecasts) atmospheric reanalyses (ERA5, Hersbach and Dee 2016) and the climatological data of global zonal mean temperature and wind above the 1-hPa level to 0.1 hPa at $5°$ latitudes interval from the COSPAR (Committee on Space Research) International Reference Atmosphere (CIRA86) are used to evaluate the vertical structure of atmospheric temperature and wind. The 1995-2014 monthly global gridded wind data from ERA5 are also used to evaluate the quasi-biennial oscillation (QBO) of the equatorial zonal wind

between easterlies and westerlies in the tropical stratosphere. CIRA-86 (available at https://catalogue.ceda.ac.uk/uuid/4996e5b2f53ce0b1f2072adadaeda262) includes a global climatology of zonal atmospheric temperature and velocity extending from pole to pole on a 5-degree latitude grid and 0-120 km approximately at 2 km vertical resolution. It is derived from a combination of satellite, radiosonde and ground-based measurements (Fleming et al., 1990).

(c) The 1995−2014 monthly global observed precipitation at 2.5 ° resolution is taken from the Global Precipitation Climatology Project (GPCP version 2.2; Adler et al., 2003) dataset and used to evaluate the global distribution of precipitation climatology.

(d) The 2001-2014 quasi-global (60° N–60° S) $0.1° \times 0.1°$ gridded half-hourly precipitation estimates of Global Precipitation Measurement (GPM) Integrated Multi-satellitE Retrievals for GPM (IMERG) products (available at https://gpm1.gesdisc.eosdis.nasa.gov/data/GPM_L3/GPM_3IMERGHH.06/) are used to derive 3-hourly data, and then to evaluate the spectrum of precipitation intensity. IMERG uses inter-calibrated estimates from the international constellation of precipitation-relevant satellites and other data sources, including surface precipitation gauge analyses (Huffman et al., 2019).

(e) Two datasets (CRUTEM and HadISST) of the 1995-2014 monthly global 1 °×1 ° gridded surface temperature for the land (Jones et al., 2012) and ocean (Rayner et al., 2003), and gridded sea ice concentration are used to evaluate the model biases of land and ocean temperatures as well as sea ice cover. For the assessment of the ENSO cycle variation, a longer period of 1950-2014 is used from the global monthly HadISST dataset.

(f) The 1995 to 2014 daily global 0.25 °×0.25 ° wind from ERA5, daily global 2.5 °×2.5 ° outgoing longwave radiation (OLR) from NOAA (Liebmann and Smith, 1996), and daily global 2.5 °×2.5 ° precipitation from GPCP (Adler et al., 2003) are used to diagnose the Madden-Julian Oscillation (MJO), which is the dominant mode of sub-seasonal variability in the tropical troposphere (Madden and Julian, 1971). All the data firstly undergo the 20–100-day band-pass-filter.

An analysis of multivariate empirical orthogonal functions (EOFs) and principal
components (PCs) is then performed on intra-seasonal OLR, 850-hPa and
200-hPa zonal wind anomalies averaged over 10°S–10°N. Eight MJO phases
defined by the inverse tangent of the ratio of PC2 to PC1 as in Wheeler and
Hendon (2004) are also reconstructed.
(g) The 1995–2014 6-hourly tropical cyclones observations from International Best
Track Archive for Climate Stewardship (IBTrACS; Knapp et al., 2010) provide
information of all tropical cyclones, including latitude-longitude position,
minimum central pressure, and maximum sustained winds (instantaneous values)
at a time frequency of every 6 hours. We use the multiple criteria reported by
Murakami (2014) to detect TCs with 6-hourly outputs from models
(instantaneous values from BCC-CSM2-HR, but accumulated values from
BCC-CSM2-MR). (1) The maximum of relative vorticity of a TC-like vortex at
850 hPa exceeds $15 \times 10^{-5}$ s$^{-1}$ (a threshold that can vary from $1 \times 10^{-5}$ s$^{-1}$ to $15 \times$
$10^{-5}$ s$^{-1}$ in function of resolution (Murakami, 2014). (2) The warm-core above the
TC-like vortex, which is presented as the sum of the air temperature deviations
(subtracting the maximum temperature from the mean temperature within the
TC-like vortex center for an area of 10°×10°) at 300, 500 and 700 hPa, exceeds
0.8 K, a threshold falling in the range 0.6~1.0K that are recommended in
Murakami (2014); (3) The maximum wind speed at 850 hPa is higher than that at
300 hPa; (4) The maximum wind speed at 10 m within the TC-like vortex center
for an area of 3°×3° grid is higher than 10 m s$^{-1}$; (5) The genesis position of the
TC-like vortex is over the ocean; (6) The duration of the TC-like vortex satisfied
above conditions exceeds 48 hours.

## 491  4. Results

Data analysis and visualization are generally on the original or native grid of
observation and models. An exception is on the assessment of models' biases with
contrast to observation. In this case, simulations are re-gridded onto the grid of
corresponding observation.

**4.1 Global mean surface air temperature variations from 1950 to 2014**

The historical simulation from 1950 to 2014 allows us to evaluate the ability of models to reproduce the global warming of near surface temperature. Figure 2 presents global-mean surface air temperature evolutions for HadCRUT4 data and the two BCC models, in which the climatological mean is calculated for the reference period 1961–1990 and removed from the time series to better reveal long-term trends. The interannual variability of both simulations is qualitatively comparable to that observed, and the correlation coefficients reach to 0.84 in both models. A remarkable feature in Figure 2 is the presence of a global warming hiatus or pause for the period from 1998 to 2013 when the observed global surface air temperature warming slowed down. It is interesting that both models reproduce a hiatus, from 2002 to 2010 in BCC-CSM2-MR and from 2004 to 2012 in BCC-CSM2-HR. This warming hiatus is a hot topic (e.g. Fyfe et al., 2016; Medhaug et al., 2017; Wu et al., 2019), largely debated in the scientific research community. The reason why the BCC models simulate the recent global warming hiatus is beyond the scope of this paper and will be explored in other works.

**4.2 Global energy budget**

It is to be noted that only the period 2001–2014 is available for CERES-EBAF. For the consistency of comparison, we also shortened data from models and keep the same time interval as in CERES-EBAF. As shown in Table 2, the globally-averaged TOA net energy is $2.12\pm0.40$ W $m^{-2}$ in BCC-CSM2-MR and $1.51\pm0.57$ W $m^{-2}$ in BCC-CSM2-HR for the same period from 2001 to 2014. The energy equilibrium of the whole earth system in BCC-CSM2-HR is slightly improved. The TOA shortwave and longwave components for clear sky in BCC-CSM2-HR are also much closer to CERES-EBAF than in BCC-CSM2-MR. We noted that the TOA shortwave and longwave components for all sky in BCC-CSM2-HR gets lower than CERES-EBAF data and are not improved from BCC-CSM2-MR. This is related to cloud radiative forcing. Clouds constitute a major modulator of the radiative transfer in the atmosphere, and their radiative properties exert strong impacts on the equilibrium and

variation of the radiative budget at TOA. The globally-averaged shortwave cloud radiative forcing in BCC-CSM2-HR is slightly stronger than that in CERES-EBAF (-47.16±0.24 W $m^{-2}$) about 3 W $m^{-2}$ of cooling effect, and the globally-averaged longwave cloud radiative forcing in BCC-CSM2-HR is also stronger than the CERES-EBAF data (25.99±0.25 W $m^{-2}$) near 2 W $m^{-2}$ of warming effect (biases). The globally-averaged shortwave and longwave cloud radiative forcing in BCC-CSM2-MR are much closer to CERES-EBAF.

The obvious biases of model with contrast to CERES-EBAF are mainly located in the mid-latitudes and subtropics. Figure 3 shows the annual and zonal mean of shortwave, longwave and net cloud radiative forcing for the two model versions and observations. The longwave and net cloud radiative forcing are overall consistent with CERES-EBAF in most latitudes. In mid-latitudes of both the hemispheres, the shortwave cloud radiative forcing from BCC-CSM2-HR is much closer to CERES-EBAF than from BCC-CSM2-MR. But in low latitudes between 30 °S and 30 °N, BCC-CSM2-HR simulates excessive cloud shortwave radiative forcing which mainly comes from evident biases over the eastern tropical Pacific and tropical Atlantic oceans (Figure 4). These biases are possibly attributable to new treatments for boundary layer processes.

**4.3 Present-day mean climate**

**4.3.1   Vertical structure of the atmosphere temperature and wind**

Figure 5 presents zonally averaged vertical profiles of air temperature and zonal wind for December-January-February (DJF) and June-July-August (JJA) as simulated by BCC-CSM2-MR and BCC-CSM2-HR, with contrast to the ERA5 reanalysis below the 1-hPa level (Hersbach and Dee 2016) and climatological values above the 1-hPa level from CIRA86 (Fleming et al., 1990). The observed vertical profile of atmospheric temperature shows a clear structure of stratification, with an evident seasonal transition. In DJF, it is characterized as cool layers over broader latitudes spanning the transition from troposphere to stratosphere over the Northern Hemisphere, and warm layers spanning from the top of the stratosphere to mesosphere

over the Southern Hemisphere. Those different vertical structures in both hemispheres during DJF are almost reversed in JJA. BCC-CSM2-HR is capable of capturing the structure of upper stratosphere and the transition to mesosphere while BCC-CSM2-MR cannot.

Figure 6 shows biases of the zonally-averaged annual air temperature, relative to ERA5. Only model data from 5 hPa to 1000 hPa are evaluated as there are spare station-based observations above 5 hPa and it is generally recognized that most of stations don't reach their best-practice altitude of 5 hPa (https://gcos.wmo.int/en/atmospheric-observation-panel-climate). Temperature biases in lower to middle troposphere are relatively small, about -2K to 2K in BCC-CSM2-MR and -1K to 1 K in BCC-CSM2-HR in most latitudes, except in the southern polar region where temperature below 700 hPa are extrapolated values for ERA5 observation and models. The two models BCC-CSM2-MR and BCC-CSM2-HR have a cold bias of air temperature that appears near the tropopause and extends to the stratosphere in the subpolar and polar regions. There is also a thicker layer of warm biases in the lower stratosphere over the tropics and mid-latitude. Those temperature biases are not really reduced in BCC-CSM2-HR with a higher horizontal resolution. The cold bias in the troposphere was also reported in many CMIP5 models (see Charlton-Perez et al., 2013; Tian et al., 2013),

As shown in Figure 5, the basic pattern of vertical structures of westerly and easterly zones and their changes in DJF and JJA are generally well simulated by BCC-CSM2-MR and BCC-CSM2-HR. Both models have westerly wind biases of annual means that are located in the upper troposphere and stratosphere near 60 °S and 60 °N (Figures 6b and 6d), and reflect the meridional structure of temperature biases (Figures 6a and 6c) in accordance with the thermal–wind relationship.

**4.3.2 Precipitation**

Figure 7 shows the spatial distribution of DJF and JJA mean precipitation for BCC-CSM2-MR and BCC-CSM2-HR, compared to GPCP data. The two versions of BCC-CSMs were both able to reproduce the global observed precipitation patterns

and there is an evident improvement in the high-resolution model (BCC-CSM2-HR). Improvements are particularly clear in the Pacific, Indian, and Atlantic Oceans. The double-ITCZ issue is one of the most significant biases that persists in many climate models (e.g., Hwang and Frierson, 2013; Li and Xie, 2014). It exists in BCC-CSM2-MR, with excessive precipitation in the South Pacific Convergence Zone (SPCZ). This bias almost disappears in BCC-CSM2-HR. As shown in Figure 8, there is too much precipitation along the southern intertropical convergence zone (ITCZ) in BCC-CSM2-MR, which is mainly caused by excessive precipitation in the southern intertropical zone in DJF. This systematic bias is evidently reduced in BCC-CSM2-HR, especially with weakened precipitation in the South Pacific Convergence Zone (SPCZ). The improvement of SPCZ precipitation in BCC-CSM2-HR might be attributed to the implementation of the UWMT scheme which improved the simulation of low-level clouds over the tropical eastern South Pacific (Lu et al., 2020b) and reduced warm biases there (Fig. 10c). But the intensity of precipitation in the northern intertropical convergence zone in BCC-CSM2-HR is stronger than that from GPCP, which is partly attributed to the excessive precipitation in the tropical oceans, especially in the eastern tropical North Pacific (Figure 7e). A strong negative bias of JJA precipitation over the Amazon region exists in the two models. In Figure 7f, we also noted that the amount of JJA precipitation in east of the Philippines and near the Pacific warm pool is worsened, since it is smaller in BCC-CSM2-HR than in BCC-CSM2-MR and GPCP data. This bias of lacking precipitation in BCC-CSM2-HR may partly be caused by a cold-SST bias over the western Pacific warm pool (Fig.10c).

Figure 9 shows the probability density of 3-hourly precipitation between 40°S and 40°N in function of precipitation intensity with intervals of 1 mm/hour. The frequency of light rainfall events, smaller than 1 mm/hour, in BCC-CSM2-MR is higher than in IMERG. But strong precipitation events exceeding 10 mm/hour, are clearly insufficient. This is a common bias in many global climate models raising concerns for any studies on precipitation extremes. Compared to BCC-CSM2-MR, BCC-CSM2-HR with resolution increased shows substantial improvements for its

precipitation spectrum: reduced light rainfall and enhanced heavy rainfall events. The
spectral distribution of precipitation in BCC-CSM2-HR is much closer to IMERG.
**4.3.3  SST**
Figure 10 shows a spatial-distribution map of the 1995-2014 annual mean SST for
HadISST and the biases for BCC-CSM2-MR and BCC-CSM2-HR relative to
HadISST. BCC-CSM2-MR is generally warmer, while BCC-CSM2-HR is colder than
what observed. A warm SST bias in BCC-CSM2-MR spreads throughout most oceans,
except the north Pacific and north Atlantic. Such warm biases do not appear in
BCC-CSM2-HR, and the cold SST biases in the eastern subtropical south Pacific are
possibly attributed to excessive clouds there, also manifested by strong cloud
shortwave radiative forcing (Figure 4e). The warm biases in the eastern tropical ocean
basins in BCC-CSM2-MR are associated with a deficit of stratiform low-level clouds,
a common and systematic bias for many climate models (Richter, 2015). The cold
biases there in BCC-CSM2-HR, similarly, are associated with too much low cloud,
except over the tropical north Pacific. We also noted that a belt of warm SST biases in
the Kuroshio extension and in the North Atlantic in both models (Figures 10b and
10c), especially in the high-resolution model. This bias may be partly resulted from
the coarse resolution of HadISST data used, as SST near the Kuroshio shows strong
temperature gradients with filamentous structures (Shi and Wang, 2020).
**4.3.4  Land-surface air temperature**
Figure 11 shows the simulation biases of annual mean land-surface air
temperature from BCC-CSM2-MR and BCC-CSM2-HR. The near-surface air
temperature over land in BCC-CSM2-MR is generally colder than the CRUTEM
observations, particularly exhibiting severe cold biases in North Europe. As there are
no physical (but only resolution) changes in the land modeling component in the two
models, the systematic biases of near-surface air temperature over land are very
similar to each other. Increasing atmospheric resolution in BCC-CSM2-HR does not
seem to show amelioration, and the surface air temperatures in BCC-CSM2-HR
exhibits rather similar patterns as in BCC-CSM2-MR with biases of -2 to 2 K in most
land regions between 50 °N and 50 °S compared to CRU data.

### 4.3.5 Sea ice

Figure 12 shows the annual mean sea ice concentration simulated by
BCC-CSM2-MR and BCC-CSM2-HR over the period 1995–2014, compared to
HadISST observation data. The simulated geographic distribution of sea ice in the
Arctic is overall realistic, except that the sea ice concentration in the Atlantic is
slightly overestimated in both models. This overestimation of sea ice possibly has a
consequence for the severe cold biases of surface air temperature in North Europe
(Figure 11). In the Antarctic, sea ice concentration simulated by BCC-CSM2-MR is
smaller than HadISST data, especially from 60 °W to 60 °E in the subpolar region
where the simulated SST is warmer compared to HadISST data (Figure 10b). Those
deficiencies in BCC-CSM2-MR (Figure 12e) are largely reduced in BCC-CSM2-HR
(Figure 12f).
Figure 13 shows the monthly sea ice covers for the Arctic and Antarctic from
BCC-CSM2-MR and BCC-CSM2-HR. HadISST observations show that the Arctic
sea ice cover reaches a minimum extent of $6.9 \times 10^6$ km$^2$ in September and rises to a
maximum extent of $16.0 \times 10^6$ km$^2$ in March, and the Antarctic sea ice cover reaches a
minimum extent in February and a maximum extent in September. The seasonal cycle
amplitude and phase of sea ice area are well captured by the two models, and their
biases are mostly smaller than $1 \times 10^6$ km$^2$ while compared to HadISST observations.
We note that the extents of the Arctic sea ice for each month in BCC-CSM2-MR are
slightly but systematically smaller than HadISST, and in the Antarctic are smaller in
February and March but larger in other months than HadISST. BCC-CSM2-HR
slightly overestimated sea ice concentration by about $1 \times 10^6$ km$^2$ in both hemispheres
with reference to HadISST.

### 4.4 Variabilities in the Tropics

The tropical cyclone (TC), also known as typhoon or hurricane, is among the most
destructive weather phenomena. The Madden-Julian Oscillation (MJO) is the
dominant mode of sub-seasonal variability in the tropical troposphere (Madden and
Julian, 1971), and the quasi-biennial oscillation (QBO) is a quasiperiodic oscillation
of the equatorial zonal wind between easterlies and westerlies in the tropical
stratosphere. TC, MJO and QBO are very important variabilities in the tropics, with
consequences to global weather and climate.
**4.4.1 Tropical Cyclones**
In Figure 14, we evaluate the average TC frequency over twenty years
(1995-2014) from BCC-CSM2-MR and BCC-CSM2-HR, with contrast to the
climatology of 1995-2014 observations from International Best Track Archive for
Climate Stewardship (IBTrACS; Knapp et al., 2010). It is clear that TC activity is
increased with resolution enhanced. The averaged total global TC numbers per year
are 49.6 in BCC-CSM2-MR and 94.4 in BCC-CSM2-HR, and the global TC numbers
in BCC-CSM2-HR is much closer to the IBTrACS observation (90.2). The global TC
number is slightly influenced by the threshold ($15 \times 10^{-5}$ $s^{-1}$ in Figure 14) of relative
vorticity at 850 hPa used to detect TC. If this threshold gets looser to $5 \times 10^{-5}$ $s^{-1}$, the
averaged total global TC numbers per year in BCC-CSM2-MR and BCC-CSM2-HR
would enhance to 55.9 and 101.5 (not shown), respectively. The low TC number in
BCC-CSM2-MR is furthermore explained by the fact that its 6-hourly data used to
detect TC are averaged values in the 6-hour interval, while instantaneous values
would be more appropriate as in IBTrACS and BCC-CSM2-HR. Spatially,
BCC-CSM2-HR generates excess TC activity in the eastern North Pacific, Northern
Indian Ocean, and Southern Hemisphere. But both models severely underestimate TC
activity in the North Atlantic and in the Caribbean Sea. The general overestimation of
TC activity in the eastern North Pacific and the opposite in the North Atlantic in
BCC-CSM2-HR may be related to the warmer SST in the eastern tropical North
Pacific and colder SST in the tropical Atlantic with contrast to HadISST data (Figure
10c), but other factors such as the entrainment in the parameterization of convection
(Zhao et al., 2012) and air-sea coupling (Li and Sriver, 2018) may also have an
influence. The study of Li and Sriver (2018) showed that ocean coupling influences
simulated TC frequency, geographical distributions, and storm intensity, and TC
tracks are relatively sparse in the coupled simulations than in un-coupled simulations.
Figure 15 shows the maximum surface wind speed versus minimum sea level
pressure for the tropical cyclones that are derived from the 1995-2014 IBTrACS
observation (black dots and line), and simulations of BCC-CSM2-MR and
BCC-CSM2-HR. Here, the maximum surface wind speed (minimum sea level
pressure) of a given TC was defined as the instantaneous maximum (minimum) of the
6-hours interval in IBTrACS and BCC-CSM2-HR, but averaged value in
BCC-CSM2-MR for wind speed at 10m (sea level pressure). Instantaneous values of
wind speed and sea level pressure were not recorded as output in BCC-CSM2-MR.
Maximum wind speeds for TC lifetime in BCC-CSM2-MR are consistently weaker
than BCC-CSM2-HR and IBTrACS, which is understandable given the coarser
resolution. BCC-CSM2-MR cannot capture strong storms, and maximum wind speeds
at 10m only reach to 30 m s$^{-1}$. BCC-CSM2-HR, as expected, can reproduce those
strong TCs for which minimum pressure of TC lifetime may reach to 960 hPa and
maximum wind speed at 10m may reach to 50 m s$^{-1}$. The fitting line of maximum
wind speeds with minimum center pressures in BCC-CSM2-HR almost matches that
from IBTrACS observation (Figure 15). The BCC-CSM2-HR simulations just as
previous studies have shown (e.g., Murakami et al., 2012; Sugi et al., 2017; Vecchi et
al., 2019) demonstrate that the maximum wind speed of TC simulated by a model
with approximately 50 km resolution can reach up to 50~60 m s$^{-1}$.
**4.4.2 Madden–Julian Oscillation**
MJO is the dominant mode of sub-seasonal variability in the tropical troposphere
(Madden and Julian, 1971), and characterized by eastward propagation of deep
convective structures along the Equator with an average phase speed of around 5
m s$^{-1}$ at the intraseasonal time scale of 20–100 days (Wheeler and Kiladis, 1999).
MJO generally forms over the Indian Ocean, strengthens over the Pacific Ocean, and
weakens due to interaction with South America and cooler eastern Pacific SSTs
(Madden and Julian, 1971). Figure 16 gives the time lag-longitude evolution of 10 °S–
10 °N-averaged intraseasonal precipitation anomalies for the left panels and time
lag-longitude evolution of 80 °–100 °E-averaged intraseasonal precipitation anomalies
correlated against the precipitation over the equatorial eastern Indian Ocean for the
right pancels. Both versions of BCC-CSMs reasonably reproduce the eastward
propagating feature of convection from the Indian Ocean across the Maritime
Continent to the Pacific (Figs. 16b and 16c), as well as the apparent poleward
propagations from the equatorial Indian Ocean into the Northern Hemisphere and the
Southern Hemisphere (Figs. 16e and 16f). The signal of northward propagation is
more evident in BCC-CSM2-HR than in BCC-CSM2-MR. The average phase speed
of eastward propagation of deep convection in BCC-CSM2-HR is much closer to
GPCP data denoted by the dashed line in Fig 16c. Figure 16b shows that the eastward
propagation of deep convection in BCC-CSM2-MR is too fast, compared to GPCP
data.
MJO activity can be generally featured by a life cycle of eight phases (Wheeler
and Hendon, 2004). Intensity of outgoing longwave radiation (OLR) is often used for
this purpose to represent the activity of convection. Figure 17 shows the MJO
phase-latitude diagram of composited outgoing longwave radiation (OLR) and
850-hPa zonal wind anomalies averaged over 10 °S–10 °N. In observation, MJO
convection initiated from Africa and the western Indian Ocean at phases 1–2,
propagates eastward from the Indian Ocean across the Maritime Continent to the
western Pacific at phases 3–6, and finally disappears in the western hemisphere at
phases 7–8. BCC-CSM2-MR generally captures the evolution of convection with
MJO phases, but shows faster propagative speed and apparently underestimates the
intensity compared to the observation. In contrast, BCC-CSM2-HR shows an
obviously improved MJO phase transition and convection intensity.
**4.4.3  The stratospheric quasi-biennial oscillation**
The alternative oscillation between westerly and easterly winds in the tropical
stratosphere constitutes the characteristic feature of the quasi-biennial oscillation
(QBO). A good simulation of QBO still remains nowadays a challenge for all
state-of-the-art climate models. In a recent work, Kim et al. (2020) showed that only
half (15 out of 30) of the CMIP6 models can internally generate QBO
(BCC-CSM2-MR was in the good half). We should however recognize that there was
a huge progress in CMIP6, since in CMIP5 only five models (about 10% of the total)
were able to simulate a realistic QBO (Schenzinger et al., 2017).

To evaluate model performance in simulating the QBO, the time-height cross
sections of the tropical zonal winds averaged from 5°S to 5°N for BCC-CSM2-MR
and BCC-CSM2-HR are compared with contrast to the ERA5 reanalysis. As shown in
Figure 18, ERA5 shows alternative westerlies and easterlies in the lower stratosphere
with a mean periodicity of about 28 months. The two BCC models are both able to
generate a reasonable QBO, and the observed asymmetry with the easterlies being
stronger than the westerlies are also well reproduced. The general performance of
QBO in BCC-CSM2-MR was evaluated in Wu et al. (2019). A detailed assessment of
the underlying mechanism involving wave dynamics and the associated forcing to
drive QBO is presented in Lu et al. (2020a). The simulated QBO has stronger
amplitudes in BCC-CSM2-HR than in BCC-CSM2-MR. As the horizontal resolution
and physics package are changed from BCC-CSM2-MR to BCC-CSM2-HR, the
parameterized convective gravity wave forcing for QBO could be potentially
enhanced in BCC-CSM2-HR. On the other hand, changes in the convective cumulus
parameterization can also affect the simulation of the resolved convectively coupled
equatorial waves (i.e., the Kelvin wave) driving the QBO, and lead to stronger QBO
amplitudes in BCC-CSM2-HR.

In the two BCC models, the downward propagation of QBO occurs in a regular
manner, but does not sufficiently penetrate to low altitudes below 50 hPa. The vertical
resolution is similar below ~10 hPa in both BCC-CSM2-MR and BCC-CSM2-HR
(Figure 1). A further downward propagation to lower altitudes can be expected by
increasing the vertical resolution finer than 500 m to adequately resolve the
wave-mean flow interaction in the upper troposphere-lower stratosphere (Geller et al.
2016; Garcia and Richter 2019).
**4.4.4 Niño3.4 SST variability**

Figure 19 presents time series of the monthly Niño3.4 SST (5°N−5°S,

170°W−120°W) anomalies from BCC-CSM2-MR and BCC-CSM2-HR, with
reference to HadISST data from 1950 to 2014. The amplitude of interannual variation
of the Nino3.4 index in BCC-CSM2-HR and BCC-CSM2-MR are both stronger than
in HadISST. Those strong amplitudes may partly come from the slight warming
trends in both models. The power spectrum analysis of the Niño3.4 index from the
HadISST observations shows significant peaks at 4-6 years and 2-3 years. The
periodicity of the ENSO cycle in BCC-CSM2-MR is mainly at 2-3 years. It is
prolonged to 3-6 years in BCC-CSM2-HR. As in Figure 19h, the Niño3.4 SST
variability from HadISST data reaches its maximum in the period from November to
January. The phase locking (i.e., the preferred timing in the year for the peak of
ENSO) simulated by BCC-CSM2-MR occurs in autumn. The simulated ENSO phase
locking in BCC-CSM2-HR is partly improved, since ENSO events tend to reach their
maximum toward winter.
Recent studies of Hayashi et al. (2020) showed that the ability to simulate the
asymmetry between warm (El Niño) and cold (La Niña) phases as recorded in
observations is still very poor for most CMIP5 and CMIP6 models. This imperfection
also exists in both BCC-CSM2-HR and BCC-CSM2-MR. The asymmetry in SST
anomalies is often measured by the normalized third statistical moment, i.e., skewness
(Burgers and Stephenson, 1999). Figures 19d-f show spatial maps of the skewness of
monthly SST anomalies (SSTA) in the tropical Pacific that are calculated following
the methodology in Burgers and Stephenson (1999). In the eastern Pacific, the ENSO
signal from HadISST data is the strongest and the observed SSTA skewness is highly
positive (Fig. 19d) due to the presence of extreme El Niño events and absence of
extreme La Niña events. The skewness values of SSTA in both models (Figs. 19e and
19f) are underestimated with contrast to HadISST observation, and the area of
positive skewness in the eastern tropical Pacific from BCC-CSM2-HR simulations is
much closer to HadISST data.
Figure 20 presents the spatial patterns of correlation coefficients between the
Niño3.4 index and the corresponding global SST anomalies from 1950 to 2014 for the
HadISST observation and the two BCC models. Both BCC-CSM2-HR and
BCC-CSM2-MR simulate a positive correlation structure over the equatorial region of
the central and eastern Pacific, which is consistent with the analysis from HadISST
despite an over extension into the western Pacific. The HadISST data show clearly
that the zone of positive correlation of SST with the Niño3.4 index in the equatorial
eastern Pacific expands to extra-tropics. Especially along the eastern border of the
Pacific, the areas of high values of positive correlations in BCC-CSM2-HR are larger
than BCC-CSM2-MR, and much closer to HadISST. Compared to BCC-CSM2-MR,
BCC-CSM2-HR improves the simulation in the equatorial Indian Ocean and the
eastern tropical Atlantic where there are also remarkable areas of positive correlation.
We also note that areas of negative correlation of SST with the Niño3.4 index in the
western equatorial Pacific extend to the south and north Pacific in HadISST, which is
clearer in BCC-CSM2-HR than in BCC-CSM2-MR.
**5. Conclusions and discussions**
This paper was devoted to the presentation of the high-resolution version
BCC-CSM2-HR and to the description of its climate simulation performance. We
focused on its updating and differential characteristics from its predecessor, the
medium-resolution version BCC-CSM2-MR. BCC-CSM2-HR is our model version
participating in HighResMIP, while BCC-CSM2-MR is our basic model version
participating in   other CMIP6-endorsed MIPs (Wu et al., 2019; Xin et al. 2019).
The atmosphere resolution is increased from T106L46 in BCC-CSM2-MR to
T266L56 in BCC-CSM2-HR, and the ocean resolution from $1°×1°$ to $1/4°×1/4°$. A
few novel developments were implemented in BCC-CSM2-HR for both the
dynamical core and model physics in the atmospheric component: First, a
spatially-varying damping for the divergence field was used to improve the
atmospheric temperature simulation in the stratosphere at polar areas. It helps to
control high-frequency noise in the stratosphere and above; Second, the deep cumulus
convection scheme originally described in Wu (2012) was further ameliorated to
allow detrained cloud water be transported to adjacent grids and downward to lower
troposphere; Third, we modified the relevant schemes for the boundary layer
turbulence and shallow cumulus convection to improve the simulation of ITCZ
precipitation; Fourth, the UWMT scheme is used to improve the simulation of
low-level clouds over eastern basins of subtropical oceans. The land model
configuration in BCC-CSM2-HR is the same as in BCC-CSM2-MR. Major land
surface biophysical and plant physiological processes of BCC-AVIM2 implemented
in BCC-CSM2-MR and BCC-CSM2-HR keep the same, and only differences are in
the sub-grid surface classification. The ocean component of BCC-CSM2-HR is
upgraded from MOM4 in BCC-CSM2-MR to MOM5. The sea ice component is also
updated from SIS4 to SIS5.
For the sake of a rigorous comparison, historical simulations with fully coupled
BCC-CSM2-MR and BCC-CSM2-HR are analyzed over a 65 year period from 1950
to 2014. The long-term trends of 1950-2014 globally-averaged annual-mean surface
air temperature from both BCC-CSM2-MR and BCC-CSM2-HR are highly correlated
to HadCRUT4 observation. The global warming in the latter half of the $20^{th}$ century is
well simulated, and the observed global warming hiatus or slowdown in the period
from 1998 to 2013 is generally captured by both model versions.
We compared the 1995-2014 basic climate features in relation to atmospheric
temperature, circulation, precipitation, surface temperature, and sea ice between the
two simulations and we evaluated them against observation-based and reanalysis data.
With contrast to the medium-resolution BCC-CSM2-MR, the high-resolution
BCC-CSM2-HR has slightly improved energy equilibrium for the whole earth system.
The global mean TOA net energy balance is about 1.51 W $m^{-2}$ in BCC-CSM2-HR for
the period from 1995 to 2014, showing an evident improvement compared to 2.12
W $m^{-2}$ in BCC-CSM2-MR. The longwave and net cloud radiative forcing are overall
consistent with CERES-EBAF in most latitudes, but excessive cloud radiative forcing
for shortwave radiation is found over the eastern tropical Pacific and tropical Atlantic
in BCC-CSM2-HR. Temperature biases in the low- to mid-troposphere below 300
hPa in BCC-CSM2-HR are relatively small, within the range of -1K to 1K. Both
versions of BCC-CSMs have a cold air temperature bias that appears above 250 hPa
in the subpolar and polar region, and a warm bias in the upper stratosphere in the
mid-latitudes, which caused westerly wind biases in the upper troposphere and in the
stratosphere. Although those prominent systematic biases in temperature and wind
seem relatively insensitive to changes in atmospheric resolution, the ability to capture
the winter to summer seasonal transition in the vertical structure of temperature and
wind in the upper stratosphere is strengthened in BCC-CSM2-HR.

The two versions of BCC-CSMs were both able to reproduce the observed global

precipitation patterns and there is a remarkable improvement in precipitation centers
over the Pacific, Indian, and Atlantic Oceans in the high-resolution model. The
double-ITCZ biases in BCC-CSM2-MR are reduced in BCC-CSM2-HR and
excessive precipitation in the South Pacific Convergence Zone is also strongly
reduced in BCC-CSM2-HR. The climatological SST in BCC-CSM2-HR, relative to
the observation-based HadISST data, shows cold biases but reduced compared to
BCC-CSM2-MR. Such SST cold biases are partly attributable to different ocean
components, MOM4 in BCC-CSM2-MR and MOM5 in BCC-CSM2-HR. The
seasonal cycles of amplitude and phase of sea ice in both hemispheres are generally
well captured in BCC-CSM2-HR, but with a small excess all year round in the
Northern Hemisphere, especially in the Atlantic.

We also conducted an assessment on a few important phenomena of the tropical

climate, such as TC (tropical cyclone), MJO (Madden-Julian oscillation), QBO
(quasi-biennial oscillation), and ENSO (El Nino – southern oscillation). The averaged
total number of global TC in BCC-CSM2-HR is a bit larger than in IBTrACS
observation. BCC-CSM2-HR can simulate main TC activities in the eastern North
Pacific, Northern Indian, and in the Southern Hemisphere but misses the TC activities
in the North Atlantic. BCC-CSM2-HR is able to capture a realistic MJO signal
including the eastward-propagating behavior of MJO and its phase speed. The
QBO-related alternative westerlies and easterlies in the tropical lower stratosphere
with a mean periodicity of about 28 months are well simulated. The weakness in
downward propagation of the simulated QBO (insufficient penetration of the signal to
low altitudes) in BCC-CSM2-MR is slightly improved in BCC-CSM2-HR. Main
features of the ENSO cycle such as the periodicity and phase locking are captured by
BCC-CSM2-HR although its main ENSO periodicity of 3-6 years is still shorter
compared to HadISST observations.

Our work shows that enhancing resolution does not noticeably improve climate

mean state and deterioration is even possible. For example, the decrease of JJA
precipitation over the warm pool in our high-resolution model is still an important
issue which certainly deserves further investigations with multiple models and
simulations. Actually, other studies also reported similar issues. Haarsma et al. (2020)
shows that increasing resolution in the EC-Earth model deteriorated the wet bias over
the western Pacific warm pool. Bacmeister et al. (2014) analysed the high-resolution
climate simulations performed with the Community Atmosphere Model (CAM), and
showed that dry bias over the same region with enhanced resolution. Over the western
Pacific warm pool, the atmospheric circulation and precipitation undergoes not only
the impact of tropical variations such as MJO and TC, but also strong regional air-sea
coupling.

We finally should note that there exist some systematic biases in our

high-resolution model, such as the excessive cloud radiative forcing for shortwave
radiation over the eastern tropical Pacific, cold biases in the near surface temperature
over North Europe and over the tropical Atlantic, insufficient TC activities over the
North Atlantic and the Caribbean Sea. These are all important issues motivating us to
develop and implement more physically-based parameterizations in our future work.
For the lack of sufficient TC activities in the North Atlantic, it seems that this bias
also exists in other models (e.g., Bell et al., 2013; Strachan et al., 2013; Small et al.,
2014) and still remains a challenging issue for the climate modelling community. A
recent study reported by Davis (2018) showed that models with horizontal grid
spacing of one fourth degree or coarser could not produce a realistic number of
category 4 and 5 storms in the tropical Atlantic. The spatial resolution even in our
current high-resolution model seems too coarse.


**Code and data availability**

Source codes of BCC-CSM-HR model can be accessed at a DOI repository

http://doi.org/10.5281/zenodo.4127457 (Wu et al., 2020b). Model output of BCC models for CMIP6 simulations described in this paper is distributed through the Earth System Grid Federation (ESGF) and freely accessible through the ESGF data portals after registration (http://doi.org/10.22033/ESGF/CMIP6.2921, Jie et al., 2020). Details about ESGF are presented on the CMIP Panel website at http://www.wcrp-climate.org/index.php/wgcm-cmip/about-cmip. All source code and data can also be accessed by contacting the corresponding author Tongwen Wu (twwu@cma.gov.cn).

**Author contributions**

Tongwen Wu led the BCC-CSM development, and all other co-authors contributed to it. Tongwen Wu, Weihua Jie, Xiaoge Xin, and Jie Zhang designed the reported experiments and carried them out. Tongwen Wu, Laurent Li, Yixiong Lu, Junchen Yao, and Fanghua Wu wrote the final document with contributions from all other authors.

**Competing interests**

The authors declare that they have no conflict of interest.

**Acknowledgements**

This work was supported by The National Key Research and Development Program of China (2016YFA0602100).

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

Table 1. Constituents and configurations of BCC-CSM2-MR and BCC-CSM2-HR.

| | | BCC-CSM2-MR | BCC-CSM2-HR |
|---|---|---|---|
| **Atmosphere component (BCC-AGCM3)** | **Resolution** | T106 (~110km), 46 layers with top layer at 1.979hPa and model lid at 1.459 hPa | T266 (~45km), 56 layers with top layer at 0.156 hPa and model lid at 0.092 hPa |
| | **Dynamical core** | Spectral framework described in Wu et al. (2008) | Same as in BCC-CSM2-MR but including spatially-varying divergence damping. |
| | **Deep convection** | A modified Wu'2012 scheme (Wu, 2012) described in Wu et al. (2019) | Revised Wu et al. (2019) scheme, including the effects of convective downdraft in neighboring grids. |
| | **Shallow/Middle Tropospheric Moist Convection** | Hack (1994) | Modified Hack (1994) scheme described in Lu et al. (2020b), incorporating a trigger based on lower tropospheric stability. |
| | **Cloud macrophysics** | Diagnosed cloud fraction described in Wu et al. (2019) | Revised Wu et al. (2019) scheme, excluding the special treatment for the marine stratocumulus. |
| | **Cloud microphysics** | Modified scheme of Rasch and Kristj´ansson (1998) by Zhang et al. (2003), but included the aerosol indirect effects in which liquid cloud droplet number concentration is diagnosed using the aerosols masses. | Same as in BCC-CSM2-MR. |
| | **Gravity wave drag** | Gravity wave drag generated by both orography (Mcfarlane 1987) and convection (Beres et al., 2004). | Same as in BCC-CSM2-MR, but using tuned parameters related to model resolutions. |
| | **Surface orographic drag** | No treatment. | The turbulent mountain stress scheme as in Richter et al. (2010). |
| | **Radiative transfer** | Radiative transfer scheme used in CAM3 (Collins et al., 2004), but including the aerosol indirect effects, and the effective radius of the cloud droplet for liquid clouds is diagnosed using liquid cloud droplet number concentration. | Same as in BCC-CSM2-MR. |
| | **Boundary Layer** | Parameterization of Holtslag and Boville (1993), but modified PBL height computation as in Zhang et al. (2014) | The University of Washington Moist Turbulence scheme (Bretherton and Park, 2009) |
| **Land surface component (BCC-AVIM2)** | **Resolution** | Horizontal resolution same as in the atmosphere component. 10 layers for soil and up to five layers for snow. | Horizontal resolution same as in the atmosphere component. 10 layers for soil and up to five layers for snow. |
| | **Biophysical process** | CLM3 (Oleson et al., 2004) | CLM3 (Oleson et al., 2004) |
| | **Plant physiological and Soil carbon-nitrogen dynamical processes** | BCC-AVIM2 (Li et al., 2019) | BCC-AVIM2 (Li et al., 2019) |
| **Ocean Component (MOM)** | **Resolution** | 1°×1° with a tri-pole grid, but 1/3° latitude between 30°S and 30°N to 1.0° at 60° latitude, 40 layers in vertical | 1/4°×1/4° with a tri-pole grid at north to 60°N, 50 layers in vertical |
| | **Tracer advection scheme** | MOM4 (Griffies, 2005), Sweby advection scheme (Sweby, 1984) | MOM5 (Griffies, 2012), multi-dimensional piecewise parabolic method |
| | **Neutral diffusion scheme** | Griffies et al. (1998) with a constant diffusivity of 600 $m^2 s^{-1}$ | None |
| | **Surface boundary layer processes** | K-profile parameterization (KPP, Large et al., 1994) | Same as in MOM4 |
| | **Submesoscale parameterization** | None | Fox-Kemper et al. (2008) |

| | | | |
|---|---|---|---|
| | scheme | | |
| | shortwave penetration | Morel and Antoine (1994), with the maximum depth of 100m | Manizza et al. (2005), with the maximum depth of 300m |
| **Sea Ice Component (SIS)** | **Resolution** | Same as in the ocean component MOM4, 3 vertical layers including 1 snow cover and 2 ice layers of equal thickness | Same as in the ocean component MOM5, 3 vertical layers including 1 snow cover and 2 ice layers of equal thickness |
| | **Model physics** | SIS4 (Winton, 2000), Elastic-viscous-plastic dynamical processes, Semtner's thermodynamic processes | SIS5 (Delworth et al., 2006), Elastic-viscous-plastic dynamical processes, Semtner's thermodynamic processes |
| | **Snow albedo** | 0.80 | 0.85 |
| | **Ice albedo** | 0.5826 | 0.68 |



Table 2. Energy balance and cloud radiative forcing at the top-of-atmosphere (TOA) in
the models with contrast to CERES-EBAF observations. Units: W m$^{-2}$.

| | BCC-CSM2-MR | BCC-CSM2-HR | CERES-EBAF |
|---|---|---|---|
| Net energy at TOA | 2.12±0.40 | 1.51±0.57 | 0.84 ±0.33 |
| TOA outgoing longwave radiative flux | 239.18±0.20 | 237.85±0.18 | 239.69 ±0.25 |
| TOA net shortwave radiative flux | 241.29±0.35 | 239.35±0.49 | 240.53 ±0.19 |
| TOA outgoing longwave radiative flux in clear sky | 265.10±0.20 | 265.28±0.22 | 265.67 ±0.37 |
| TOA net shortwave radiative flux in clear sky | 291.13±0.25 | 290.06±0.15 | 287.68 ±0.14 |
| TOA incoming shortwave radiation | 340.34±0.09 | 340.35±0.09 | 340.14 ±0.09 |
| Shortwave cloud radiative forcing | -49.83±0.27 | -50.71±0.48 | -47.16 ±0.24 |
| Longwave cloud radiative forcing | 25.92±0.08 | 27.43±0.11 | 25.99 ±0.25 |


Notes: Mean value and standard deviation are calculated from 2001-2014 yearly global means
of the simulations for BCC-CSM2-MR, BCC-CSM2-HR, and the CERES-EBAF Ed4.1
data set.



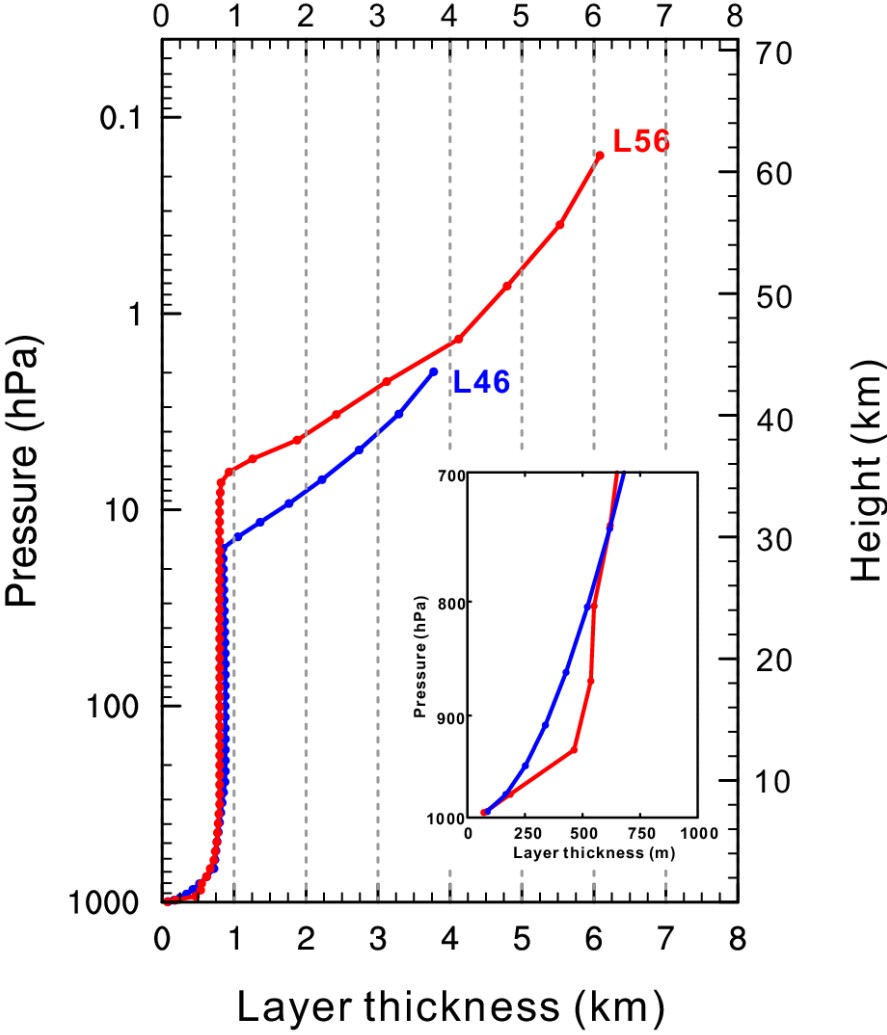



Figure 1. The profiles of layer thickness against height for 46 vertical layers in
BCC-CSM2-MR (blue) and 56 vertical layers in BCC-CSM2-HR (red).



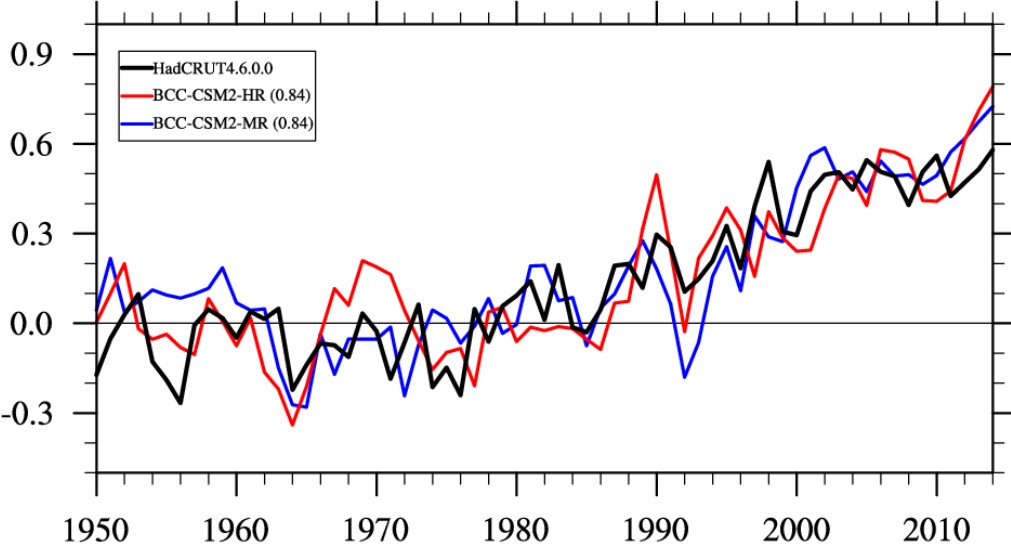


Figure 2. Time series of anomalies in the global mean surface air temperature from 1950 to
2014. The reference climate to deduce anomalies is for each individual curve from 1961 to
1990. The numbers in the parentheses denote the correlation coefficient of 11-year smoothed
simulations with HadCRUT4.6.0.0 (Morice et al., 2012) observation.





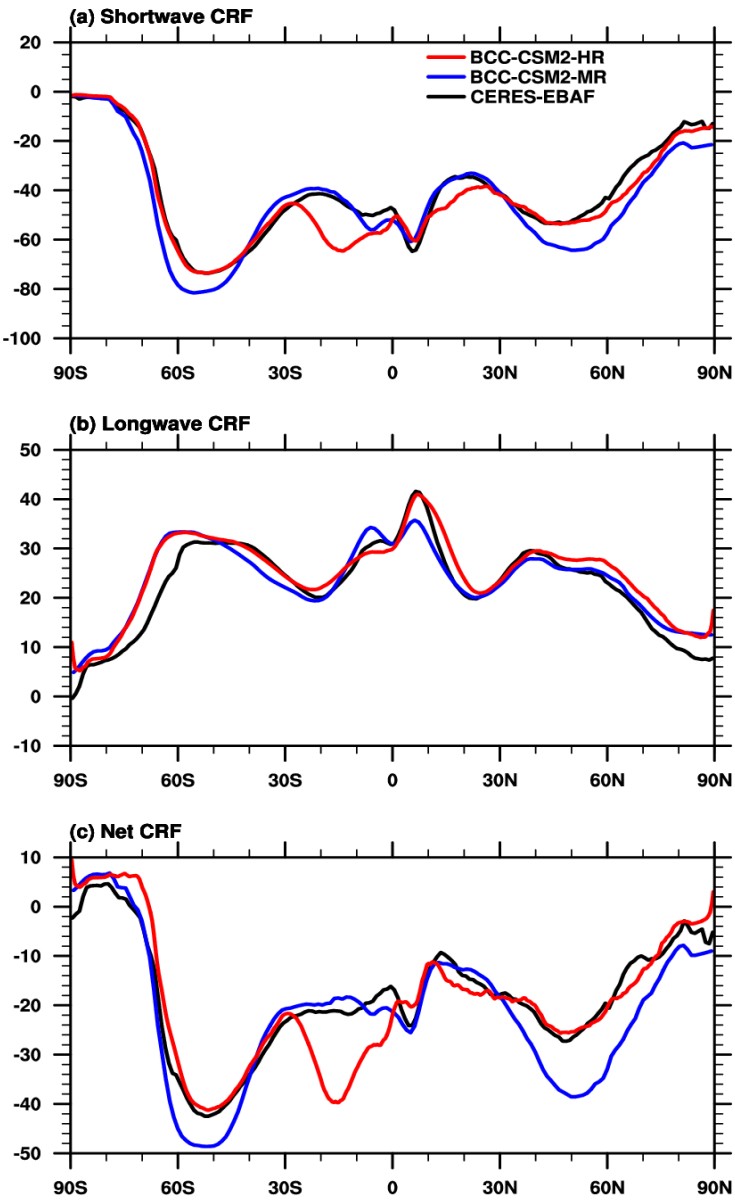



Figure 3. Zonal averages of (a) shortwave, (b) longwave, and (c) net cloud radiative forcing (CRF, in W m$^{-2}$) for the historical simulations (2001-2014) of BCC-CSM2-MR (blue lines) and BCC-CSM2-HR (red lines), compared to the 2001-2014 CERES-EBAF observations (black lines).



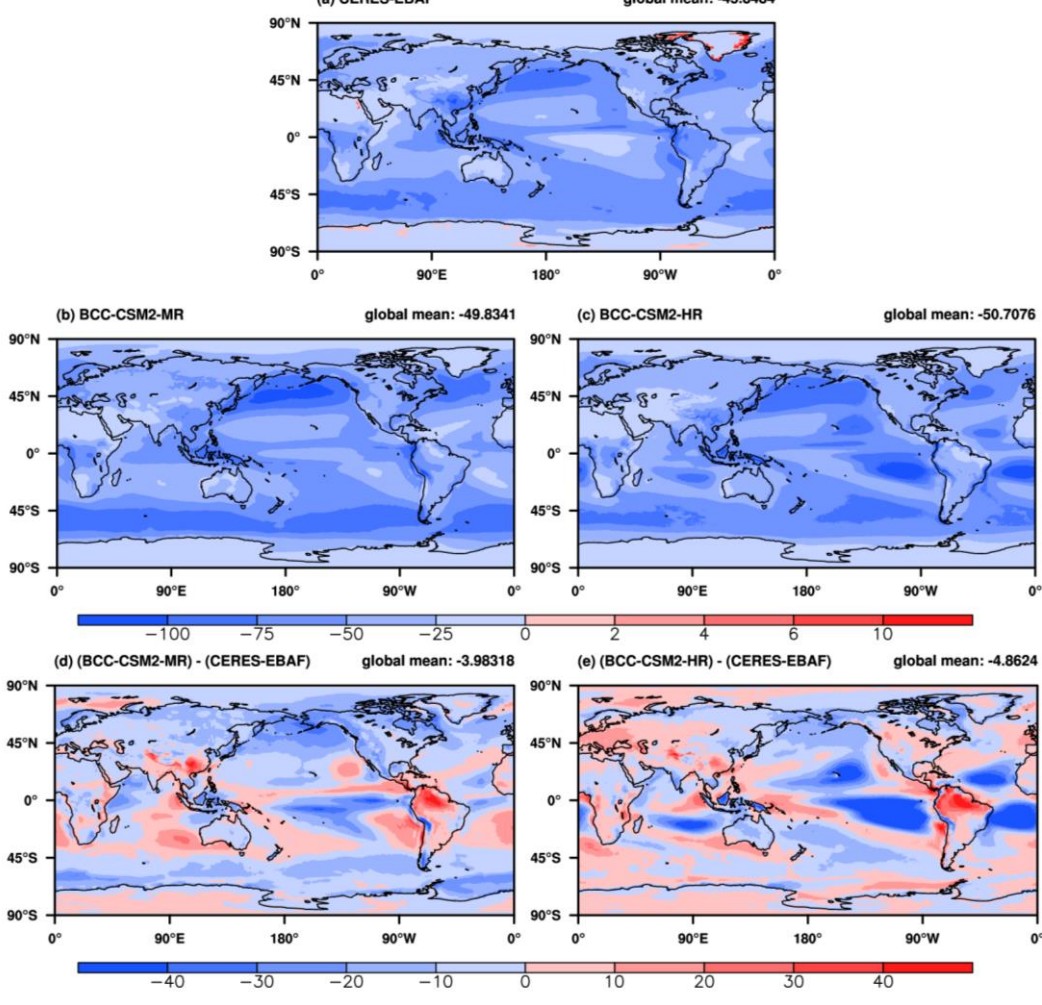



Figure 4. The 2001-2014 averaged annual-mean shortwave cloud radiative forcing for (a) the
CERES-EBAF observations, the historical simulations from (b) BCC-CSM2-MR and (c)
BCC-CSM2-HR and their biases (d and e) against CERES-EBAF data. Units: W m$^{-2}$.


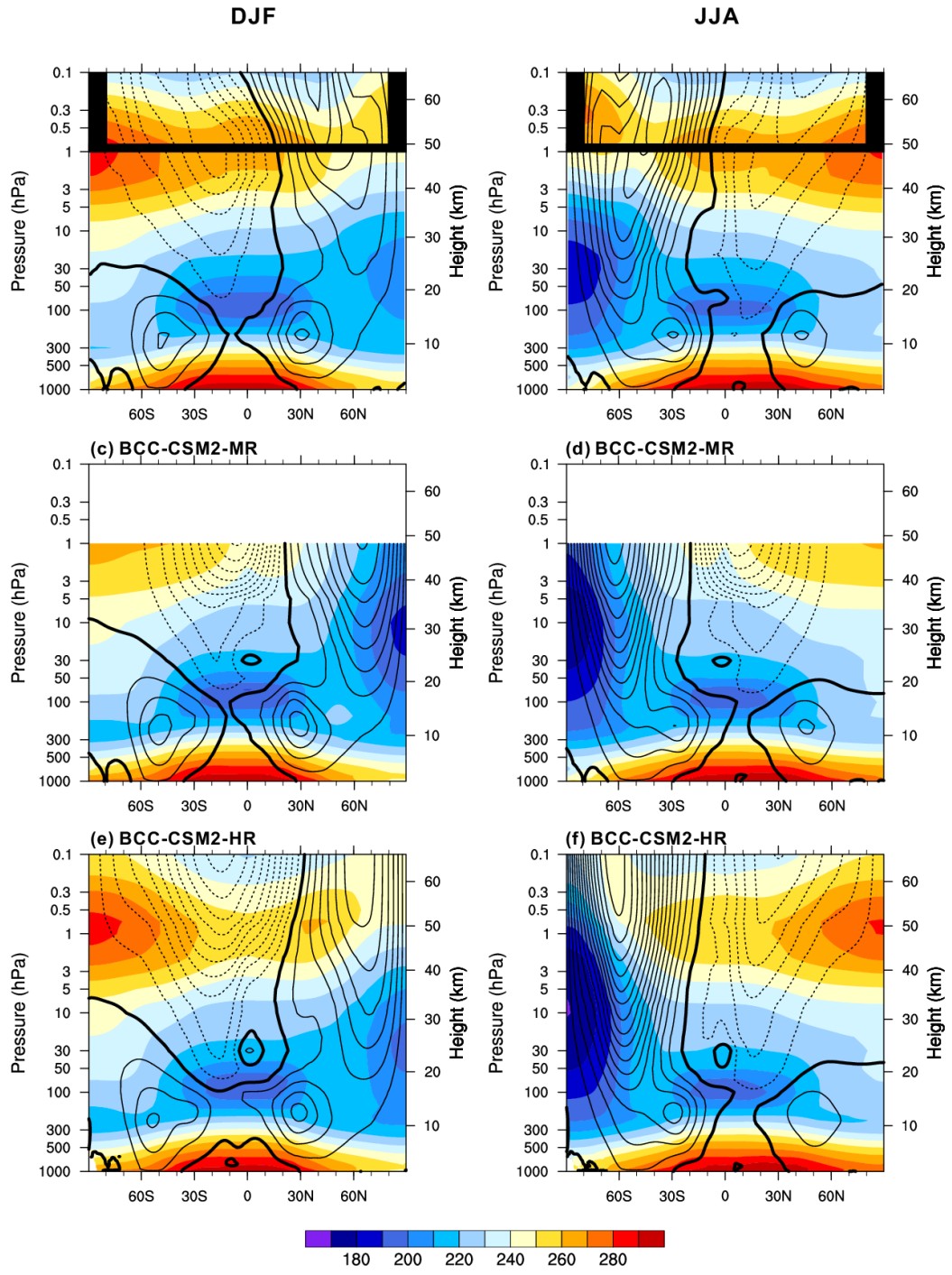

Figure 5. The zonal means of temperature (colors; K) and zonal wind (contours; m s$^{-1}$) averaged for December-January-February (left panel) and Jun-July-August (right panel) from 1995 to 2014 for (a,b) ERA5/CIRA86, (c,d) BCC-CSM2-MR, (e,f) BCC-CSM2-HR. Positive (negative) zonal winds are plotted with solid (dashed) lines with a contour interval of 10 m s$^{-1}$. Thick contour line denotes zero zonal wind speed. In (a) and (b), the values above 1 hPa from the COSPAR International Reference Atmosphere (CIRA86, Fleming et al., 1990) and below 1 hPa from the ERA5 reanalysis.


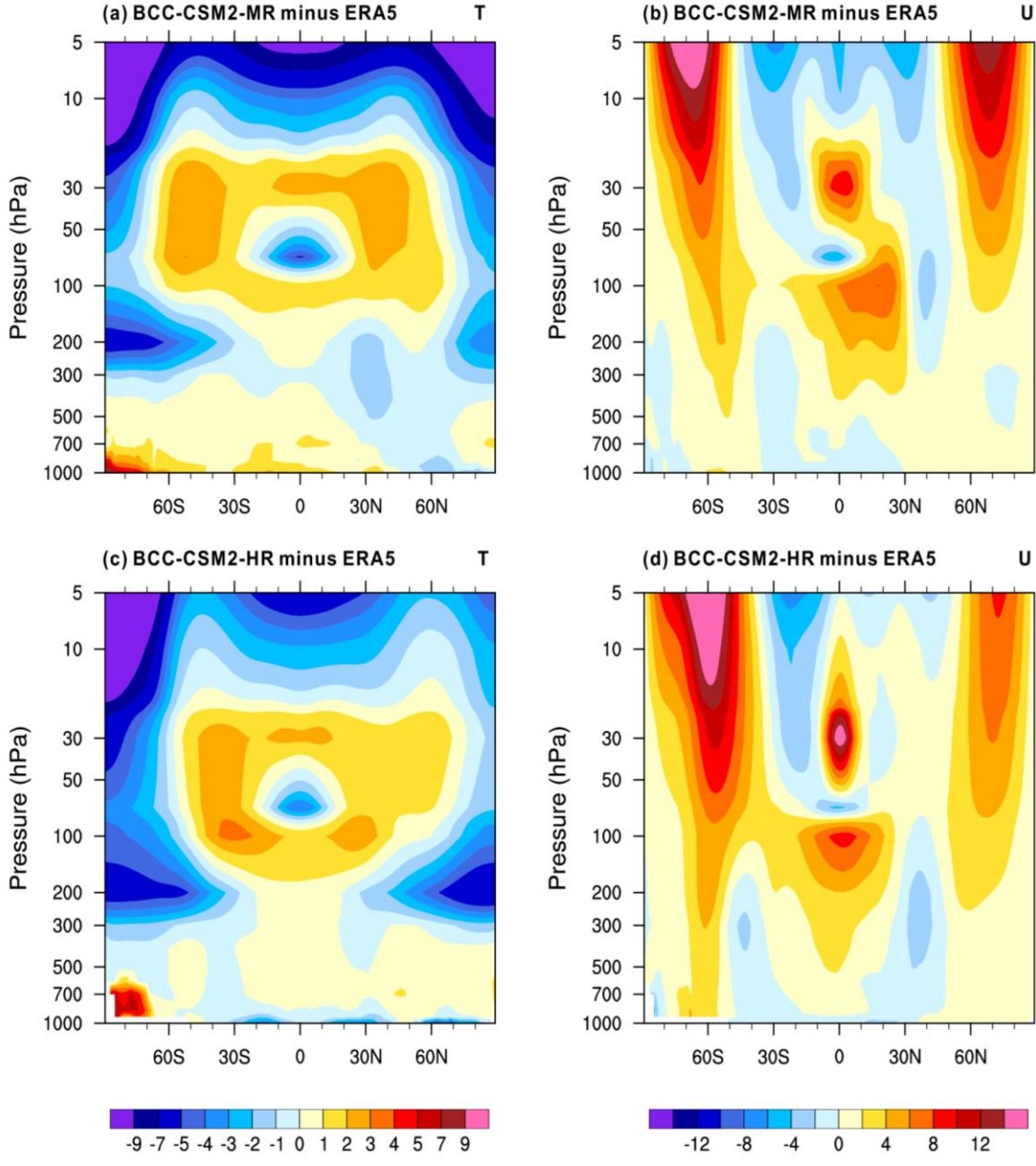



Figure 6. Zonally-averaged annual mean temperature biases (left panel, in K) and zonal wind biases (right panel, in m s$^{-1}$) averaged for the period from 1995 to 2014 for (a,b) BCC-CSM2-MR, and (c,d) BCC-CSM2-HR, with respect to the ERA5 reanalysis data.


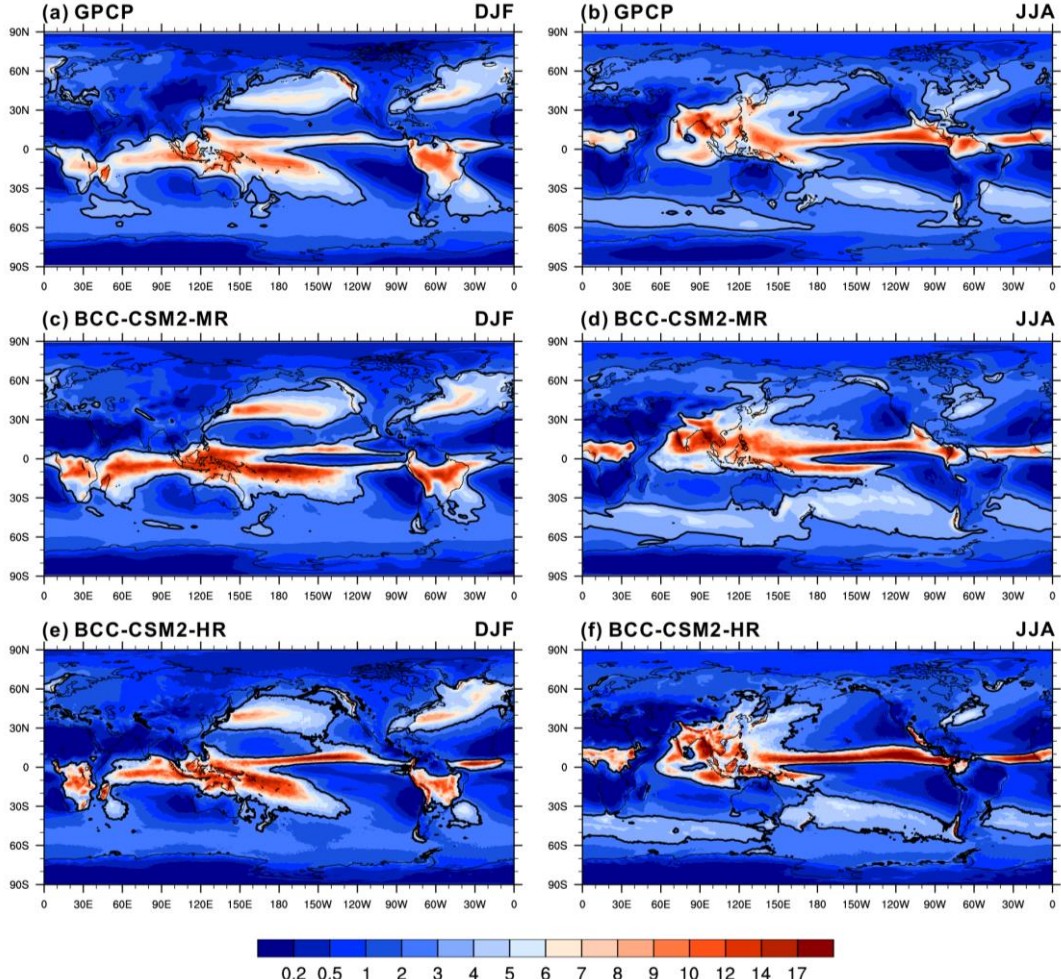

Figure 7. The 1995-2014 averaged mean precipitation rate of December-January-February (left panel) and June-July-August (right panel) for (a,b) GPCP observations, (c,d) BCC-CSM2-MR, and (e,f) BCC-CSM2-HR. Units: mm day$^{-1}$. The 3 mm day$^{-1}$ contour line is in bold as a reference to facilitate the visual inspection.




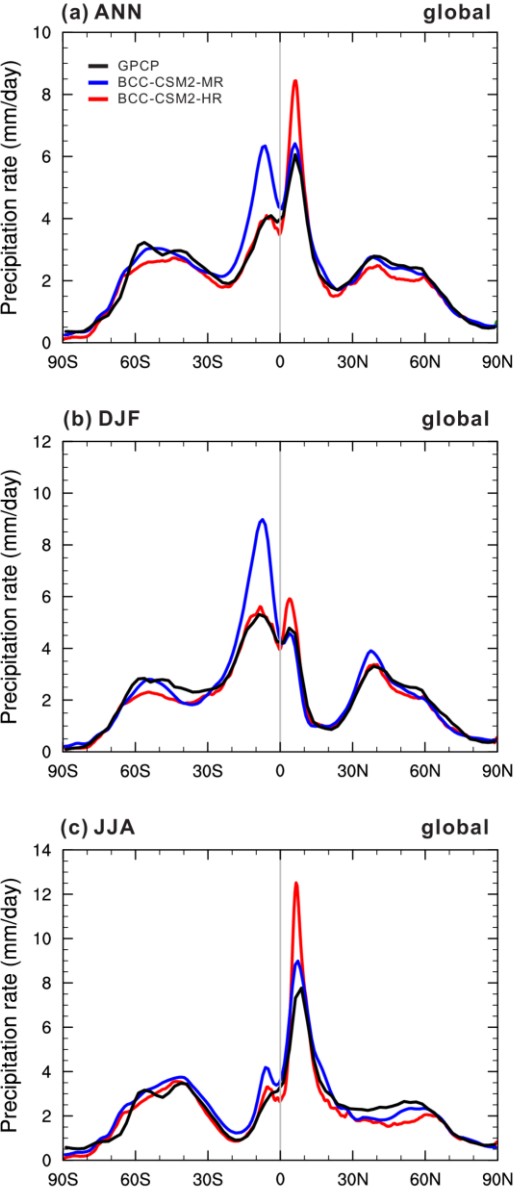



Figure 8. The 1995-2014 averaged zonal mean precipitation rate (mm day$^{-1}$) for (a) the annual
mean, (b) December-February-February, and (c) June-July-August. The solid black lines
denote GPCP data, and the color lines show BCC-CSM2-MR (blue) and BCC-CSM2-HR (red)
simulations. Units: mm day$^{-1}$.


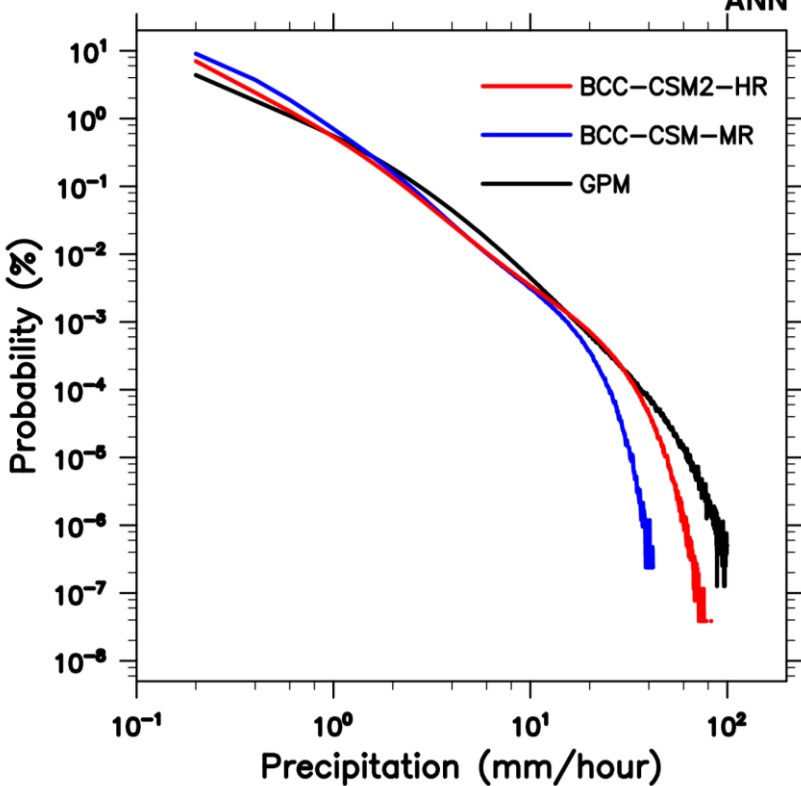

Figure 9. The probability density of 3-hourly precipitation between 40°S and 40°N
and during the period from 2001 to 2014, in function of precipitation intensity with
intervals of 1 mm/hour, for IMERG Precipitation (black line),BCC-CSM2-MR (blue
line) and BCC-CSM2-HR (red line), respectively. Two simulations were re-gridded to
the grid of IMERG before processing.



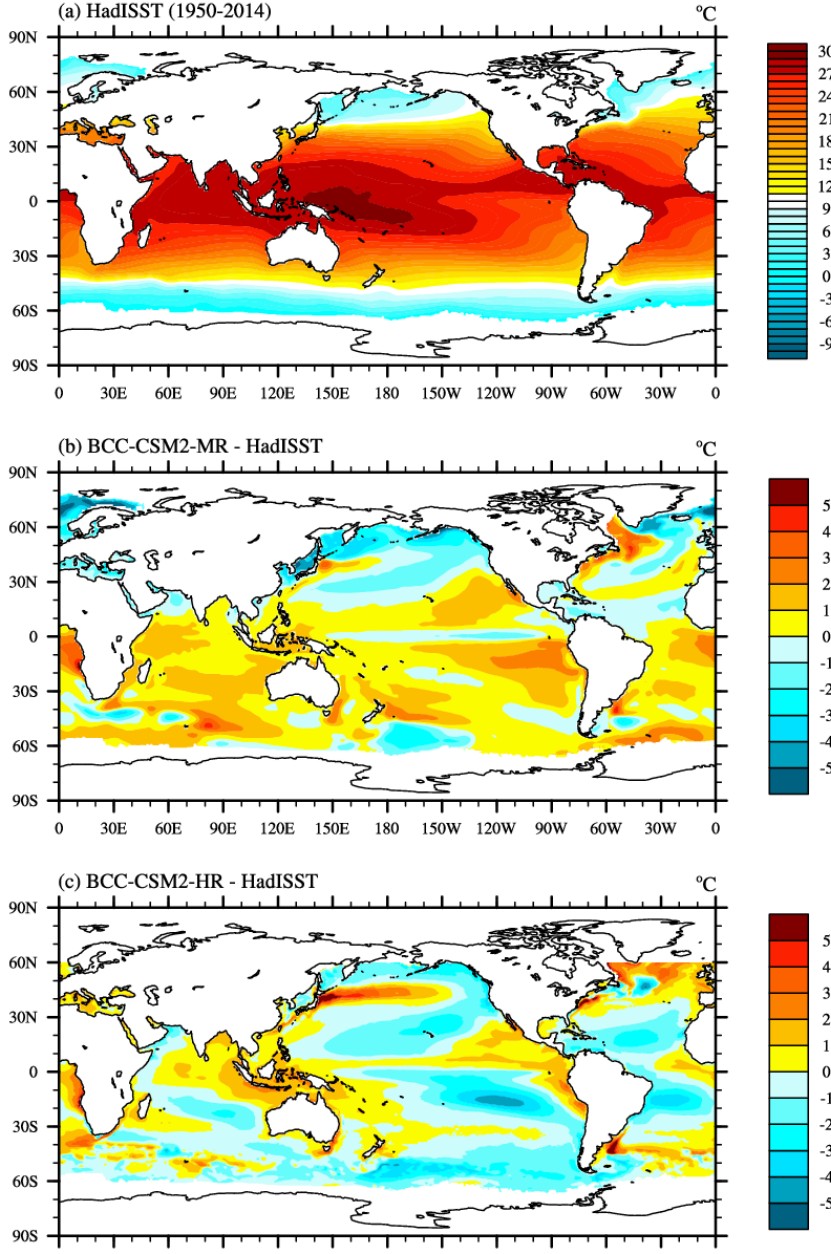



Figure 10. The global distributions of the 1995-2014 annual mean sea surface
temperature for (a) the observations from HadISST, and the simulation biases in (b)
BCC-CSM2-MR and (c) BCC-CSM2-HR.


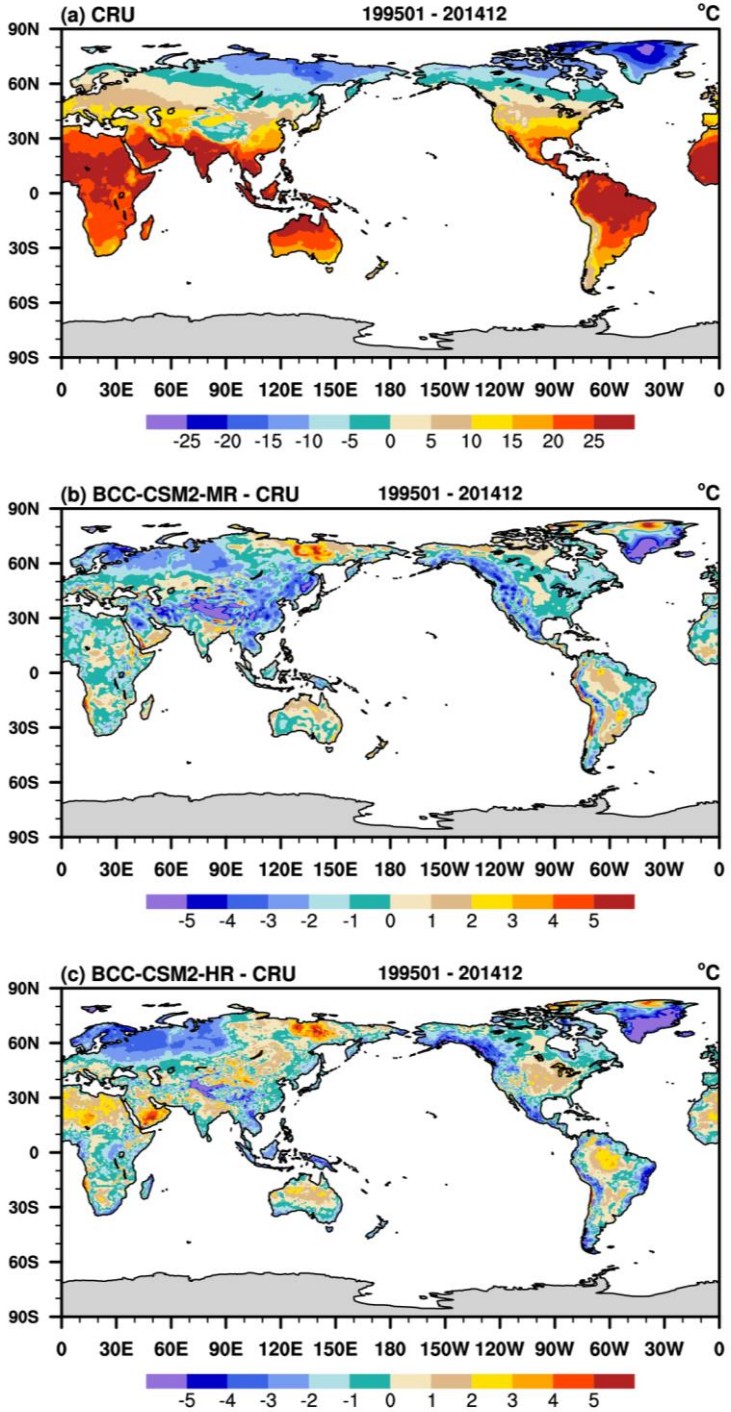


Figure 11. The 1995-2014 averaged global land-surface air temperature for (a)

CRUTEM observations, and simulation biases in (b) BCC-CSM2-MR and (c)

BCC-CSM2-HR.



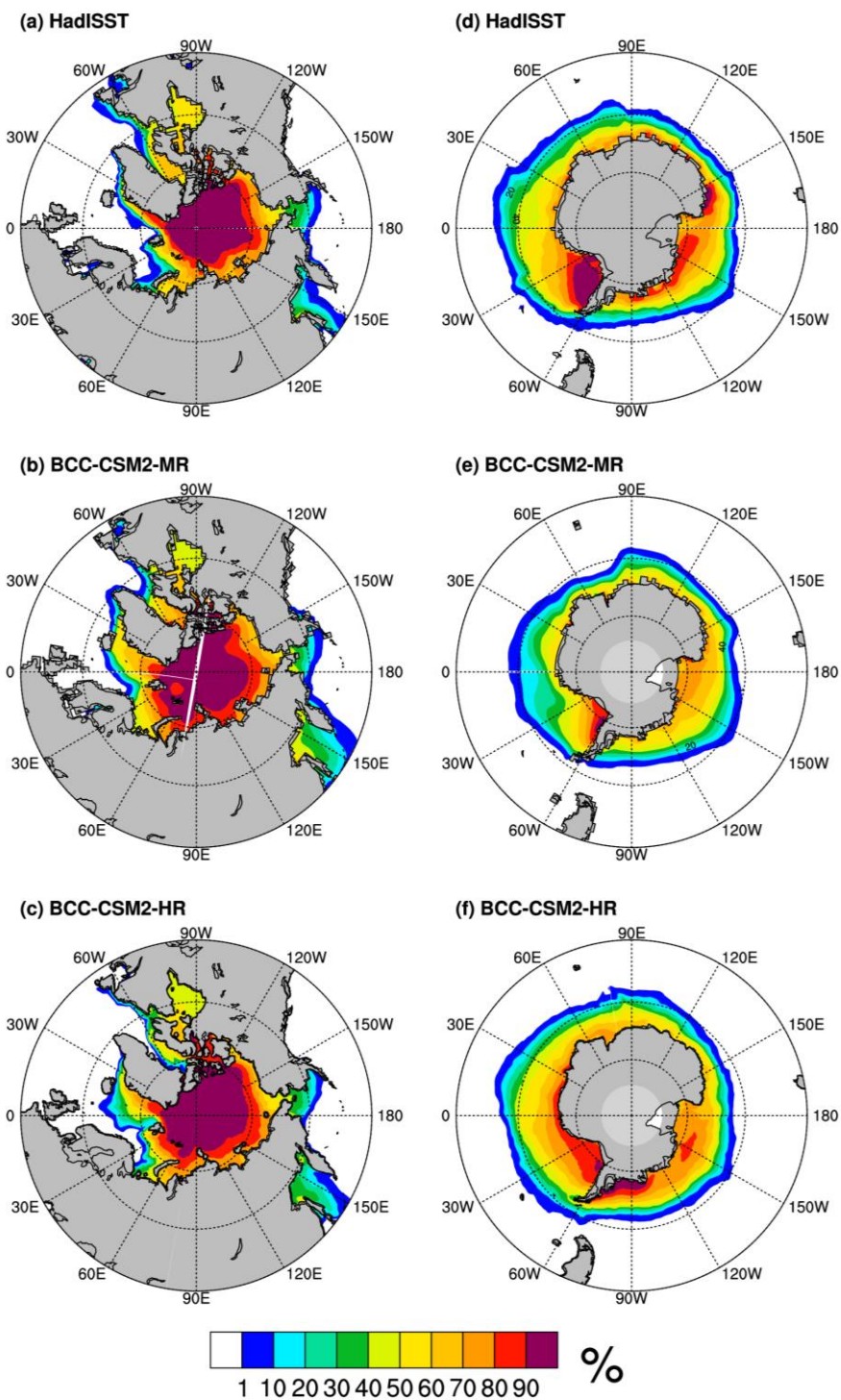


Figure 12. The spatial distribution of annual mean sea ice concentration from BCC-CSM2-MR and BCC-CSM2-HR with contrast to the observations from the Hadley Centre Sea Ice data set from 1995 to 2014.



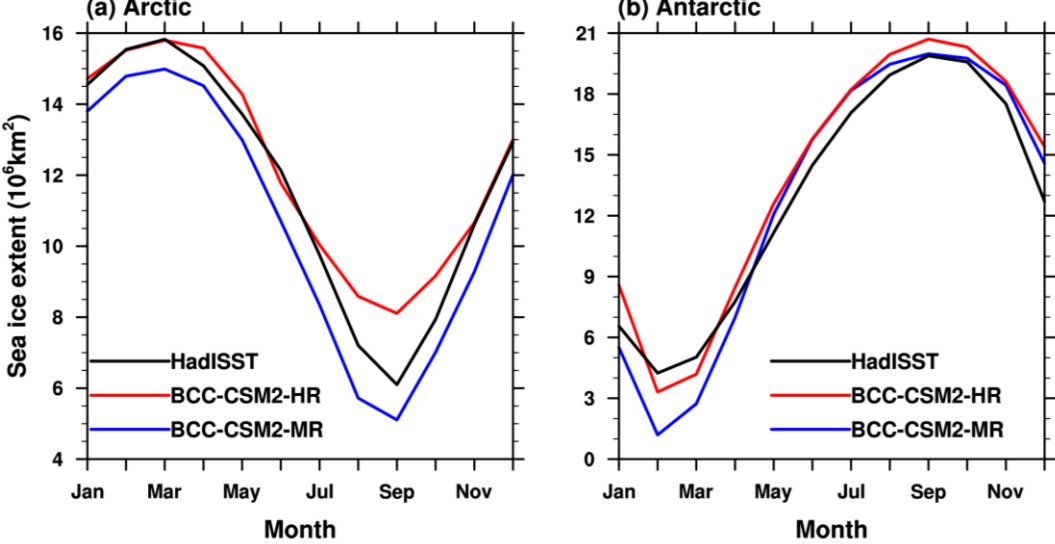

Figure 13. The mean (1995-2014) seasonal cycle of sea-ice extent (with a sea-ice
concentration of at least 15 %) in (a) the Northern Hemisphere and (b) the Southern
Hemisphere for the observations from the Hadley Centre Sea Ice and Sea Surface
Temperature dataset (black lines) and the simulations from BCC-CSM2-MR (blue
lines), BCC-CSM2-HR (red lines).

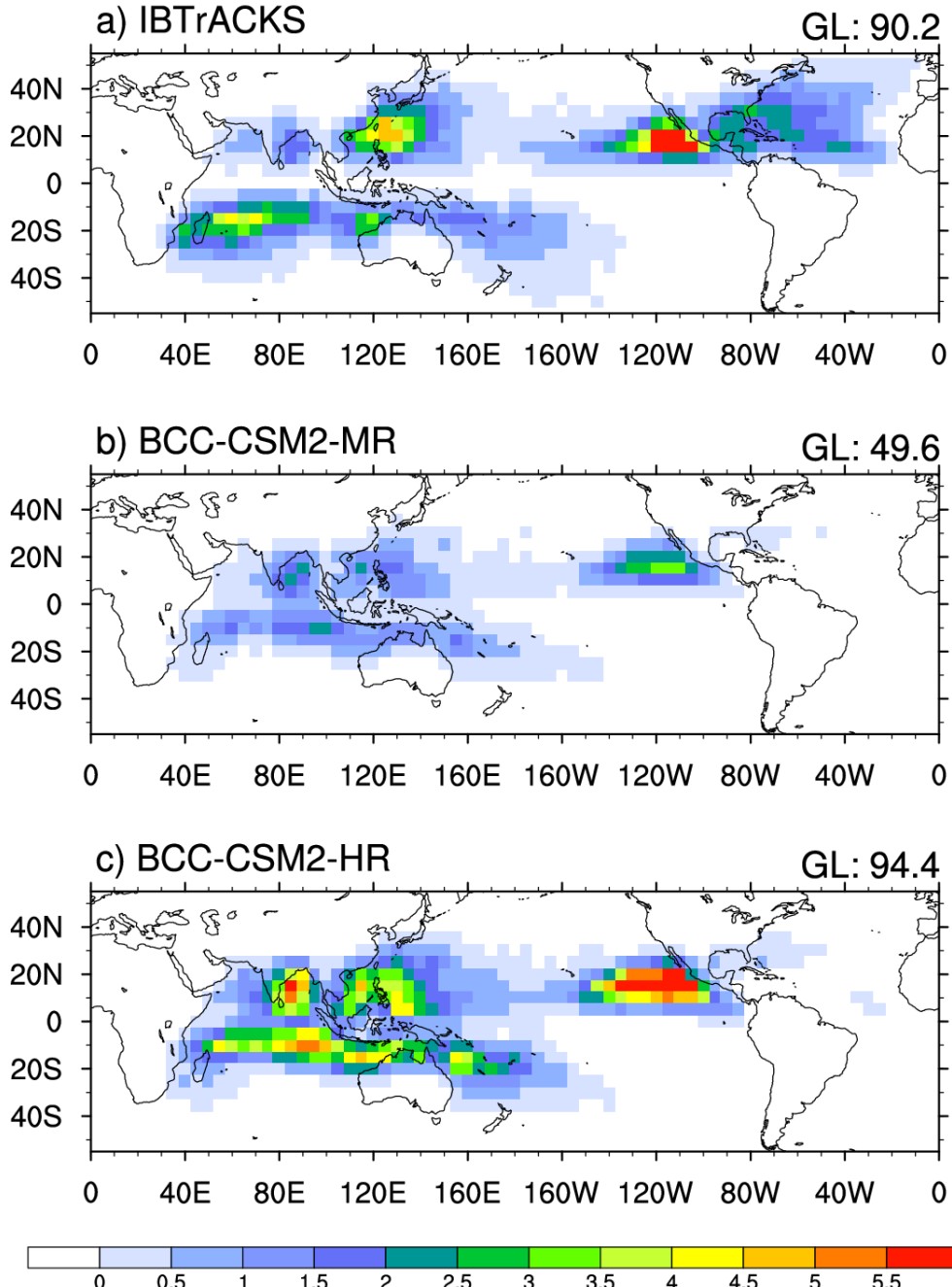


Figure 14. The global distribution of tropical cyclone (TC) densities (number per year)
averaged for the period of 1995-2014 from (a) the IBTrACS_wmo observations, and
the simulations of (b) BCC-CSM2-MR and (c) BCC-CSM2-HR. The value on the
upper-right corner denotes the total number of global TCs on 5 °×5 °grid box.



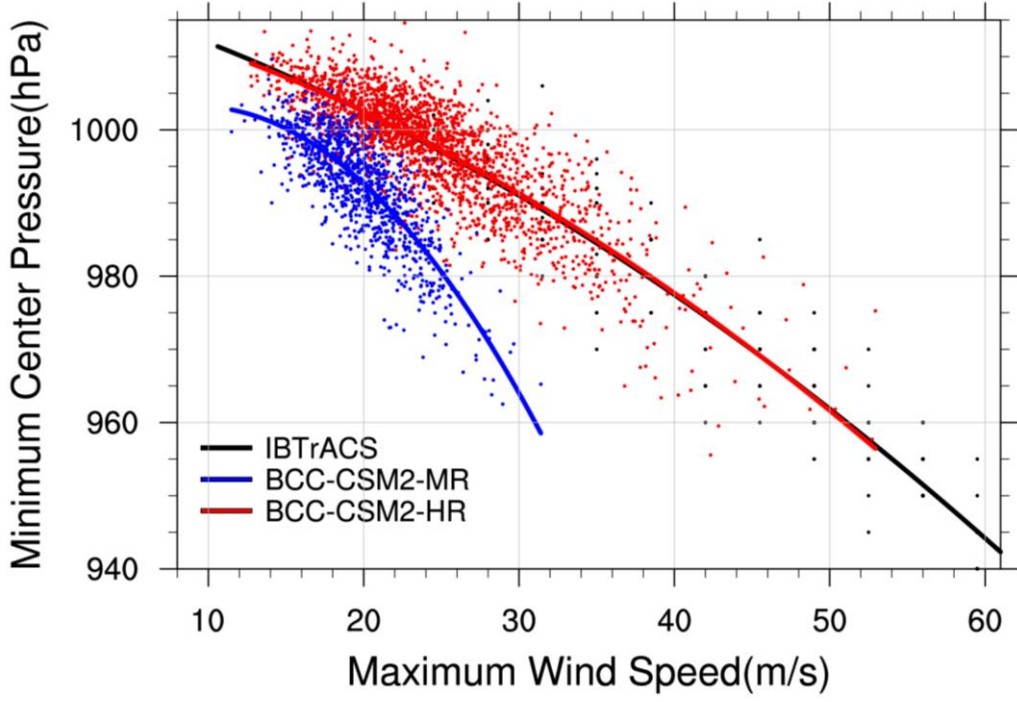



Figure 15. Maximum surface wind speed (m s$^{-1}$) versus minimum sea level pressure (hPa) for tropical cyclones from 6-hourly data for 1995-2014 IBTrACS observation (black dots and fitting line), and simulations from BCC-CSM2-HR (red dots and fitting line) and BCC-CSM2-MR (blue dots and fitting line). Each dot denotes the maximum surface wind speed and its corresponding minimum sea level pressure for a tropical cyclone during its lifetime.

1631

1632

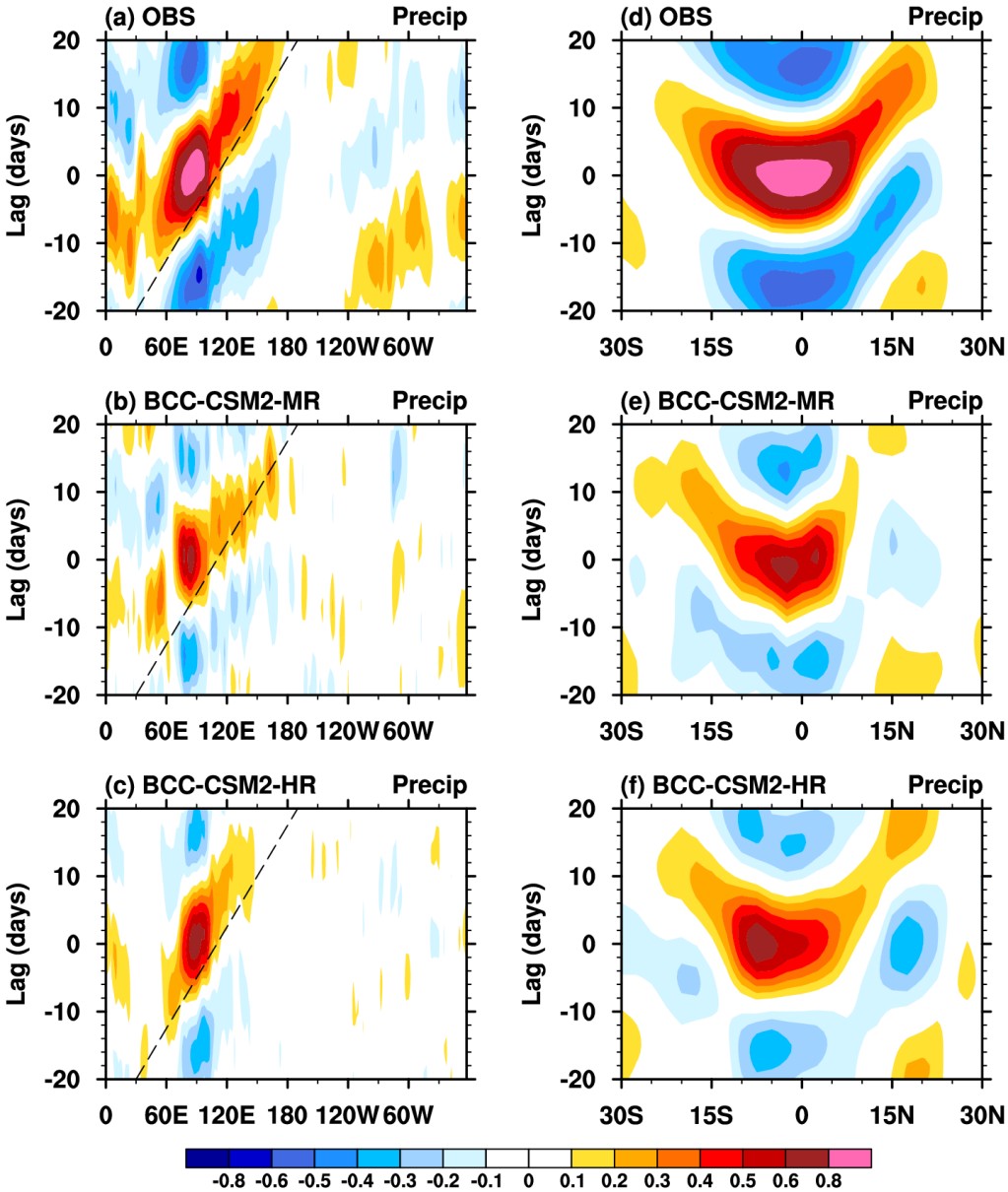

Figure 16. Left panels: longitude-time evolution of lagged correlation coefficient for the 20–100-day band-pass-filtered precipitation anomaly (averaged over 10 °S–10 °N) against regional averaged precipitation over the equatorial eastern Indian Ocean (80 °– 100 °E, 10 °S–10 °N). Right panels: same as the left panels, but for the latitude-time evolution of lagged correlation coefficient for filtered precipitation anomaly (averaged over 80 °–100 °E) against the regional averaged precipitation over the equatorial eastern Indian Ocean. Dashed lines in each panel denote the 5 m s$^{-1}$ eastward propagation speed. The observations in (a, b) are derived from GPCP data and the simulations are from (c,d) BCC-CSM2-MR, and (e,f) BCC-CSM2-HR for the period from 1995 to 2014.

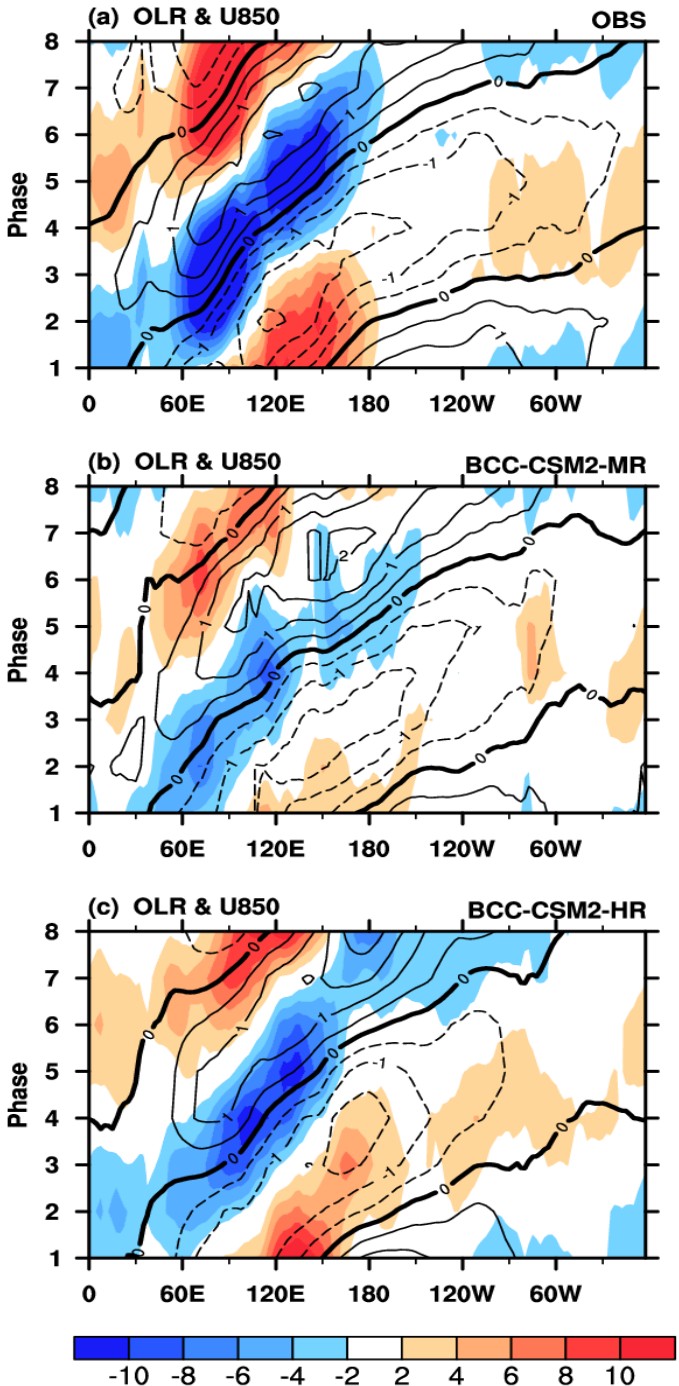

Figure 17. Hovmöller diagrams of MJO phase-composited OLR (shaded) and 850-hPa zonal wind anomalies (contour lines) averaged between 10 °S and 10 °N from (a) ERA5 wind and NOAA OLR reanalyses, (b) BCC-CSM2-MR and (c) BCC-CSM2-HR simulations for the period from 1995 to 2014. The MJO phase is defined by the two principal components corresponding to leading multivariate EOFs of OLR, 850-hPa and 200-hPa zonal wind anomalies as in Wheeler and Hendon (2004).











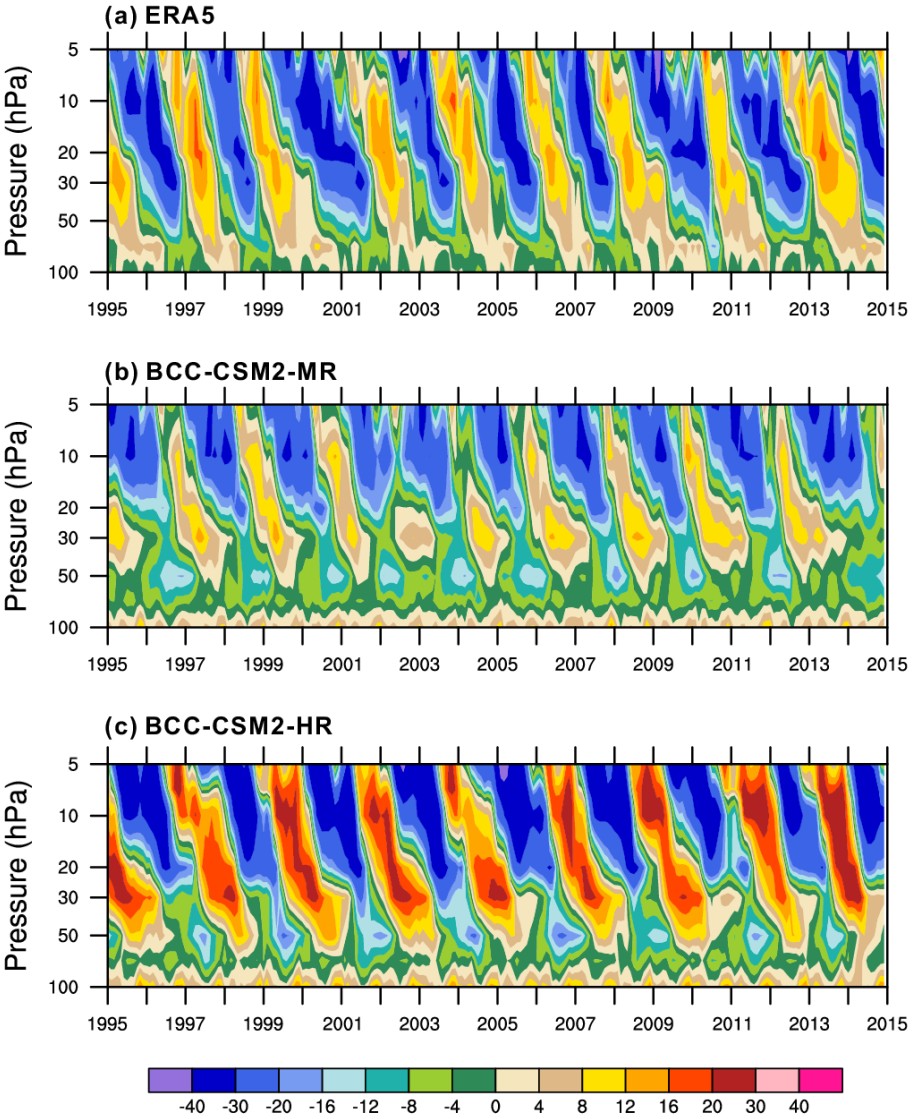



Figure 18. Tropical zonal winds (m s[-1]) between 5 °S and 5 °N in the lower stratosphere for (a)
ERA5 reanalysis, (b) BCC-CSM2-MR and (c) BCC-CSM2-HR during the period from 1995 to

1662 2014.




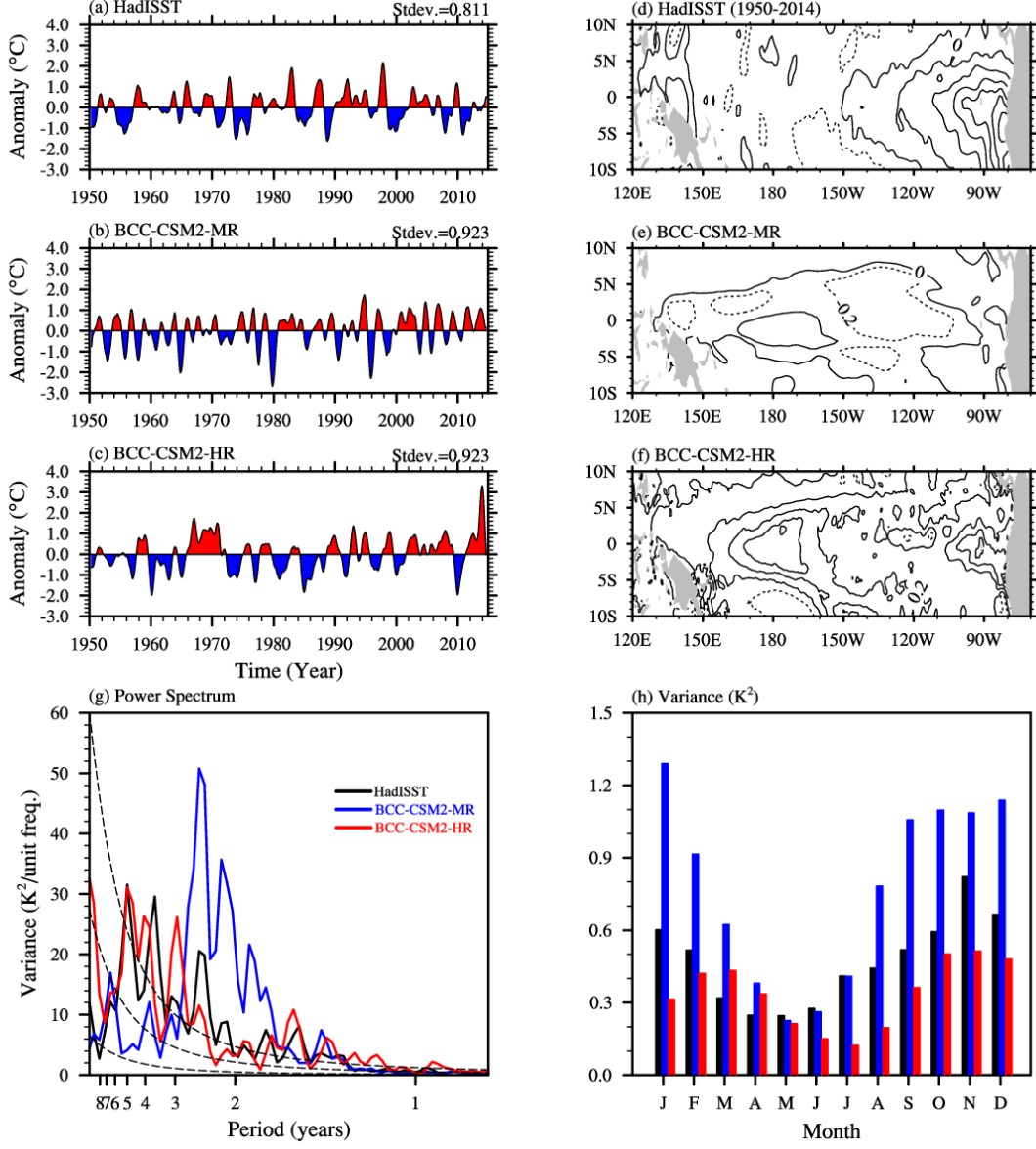


Figure 19. The time series of monthly Niño3.4 SST (5°N−5°S, 170°W−120°W)
anomalies and spatial distribution of their skewness for (a, d) HadISST observation,
(b, e) BCC-CSM2-MR, and (c, f) BCC-CSM2-HR during the period 1950-2014. (g)
and (h) show their power spectrums and variances, respectively, and the black, blue,
and red solid lines denotes the results from HadISST, BCC-CSM2-MR, and
BCC-CSM2-HR.



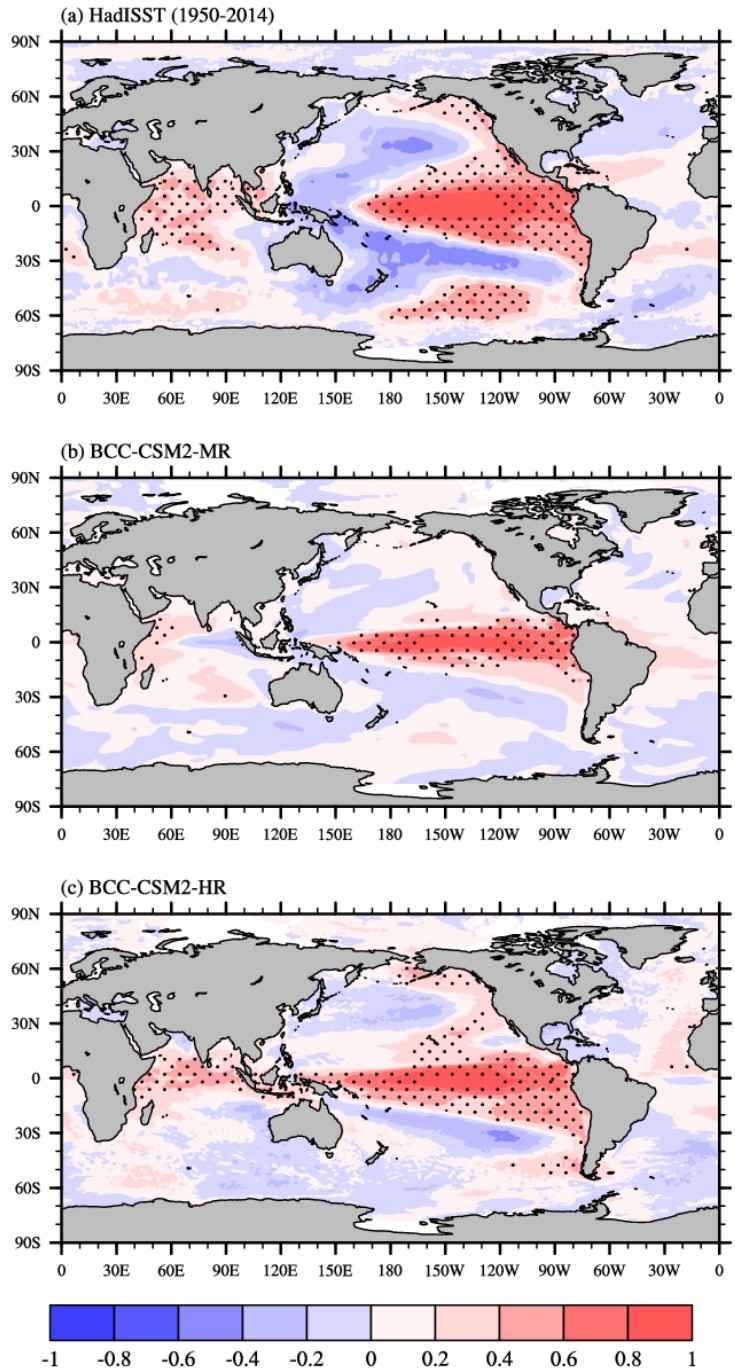



Figure 20. Correlation coefficients between SST and the Nino3.4 index from 1950 to
2014 for (a) HadISST data, (b) BCC-CSM2-MR, and (c) BCC-CSM2-HR. Contour
intervals are 0.2. Values significant at the 99% level using a Student's t-test are
stippled.