# Peer review of "BCC-CSM2-HR: A High-Resolution Version of the Beijing Climate"

_Geoscientific Model Development, 2020_

## Referee Comment (RC1) · Anonymous Referee #1 · 31 Dec 2020

This is an excellent model description paper from the team of Beijing Climate Center (BCC), which is a major climate modeling center in China and has been engaged in the development of climate/earth system models in the past decades, with excellent contributions to previous phases of CMIP. In this manuscript, the authors have documented the key technical details of the model which are crucial to the users of CMIP data of BCC models. The results are also useful to model developers as a reference for model development and improvement. In the manuscript, a comprehensive comparison of the historical simulations from middle- and high-resolution models of BCC is performed. The logic of the manuscript is well organized and helps model developers and users to know what kind of improvements can be achieved by developing a high

resolution ocean-atmosphere coupled climate system model. While I believe that the manuscript can be accepted for publication in GMD, I still find some spaces for a further improvement, such as the inconsistent period of model and observations, lack of explanations on the improvement or backset in high resolution modelling ect. A moderate revision is needed.

Major comments: 1. Logic of the manuscript: It would be better to collect the descriptions of observational datasets in a new section before "Results", instead of in each result subsection. The resolution of all the observational data used should also be marked. 2. How did you compare the low- and high-resolution data on a lat-lon map? Do you interpolate from low to high or high to low? and Why? Similar to the observational data, all the methods used should also be introduced and summarized before showing the results. 3. In Table 2 and related subsection, I think using the same period as CERES-EBAF product to evaluate the two-version models is better. What is the meaning of errors (how do you calculate it) in Table 2 and text? 4. I wonder why the period of 1971-2000 is used. According to the description of historical simulation of these two models (L356-359), they both ends at 2014 as recommended by CMIP6. So, using period of 1995-2014 should be better as more observational data are available. 5. Figure 7 uses a different color set to represent high and mid resolution models from Figure 2. For reading more easily, I recommend to make the color legend consistent throughout the manuscript. 6. L463-464: Can you explain why HR model improves the DJF precipitation in the SPCZ? Is it controlled by resolution or parameterization? Such kinds of information are very helpful to other model developers. 7. Figure 8: Here I think you should use the period of 2001-2014. 8: Figure 9: Large biases in Kuroshio extension and North Atlantic in higher resolution model should be marked and give possible reasons. I wonder whether this bias is resulted from the coarse resolution of observation, viz. the observation is "wrong" here due to its low resolution. 9: Figure 10: The color bar is weird. It is not easy to capture the relative magnitude, especially the areas with biases around zero value. 10: Why does HR model improve the TC density in western Pacific but not in the North Atlantic? Any explanations?

[Figure]

11: It is very interesting that the HR model can produce an excellent wind-pressure relation. Can you give a short physical explanation? 12: The color bar in Figure 17 should also be replaced by that or similar type used in Figure 16. 13: In Figure 18, the time series in (a)-(c) subpanels are not suitable for comparison. Maybe you can use probability density function to show the asymmetry and skewness of ENSO. 14. How weaknesses of observational data could influence the model evaluation, especially for the high-resolution result is recommended to be discussed. For example, low resolution SST data is unable to capture the SST gradient along the Kuroshio and Gulfstream extension regions, it would be unfair for high resolution models if you use low resolution data as observational metrics.

Minor comments:

1. L3, P77: The following two papers are useful references here on how high resolution improves the monsoon simulation: Zhang L. et al. 2018. Effect of Horizontal Resolution on the Representation of the Global Monsoon Annual Cycle in AGCMs. Adv. Atmos. Sci., 10.1007/s00376-018-7273-9. https://link.springer.com/article/10.1007/s00376-018-7273-9 Yao J. et al. 2017: Improved performance of High-Resolution Atmospheric Models in simulating the East-Asian Summer Monsoon Rainbelt. Journal of Climate 30(21), 8825-8840, https://doi.org/10.1175/JCLI-D-16-0372.1 2. L51: Sea Surface Temperature (SST): the abbreviation should be used in Line 43 and the first letters should be in lower case. 3. P4, L102: In the climate model development community of China, the BCC holds a special position in that it is engaged in the development of its own climate models. The model has been used in both operational seasonal forecast and CMIP-like climate change simulation and projection. In contrast, other CMIP6 models from China are either hybrid models developed for research and education or purely research models. You may refer to Zhou et al. (2020) for the special position of BCC models in China: Zhou, T. et al. 2020: Development of Climate and Earth System Models in China: Past Achievements and New CMIP6 Results. J. Meteor. Res., 34(1), 1-19, doi: 10.1007/s13351-020-9164-0

---

## Referee Comment (RC2) · Anonymous Referee #2 · 3 Mar 2021

Overall, this is a straightforward and easy-to-read paper that documents the high resolution BCC-CSM2-HR model within the context of a previous documented medium resolution model (BCC-CSM2-MR). The paper concludes that there are noticeable improvements in the mean state and phenomena of interest at high resolution, but biases still exist. This paper is appropriate for eventual publication in Geoscientific Model Development, but I have outlined some major comments below that need to be addressed before the manuscript can be accepted. In addition, I have provided numerous specific comments for consideration by the authors.

Major Comments:

[Figure]

1. Throughout the manuscript it is often unclear what grid(s) is being used for the analysis. That is, what grid are the observations and model results plotted on? Are they re-gridded to a common grid? Are they on the native grid? The interpretation of the results (or, in some cases, even the results themselves) can be influenced by this, I think it is important that the authors provide more detail on this topic.

2. In Section 4 details of the observational datasets, including spatial resolution, temporal frequency, and time period, is not always clear. I would recommend that the authors reorganize the manuscript and include a Section that introduces all of the datasets and their details before Section 4.

3. For all figures consistent line colors for observations, BCC-CSM2-MR and BCC-CSM2-HR should be used. In particular, Figures 1, 2, 7, 8, 12, 14 and 18 use multiple different colors (red, blue, black, green, purple, etc.) to denote the simulations and observations. They should be the same colors for all figures and datasets.

4. Finally, I think the usefulness of this manuscript would be enhanced if the Conclusions (Section 5) included some discussion of how these main results compare to other work that has explored the effect of increases in resolution in climate models on the mean state, circulation, and phenomena of interest. In particular, the authors could put this work in the context of HighResMIP results, as well as some of the papers references in the Introduction of the manuscript.

Specific Comments:

L29-31: Provide detail of the horizontal grid spacing here if possible.

L34: Consider changing "dynamic core" to "dynamical core" throughout the manuscript.

L57-58: Extend this sentence to put in context of the more recent CMIP6 (in addition to CMIP5).

L66: Remove "but."

[Figure]

L76: Change to "the QBO."

L78: Replace "and" with ",".

L94: Be specific that the authors are referring to atmosphere and ocean grids here.

L98: HighResMIP is not "the primary activity" of CMIP6, as only a subset of models has completed it. Please reword.

Table 1: There seems to be a miss formatted Wu (2012) reference in the "Deep convection" row.

L371-L372: Three models? Only two models are introduced in Section 2.

Figure 2: See major comment about regridding above.

L403-408: Figure 3: It would be easier to see the biases if the models were plotted as a difference from the observations. Consider adding additional panels to the Figure.

L446: Important variables in what way?

Figure 6: See major comment about regridding above.

L456-457: The authors should discuss the degradation in the simulation quality of precipitation east of the Philippines near the Pacific warm pool during JJA. I believe this was also seen in the Bacmeister et al. 2014 paper cited in the Introduction (see their Fig. 8). Some discussion of this degradation is needed here.

L465-468: This is an example where the underlying grid could be impacting the analysis (if the models and observations are not compared on a common grid – which is not obvious here).

Figure 8: See major comment about regridding above. The differences in grid could have implications here.

L491&L498-499: The authors could discuss how common this high-resolution cold bias is among other modeling groups, such as those that participated in HighResMIP.

Figure 10: The authors should include a panel of the observational (CRU ) data at the top of this plot, similar to what was done for Figure 9.

L503-507: Is this somewhat to be expected since there are no wholesale (besides resolution) changes to the land modeling component?

L511-512: What is the resolution of the HadISST product? Please provide that information.

L544: What impact does this threshold have on the results if the same value is used for both resolutions?

L542-551: Provide more information on the temporal frequency of the storm tracking? Is it daily for all tracking steps? Is intensity a mean or instantaneous?

Figure 14: This analysis is doing for daily storm intensities? Please provide more detail. How is the daily value (mean) calculated when storm intensities are typically represented instantaneously (as in IBTrACS). Are storms tracked daily or 6-hourly similar to IBTrACS? See comment above.

Figure 14: I find it difficult to believe that a model with ∼45 km grid spacing is replicating these high intensities (particularly, surface wind speeds) so well. But, it is also hard to interpret what a daily maximum intensity is. The authors should put this result into the context of the HighResMIP results, as well as Davis 2018 (https://doi.org/10.1002/2017GL076966).

L595: How is skillfully defined here?

L603: Provide detail of what observational dataset is used for OLR here.

L1310: Is it 3 hourly for BCC-CSM2-MR and BCC-CSM2-HR? To make this clear consider removing the "," after "2019."

---

## Referee Comment (RC3) · Anonymous Referee #3 · 22 Mar 2021

This paper intends to provide a documentation for the latest version of the high-resolution Bejing Climate Center Climate System model (BCC-CSM2-HR). It consists of a succinct overview of what are new in each of the model components and an evaluation of the model performance using a broad set of climate metrics. The evaluation of the BCC-CSM2-HR is presented in parallel with that from a medium resolution model, BCC-CSM2-MR, which differs from the HR model not only in resolution, but also in core components. Improvements of the HR over MR are highlighted throughout the text. BCC-CSM2-MR is used to participate in CMIP6, while BCC-CSM2-HR participating in HiResMIP. The materials presented regarding the models and the simulation

skills are expected to be useful for the users of the models and/or the simulated data, and thus suitable for a publication on GMD as an essential reference.

The manuscript however has much room to be improved. The manuscript apparently has writing contributions from different coauthors. The quality of the writing for different parts of the manuscript are quite different. There are many incomplete, inaccurate, inconsistent or unclear statements throughout the text. Many of them are pointed out in the specific comments below, which are not in the order of severeness of concern, but in the order of their appearances in the originally submitted manuscript. Given the broad evaluation on various model features and climate phenomena, it is understandable coauthors contribute to different sections or subsections with their expertises, and the materials in general are appropriate for a model documentation paper like this. But lack of thorough review of the final writing seems evident.

In addition to addressing the comments below to refine the writing, it is also suggested to provide a separate section to describe the observational data used in the evaluation and if any regriddings (for grid normalization) are involved. In the current form, the description for the reference data are scattered and very limited.

Specific comments

Line 31, "participation to", use "participation in" instead.

Lines 45-45, "double ITCZ . . . is . . . disappeared", just use "disappears".

Linne 65, remove 'but'

Line 68, higher resolutions are not limited to 50; change "i.e." to "e.g.".

Line 76, just say "MJO" in the context should be sufficient to convey the same message concerning MJO. Do the authors have other emphasis implied for MJO in its representations in model?

Line 78, The meaning of "small-scale" here is very vague, and storms and TCs are

not really small-scale. Suggest to change to "weather scale" or "small-scale processes associated with mid-latitude storms and tropical cyclones".

Line 92, change 'within' to 'with' or 'using'

Line 95, it is costly but certainly more than just a few research centers can perform it. Suggest to remove "and can be realized only at a few . . . .:., or as a way to transition to the follow-up description, emphasize it with " . . . costly effort but a growing number of research centers can exercise it.'

Line 110, use consistent notation for the numbers in the same phrase, either with "2 weeks to 1 yea" or "two weeks to one year".

Line 111-112, I think "the medium resolution version" is more suitable here than "the previous version of medium resolution". Both HR and MR model mentioned are for the same generation (BCC-CSM2) and the 'current' version does not seem to have an MR model that shares the same code base as the HR.

Line 123, to 'a' fine grid, just like describing the other grids.

Line 124, with an achievement to "deliver all of these model versions".

Line 125, suggest to put it as " four components – atmosphere, ocean, land and sea ice – interacting with each other".

Acronyms should be defined when first appearedïjŇe.g., AVIM, BCC-AGCM3, BCC-AGCM, among others.

Figure 1, it should be useful to have a separate subplot for the thickness of the model layers in the lowest few kilometers; but it does not have to, given no emphasis on the description of layer thickness of the lower tropospheric levels.

Liness 145 and 187, replace 'dynamic core' with 'dynamical core'.

Line 170, suggest to change to 'at different spatiotemporal scales and interactions
between them'.

Line 189, to make it clear, please use 'grid spacing dependence'.

Line 202, suggest to change to '... at the top and the kth layers of the model, respectively'.

Line 168 and more in the text, suggest to use 'Spatially-varying' in place of 'spatially-variable'. Also change it in Table 1 for the HR column of the dynamical core.

Line 205, it should be useful to explain what the 'polar instabilities' here refer to and how it increases with damping coefficient near model top.

Line 209, The meaning of the sentence starting from "This is ..." is not clear. Does this properly convey the meaning: "This is possibly due to much more damping of the meridional waves,"?

Line 231-232 on transporting entrained cloud water to its neighboring grids inside the model time step. Does it involve treatments other than what can be expected from typical process splitting? The description seems to imply that as it is said to be only in the HR model. Please elaborate, given that it appears to be a unique feature.

Line 236, use in favor of instead.

Line 248, BLs, in general, readers should be able to know what it refers to. But as always, please expand it in the first use.

Line 255, 'svl' does not have 'vl' as subscript.

Line 248, some description should be helpful on how to detect the presence of inversion layer for determining which eddy dffusivity formulation would be used for a column, especially if thd detection of inversion involves some special treatment.

Line 290, cite some references that use LTS=17.5K as a threshold criterion for BL stability to factually demonstrate the 'robustness' as written in the text. Furthermore,

the Hack scheme is said to be applied to the whole atmosphere column previously. Please elaborate if it remains so after introducing this activation threshold, and if so, justify the use beyond boundary layers.

Line 306-307, It appears that AVIM2.3 is mentioned abruptly. Is AVIM2.3 used in the HR model while AVIM2.2 in the MR version? Is the difference between v2.2 and v2.3, in sub-grid surface classification, to take advantage of the capability of the HR model? Last, provide a description for the grid configurations used by the land surface model – aware it is indicated in Table 1.

Table 1, please add references for CLM3, SISv1, SISv2.

Lines 320-323, the text for the neutral diffusion scheme with constant diffusivity here to describe the MOM5 that is used in the HR model appears inconsistent with that in Table 1, which says neutral diffusion scheme is not used.

Line 315, what does it mean by 'comforts of algorithm'?

Lines near 318, please add some reference for tracer advection scheme MDPPM.

Line 336-337, add a reference for the Semtner's scheme.

Line 342, use plural for 'simulations'.

Line 343, suggest to remove 'from 1971 to 2000', which would otherwise gives an incorrect initial impression that the simulations themselves are for that period only. It is clear enough at the beginning of the Results section and it is this period of the historical simulations that are analyzed for the paper.

Lines 371-372, Only see two models involved. Is 'three models' just a typo? Also, it is dubious to say making a 'right' intercomparison. A 'reasonable' intercomparison would sound more appropriate here. After all, the model components are quite different, so do the initialization procedure and the simulation period.

Line 375-376, the meaning of the sentence is ambiguous. suggest to change to "...

primary source of data for estimating Earth's energy balance."

Line 377 and related text: which version of CERES-EBAF data are being used?

Lines 385-386, the text says "TOA LW and SW components in HR are much closer to CERES-EBAF than the MR". The numbers in Table 2 clearly says the opposite for almost all quantities with the exception of clear sky fluxes.

Lines 395-397 and Table 2. It is not true both the MR and HR have stronger LW cloud radiative forcing than CERES-EBAF data. Only the HR model does. The sentence as is is also problematic. Rephrase to reflect that the HR model has near 2 W/m2 of additional warming effect (biases). SW cloud forcing for CERES-EBAF looks like from an earlier version of data. If using the latest version of CERES-EBAF (e.g., v4.1), the model-observation discrepancy could be much larger. Again, please indicate the version of CERES-EBAF data used, and try to use the latest version of data if not yet. Given the several inconsistencies between the text and the Table 2, I am not sure if the correct table is used in the manuscript.

Lines 406-408, suggest to change to " . . . new treatments for boundary layer processes" , because there is a new scheme introduced, as well as new treatment to go with a scheme (Hack scheme) that is also used by MR. Also suggest to remove the 2nd part of the sentence, it could be perceived like the confinement of water vapor only occurs to the HR model over the eastern ocean basins. Moreover, simply attributing to the parameterization scheme could be an understatement because differences in both horizontal and vertical resolutions could also have an impact.

Though CIRA86 appears to indicate the data period is for 1986, it is still necessary to explicitly state the time span of the data, just like for the other data products.

Lines 416-421. It is unclear whether the first sentence is to describe the observed vertical structure or the model biases. It is more likely the former but then the wordings are not appropriate. The vertical structure is the well-known Earth's atmospheric stratification, from surface up, troposphere, stratosphere and part of mesosphere in the HR model. The cool layers span the transition from troposphere to stratosphere layer. The description in the text appears like casually picking a pair of cold/warm features without reference to the actual atmosphere. The 'cool layers' as used in the text, the center of which over broader latitudes reflect the location of tropopause, and it is not centered near 300 hPa. The layer centered at 1hPa cannot be 'too warm' by itself.. It marks the top of the stratosphere and the transition to mesosphere. I think the current description is oversimplified and not acceptable, other than that the HR model is capable of capturing the structure of upper stratosphere and the transition to mesosphere while the MR model cannot, and the reversal of polar stratosphere structure from DJF to JJA. The description for Figure 4 needs to be totally rewritten.

Line 429, The description does not appear to separate well this tropical region of warm biases and the thicker layer of warm biases in broader lower stratosphere over the tropics and mid-latitude. Note that the warm biases are still situated in lower stratosphere, not 'upper stratosphere'.

Line 442-443, the reference to QBO here is purely speculative and without basis. These are biases in mean climatology, as a long term mean, even if the models have skill in simulating QBO, the signal would have been largely averaged out in long term mean. Suggest to remove this statement.

Fig 7, right panel for zonal mean precipitation over land? No words about it? HR should resolve better the precipitation features influenced by orography. If there is no intent to provide a description to highlight the improvements over land in HR, might as well not to include the Figure.

Huffman et al. 2019 for IMERG data and algorithm, please provide a URL for the reference. Also on IMERG data at line 475, while what is stated in Huffman et al. is that their algorithm can also be used to intercalibrate 'potentially other precipitation estimators', it sounds very odd that the available IMERG data product has combined 'potentially

other precipitation estimators' – which would mean the source data for IMERG are not fully clear. The 'potential' apparently is for different context, Re-examine the IMERG document if such 'other' data are included and if so make it explicit. The description in the current form is not appropriate.

Again on IMERG and Figure 8, though it can be assumed in such comparison, it should be explicitly stated that all data are rearranged to have the same grid resolution and time averaging interval. Don't see it in the text. Also, the Figure caption says the data are three hourly, but line 476 says hourly precipitation. Make them consistent. Furthermore, the description for Figure 8 is less than accurate and over simplified, which could be misleading by following the text alone. The 1mm/hr and 10/hr cutoff may be about fine for the MR model, but substantially off for the HR model. The authors may start with just describing the comparison of MR with IMERG, then highlight the improvement of HR, instead of describing them in the same sentence with the same cutoff value. The last sentence in the paragraph is well suited.

Line 486-488, can the authors provide an explanation why selecting EN4 and CRU data as references? There could be many other choices. Some description on the benefit of selecting these reference data should add credentials to the evaluation process.

Line 495, it should help by explicitly referring to Fig. 3b in addition to the text regarding strong SW cloud radiative forcing.

Figure 11, The plot should be for spatial distribution of sea ice concentration rather than overall sea ice extent. The description uses the correct term.

Line 527, "mostly smaller"? It does not read well with "almost smaller", the meaning of which would also be ambiguous.

Line 531, add 'by' to have "overestimated sea ice by about . . .".

Paragraph starting line 542, TC criteria, please explain why using a relative vorticity criterion differing by 15 times for the HR and MR models, how sensitive are the results

to this criterion and how it impacts the interpretation of the derived results.

Line 542, Citing a single work of Murakami (2014) does not appear consistent with a plural form of 'previous studies'. Suggest to cite the works that first used the criteria that are listed in the paragraph.

Line 546, please elaborate how the air temperature deviation is defined or cite references that provide clear description for the calculations. The threshold of 0.8 K as the sum of the deviations at three levels also appears to be small, compared to thresholds used in the definitions of TC warm core in other TC trackers (e.g., the warm core criterion in GFDL TSTORMS tracker is at least 1K warmer than the surrounding local mean (Zhao et al., doi: 10.1175/2009JCLI3049.1) for an averaged temperature over a depth, not even a sum).

Lines 548-549: 'within the vortex center 3 deg x 3 deg grid box', suggest to change to " within the vortex center for an area of 3 deg x 3 deg'. Please also indicate which level of wind speed is concerned here?

Line 556-559, the description of global annual mean TC numbers read like it is also higher in the MR model than IBTrACS, which is not true given the numbers in the text and Figure 13. Furthermore on the description for Figure 13, while the speculation of the factors that are possibly relevant for missing Atlantic TC are reasonable, would the authors offer some discussion if the different criteria used play a role?

Line 575 about models cannot capture weak storms with maximum wind speeds less than 10 m/s. The statement seems inconsistent. Even IBTrACS does not have it, while actually the MR model has some with max wind speed at or below 10m/s. The description should also indicate that max wind speeds in MR are consistently weaker which is understandable given coarser resolution.

Line 580, the whole paragraph is to describe Figure 14 which is indicated at the beginning. Remove the redundant mentioning of Figure 14 at the end.

Line 588, should be lag-latitude evolution for the right panels of Figure 15.

Figure 16a, which observational/reanalysis data are used to create the observed MJO life cycle, or if it is adapted from a figure in another publication?

Line 619, use instead 'A good simulation of QBO . . . '

Line 630, suggest to remove 'in amplitude'. The meaning of the sentence would remain intact. Amplitude typically would account for the extent of the oscillation between both phases.

Line 637-638, the description sounds like the authors have looked at the comparison of the parameterized convective gravity wave forcing, but the results are less than conclusive, is that so? Otherwise, without any concrete evidence or explanation, how to draw a statement that " . . . seemed enhanced . . .". A statement of the forcing could potentially be enhanced would be a better statement if all by speculation, but that should also need to be justified.

Line 658 at the beginning of the 2nd sentence, suggest to change to "The phase locking (i.e., the peak variance) . . ." to inform even less familiar readers what the term refers to.

Line 666-667, suggest to change to "despite over extension into the Western Pacific".

Line 667-671, after describing the observed expansion of influence to extra-tropics, it should be followed by the description of more equatorially/tropically confined in the models. The focus is not on observational analysis, after all. The descriptions for the models are clearly over-simplified, with a single statement that HR improves over MR. Moreover, for a plot like this and the description to compare models vs obs on both negative and positive correlation, it should be useful to denote in the plot where the correlations are statistically significant, otherwise some of the discussions could be simply irrelevant.

Line 681, use participating in.

Itemization in the paragraph starting line 683, use 'First,....  Second, ..., Third ....' instead.

Lines 702-703, the description would give an impression that the simulations are just for 30 years. Suggest to change to "historical simulations with fully coupled BCC-CSM2-MR and BCC-CSM2-HR are analyzed over a 30 year period from 1971 to 2000.".

Line 715, lower troposphere temperature biases are relatively small in both models, right? Indicate so.

Lines 719-720 "do not change at higher resolution" and "insensitive to atmospheric resolution" are apparently redundant.

---

## Author Comment (AC1) · 3 Apr 2021

**Response to comments from Reviewer #1**

*We are grateful for Reviewer #1's constructive and insight comments. We have addressed all the comments in the revised manuscript. Our point-to-point responses are hereafter in Italic Font.*

Major comments:
1. Logic of the manuscript: It would be better to collect the descriptions of observational datasets in a new section before "Results", instead of in each result subsection. The resolution of all the observational data used should also be marked.
*Response: This point was a common concern of all three anonymous reviewers. It is also our major structural revision implemented in the revised manuscript. Actually, we added a new section "3.2 Data used for evaluations" to introduce all the observational datasets that we used our purpose of model assessment, together with their basic characteristics and properties.*

*Another structural revision is the inclusion of a new subsection "4.1 Global mean surface air temperature variations from 1950 to 2014" to assess the ability of our models in reproducing the historical evolution of global climate for the recent past. Figure 2 is new and all subsequent figures are shifted and re-numbered.*

2. How did you compare the low- and high-resolution data on a lat-lon map? Do you interpolate from low to high or high to low? and Why? Similar to the observational data, all the methods used should also be introduced and summarized before showing the results.
*Response: Generally speaking, when visual inspection is the purpose, we just plot observations and simulations on their native grid. But for quantitative assessment of model biases, we re-gridded simulations onto the corresponding observation's grid. In the revised manuscript, we added a paragraph of explanation at the beginning of Section 4: "Data analysis and visualization are generally on the original or native grid of observation and models. An exception is on the assessment of models' biases with contrast to observation. In this case, simulations are re-gridded onto the grid of corresponding observation" in line 492-495..*

3. In Table 2 and related subsection, I think using the same period as CERES-EBAF product to evaluate the two-version models is better. What is the meaning of errors (how do you calculate it) in Table 2 and text?
*Response: Indeed, it is more convincing to use the same period for model/observation inter-comparison. We thus adopted the common period of 2001-2014. Values in Table 2 denote the annual mean ±interannual standard deviation for 2001-2014.*

4. I wonder why the period of 1971-2000 is used. According to the description of historical simulation of these two models (L356-359), they both ends at 2014 as recommended by CMIP6. So, using period of 1995-2014 should be better as more observational data are available.

*Response: Following your suggestion, all the analyses are now changed to the reference period of 1995-2014.*

5. Figure 7 uses a different color set to represent high and mid resolution models from Figure 2. For reading more easily, I recommend to make the color legend consistent throughout the manuscript.

*Response: Colors are now consistent among relevant figure, i.e. black, red, and blue for observations, BCC-CSM2-HR, and BCC-CSM2-MR, respectively.*

6. L463-464: Can you explain why HR model improves the DJF precipitation in the SPCZ? Is it controlled by resolution or parameterization? Such kinds of information are very helpful to other model developers.

*Response: We think the cause is multiple but we tend to conclude that the physical parameterization is the primary cause. We added some explanations in the revised manuscript. "This systematic bias is evidently reduced in BCC-CSM2-HR, especially with weakened precipitation in the South Pacific Convergence Zone (SPCZ). The improvement of SPCZ precipitation in BCC-CSM2-HR might be attributed to the implementation of the UWMT scheme which improved the simulation of low-level clouds over the tropical eastern South Pacific (Lu et al., 2020b) and reduced warm biases there (Fig. 10c)" in line 591-596*

7. Figure 8: Here I think you should use the period of 2001-2014.

*Response: Modified. It is numbered to Figure 9.*

8: Figure 9: Large biases in Kuroshio extension and North Atlantic in higher resolution model should be marked and give possible reasons. I wonder whether this bias is resulted from the coarse resolution of observation, viz. the observation is "wrong" here due to its low resolution.

*Response: The observation is certainly not wrong, but it may be not enough, with its low spatial resolution, to reveal detailed structures for the Golf Stream and the Kuroshio current, including their extensions to eastern basins. We added some explanations "We also noted that a belt of warm SST biases in the Kuroshio extension and in the North Atlantic in both models (Figures 10b and 10c), especially in the high-resolution model. This bias may be partly resulted from the coarse resolution of HadISST data used, as SST near the Kuroshio shows strong temperature gradients with filamentous structures (Shi and Wang, 2020)" in line 627-631.*

9: Figure 10: The color bar is weird. It is not easy to capture the relative magnitude, especially the areas with biases around zero value.

*Response: Modified*

10: Why does HR model improve the TC density in western Pacific but not in the North Atlantic? Any explanations?

*Response: Many studies show that increasing atmospheric resolution is helpful to improve the simulation of TC. This is also the case in our simulations with a general improvement*

*of TC density in BCC-CSM2-HR. The tropical North Atlantic, however, does not show improvement in BCC-CSM2-HR. We think that this persistent bias is possibly caused by cold SST biases. We added some explanations in lines 691-700 in the revised text.*

11: It is very interesting that the HR model can produce an excellent wind-pressure relation. Can you give a short physical explanation?

*Response: Yes, we are also happy to see that our high-resolution model can well reproduce the relation between wind speed and surface pressure for detected tropical cyclones. We think that this should not be a fortuitous result, since many other studies (e.g. Murakami et al., 2012; Sugi et al., 2017; Vecchi et al., 2019) concluded on the importance of high resolution in simulating TC. Those studies also demonstrate that the maximum wind speed of TC simulated by a model with approximately 50 km resolution can reach up to 50~60 m s-1. We added some discussions in the revised manuscript in Section "4.4.1 Tropical Cyclones".*

*Here, we can further present a case of TC simulated in the western tropical Pacific in BCC-CSM2-HR, as shown in Figure S1. It corresponds to model calendar 2003-11-23 UTC18:00:00. There is a clear TC structure with circular sea level pressure isobars and strong winds around the TC eyewall. The maximum wind speed can reach 53.9 m/s with the minimum sea level pressure of 975.2 hPa.*

[Figure]

*Figure S1. Snapshots of the TC with maximum intensities simulated by BCC-CSM2-HR in model date (a) 2003-11-23UTC12:00:00, (b) 2003-11-23UTC18:00:00 and (c) 2003-11-24UTC00:00:00, respectively. The shaded indicates the wind speed (m/s) at 10m. Contours indicate sea level pressure (hPa) with an interval of 5 hPa.*

12: The color bar in Figure 17 should also be replaced by that or similar type used in Figure 16.
*Response: The shaded areas in Figure 16 and 17 show OLR and zonal wind, respectively.*

13: In Figure 18, the time series in (a)-(c) subpanels are not suitable for comparison. Maybe

you can use probability density function to show the asymmetry and skewness of ENSO.

*Response: Following your suggestion, we added three maps to show skewness of ENSO.*

14. How weaknesses of observational data could influence the model evaluation, especially for the high-resolution result is recommended to be discussed. For example, low resolution SST data is unable to capture the SST gradient along the Kuroshio and Gulfstream extension regions, it would be unfair for high resolution models if you use low resolution data as observational metrics.

*Response: We absolutely agree with this point of view. When model's resolution reaches high levels, the lack of adequate observations becomes a major obstacle for our modeling efforts. We added some discussions in lines 627-631 in the revised manuscript.*

Minor comments:

1. L3, P77: The following two papers are useful references here on how high resolution improves the monsoon simulation: Zhang L. et al. 2018. Effect of Horizontal Resolution on the Representation of the Global Monsoon Annual Cycle in AGCMs. Adv. Atmos.Sci., 10.1007/s00376-018-7273-9.https://link.springer.com/article/10.1007/s00376-018-7273-9 Yao J. et al. 2017: Improved performance of High-Resolution Atmospheric Models in simulating the East-Asian Summer Monsoon Rain belt. Journal of Climate, 30(21), 8825-8840, https://doi.org/10.1175/JCLI-D-16-0372.1 2.

*Response: Modified*

2. L51: Sea Surface Temperature (SST): the abbreviation should be used in Line 43 and the first letters should be in lower case.

*Response: Modified.*

3. P4, L102: In the climate model development community of China, the BCC holds a special position in that it is engaged in the development of its own climate models. The model has been used in both operational seasonal forecast and CMIP-like climate change simulation and projection. In contrast, other CMIP6 models from China are either hybrid models developed for research and education or purely research models. You may refer to Zhou et al. (2020) for the special position of BCC models in China: Zhou, T. et al. 2020: Development of Climate and Earth System Models in China: Past Achievements and New CMIP6 Results. J. Meteor. Res., 34(1), 1-19, doi: 10.1007/s13351-020-9164-0

*Response: Modified.*

---

## Author Comment (AC2) · 3 Apr 2021

**Response to comments from Reviewer #2**

*We thank reviewer #2 for his/her constructive and insight comments. We have addressed these comments in the revised manuscript. All our responses are in Italic Font.*

Major Comments:

1. Throughout the manuscript it is often unclear what grid(s) is being used for the analysis. That is, what grid are the observations and model results plotted on? Are they re-gridded to a common grid? Are they on the native grid? The interpretation of the results (or, in some cases, even the results themselves) can be influenced by this, I think it is important that the authors provide more detail on this topic.

*Response:*

*Response: Generally speaking, when visual inspection is the purpose, we just plot observations and simulations on their native grid. But for quantitative assessment of model biases, we re-gridded simulations onto the corresponding observation's grid. In the revised manuscript, we added a paragraph of explanation at the beginning of Section 4: "Data analysis and visualization are generally on the original or native grid of observation and models. An exception is on the assessment of models' biases with contrast to observation. In this case, simulations are re-gridded onto the grid of corresponding observation" in line 492-495.*

2. In Section 4 details of the observational datasets, including spatial resolution, temporal frequency, and time period, is not always clear. I would recommend that the authors reorganize the manuscript and include a Section that introduces all of the datasets and their details before Section 4.

*Response: This point was a common concern of all three anonymous reviewers. It is also our major structural revision implemented in the revised manuscript. Actually, we added a new section "3.2 Data used for evaluations" to introduce all the observational datasets that we used our purpose of model assessment, together with their basic characteristics and properties.*

*Another structural revision is the inclusion of a new subsection "4.1 Global mean surface air temperature variations from 1950 to 2014" to assess the ability of our models in reproducing the historical evolution of global climate for the recent past. Figure 2 is new and all subsequent figures are shifted and re-numbered.*

3. For all figures consistent line colors for observations, BCC-CSM2-MR and BCCCSM2-HR should be used. In particular, Figures 1, 2, 7, 8, 12, 14 and 18 use multiple different colors (red, blue, black, green, purple, etc.) to denote the simulations and observations. They should be the same colors for all figures and datasets.

*Response:*

*Modified. Line colors in all figures are replotted to consistence as black, blue, and red colors to denote observations, BCC-CSM2-MR, and BCC-CSM2-HR, respectively.*

4. Finally, I think the usefulness of this manuscript would be enhanced if the Conclusions (Section 5) included some discussion of how these main results compare to other work that has explored the effect of increases in resolution in climate models on the mean state, circulation, and phenomena of interest. In particular, the authors could put this work in the context of HighResMIP results, as well as some of the papers references in the Introduction of the manuscript.

*Response: Yes, this is a good point, but it is a little beyond the scope of our manuscript which modestly focuses on the documentation of the basic performance of our high-resolution model. But in the revised manuscript, we did add some discussions on JJA precipitation over the western Pacific warm pool, and on lack of TC activities over the North Atlantic, in comparison to other works. More simulations from other groups in the framework of HighResMIP start to be gradually released, which will allow us to do further diagnostics with multiple models.*

Specific Comments:
L29-31: Provide detail of the horizontal grid spacing here if possible.
*Response: Modified.*

L34: Consider changing "dynamic core" to "dynamical core" throughout the manuscript.
*Response: Modified.*

L57-58: Extend this sentence to put in context of the more recent CMIP6 (in addition to CMIP5).
*Response: Modified.*

L66: Remove "but."
*Response: Modified.*

L76: Change to "the QBO."
*Response: Modified*

L78: Replace "and" with ",".
*Response: Modified*

L94: Be specific that the authors are referring to atmosphere and ocean grids here.
*Response: Modified. That sentence is modified as "At present day, performing high-resolution climate simulations with model grid smaller than 50 km in the atmosphere and 0.25 °in the ocean is still a very costly effort …" in line 100-102.*

L98: HighResMIP is not "the primary activity" of CMIP6, as only a subset of models has completed it. Please reword.
*Response: Modified. That sentence is rewritten as "The High Resolution Model Intercomparison Project (HighResMIP, Haarsma et al., 2016) is a CMIP6-endorsed MIP*

*(Model Intercomparison Project), which aimed to investigate the impact of model resolution on climate simulation fidelity and systematic model biases." in line 105-108.*

Table 1: There seems to be a miss formatted Wu (2012) reference in the "Deep convection" row.
*Response: Modified.*

L371-L372: Three models? Only two models are introduced in Section 2.
*Response:Corrected.*

Figure 2: See major comment about regridding above.
*Response: We have added descriptions about how to plot for different resolution data.*

L403-408: Figure 3: It would be easier to see the biases if the models were plotted as a difference from the observations. Consider adding additional panels to the Figure.
*Response: Modified. We have added model biases with comparison against CERES-EBAF data.*

L446: Important variables in what way?
*Response: In the revised version, we have re-arrange this section. The expression of L446 in the first version of manuscript "Precipitation, land surface air temperature and sea surface temperature, sea-ice concentration are important variables of general concern" is now deleted to avoid ambiguity.*

Figure 6: See major comment about regridding above.
*Response:Data are plotted on their original resolution.*

L456-457: The authors should discuss the degradation in the simulation quality of precipitation east of the Philippines near the Pacific warm pool during JJA. I believe this was also seen in the Bacmeister et al. 2014 paper cited in the Introduction (see their Fig. 8). Some discussion of this degradation is needed here.
*Response:*
*We added some discussions about the precipitation degradation over the Pacific warm pool region as shown in Figure 6, now re-numbered as Figure 7 in the new manuscript. "In Figure 7f, we also noted that the amount of JJA precipitation in east of the Philippines and near the Pacific warm pool is worsened, since it is smaller in BCC-CSM2-HR than in BCC-CSM2-MR and GPCP data. This bias of lacking precipitation in BCC-CSM2-HR may partly be caused by a cold-SST bias over the western Pacific warm pool (Fig.10c)" in line 601-605.*

*We agree with the reviewer that this precipitation imperfection in high-resolution models is an important issue. With a simple bibliographic search, we found several other models reporting precipitation biases with resolution increased. So, we added more discussions in the section "5. Conclusions and discussions". Our work shows that enhancing resolution*

*does not noticeably improve climate mean state and a deterioration is even possible. For example, the decrease of JJA precipitation over the warm pool in our high-resolution model is still an important issue which certainly deserves further investigations with multiple models and simulations. Actually, other studies also reported similar issues. Haarsma et al. (2020) shows that increasing resolution in the EC-Earth model deteriorated the wet bias over the western Pacific warm pool. Bacmeister et al. (2014) analysed the high-resolution climate simulations performed with the Community Atmosphere Model (CAM), and showed that dry bias over the same region with enhanced resolution. Over the western Pacific warm pool, the atmospheric circulation and precipitation undergoes not only the impact of tropical variations such as MJO and TC, but also strong regional air-sea coupling.*

L465-468: This is an example where the underlying grid could be impacting the analysis (if the models and observations are not compared on a common grid – which is not obvious here).
*Response: Yes, we agree that different resolutions of observation and simulation affect the analyses. As mentioned in our response to Major comment #1, we added a clear statement on how to deal this issue of spatial resolution. Actually, when simulation is quantitatively compared against observation, we re-grid the simulation onto the grid of observation. But in Figure 7, the biases of precipitation over the tropical Pacific are real, not an artifact of grid transformation.*

Figure 8: See major comment about regridding above. The differences in grid could have implications here.
*Response: Two simulations were re-gridded to the grid of IMERG before processing..*

L491&L498-499: The authors could discuss how common this high-resolution cold bias is among other modeling groups, such as those that participated in HighResMIP.
*Response: Yes, we agree that this is another important issue for HighResMIP. Up to now, we cannot find publications to explore changes of warm SST bias in the eastern basins of subtropical oceans for those models participating in HighResMIP.*

Figure 10: The authors should include a panel of the observational (CRU) data at the top of this plot, similar to what was done for Figure 9.
*Response: We have added the CRU data plot in Figure 9 which is renumber to Figure 10 in revised version of the manuscript.*

L503-507: Is this somewhat to be expected since there are no wholesale (besides resolution) changes to the land modeling component?
*Response: Yes, the result conforms to expectation, since the land modeling component keeps very close to each other in the two models, and biases of near-surface air temperature over land are very similar to each other.*

L511-512: What is the resolution of the HadISST product? Please provide that information.
*Response: Provided.*

L544: What impact does this threshold have on the results if the same value is used for both resolutions?
*Response: In the first version of the manuscript, the threshold of relative vorticity at 850 hPa to detect TC for BCC-CSM2-MR and BCC-CSM2-HR were wrongly set to different values, which created difficulties for our purpose of inter-comparison, although that practice was perfectly valid to detect TC in individual models. We unified now the threshold (to a unique value of $15 \times 10^{-5} s^{-1}$ in Figure 14). We also added some discussions on the influence of different thresholds: "If this threshold gets looser to $5 \times 10^{-5} s^{-1}$, the averaged total global TC numbers per year in BCC-CSM2-MR and BCC-CSM2-HR will enhance to 55.9 and 101.1, respectively" in line 684-686. The following figure shows the global distribution, but omitted in the main text of the revised manuscript.*

[Figure]

*Figure S1. The global distribution of tropical cyclone (TC) densities (number per year) averaged for BCC-CSM2-MR (left column), and BCC-CSM2-HR (right column). The thresholds of the relative vorticity for tracking the TC are $5 \times 10^{-5} s^{-1}$, $10 \times 10^{-5} s^{-1}$, $15 \times 10^{-5} s^{-1}$, respectively. The value on the upper-right corner denotes the total number of global TCs on $5° \times 5°$ grid box.*

L542-551: Provide more information on the temporal frequency of the storm tracking? Is it daily for all tracking steps? Is intensity a mean or instantaneous?
*Response: Added more descriptions in "3.2 Data used for evaluations". The 1995–2014 6-hourly tropical cyclones observations from International Best Track Archive for Climate Stewardship (IBTrACS; Knapp et al., 2010) are used, which contains the information of latitude, longitude position, minimum central pressure, and maximum sustained winds at a*

*time frequency of every 6 hours of instantaneous values for all tropical cyclones. Following previous studies (Murakami, 2014), we use multiple criteria to detect TCs for every 6-hours interval outputs of instantaneous values from BCC-CSM2-HR and averaged means from BCC-CSM2-MR.*

Figure 14: This analysis is doing for daily storm intensities? Please provide more detail. How is the daily value (mean) calculated when storm intensities are typically represented instantaneously (as in IBTrACS). Are storms tracked daily or 6-hourly similar to IBTrACS? See comment above.

*Response: We added more description in "4.4.1 Tropical Cyclones" to explain for Figure 14 (now renumbered to Figure 15 in the revised version). "the maximum surface wind speed (minimum sea level pressure) of a given TC was defined as the instantaneous maximum (minimum) of the 6-hours interval in IBTrACS and BCC-CSM2-HR, but averaged value in BCC-CSM2-MR for wind speed at 10m (sea level pressure) …" in line 704-707.*

Figure 14: I find it difficult to believe that a model with 45 km grid spacing is replicating these high intensities (particularly, surface wind speeds) so well. But, it is also hard to interpret what a daily maximum intensity is. The authors should put this result into the context of the HighResMIP results, as well as Davis 2018 (https://doi.org/10.1002/2017GL076966).

*Response:    We referenced to other similar simulations studies (e.g. Murakami et al., 2012; Sugi et al., 2017; Vecchi et al., 2019). Those studies also demonstrate that the maximum wind speed of TC simulated by a model with approximately 50 km resolution can reach up to 50~60 m s-1. So, we have added some discussion in the "4.4.1 Tropical Cyclones". Here, we present a case of TC occurred on 2003-11-23, UTC18:00:00 in the western tropical Pacific in BCC-CSM2-HR as shown in Figure S2. There is a clear TC structure with the circular sea level pressure isobar and strong wind around the TC eyewall in the Figure S2, and the strongest TC in BCC-CSM2-HR, whose maximum wind speed can reach to 53.9 m/s with the minimum sea level pressure of 975.2 hPa (Figure S2).*

[Figure]

*Figure S2. Snapshots of the TC with maximum intensities simulated by BCC-CSM2-HR in model date (a) 2003-11-23UTC12:00:00, (b) 2003-11-23UTC18:00:00 and (c) 2003-11-24UTC00:00:00, respectively. The shaded indicates the wind speed (m/s) at 10m. Contours indicate sea level pressure (hPa) with an interval of 5 hPa.*

*The simulation of TC in tropical Atlantic is indeed a challenge for BCC-CSM2-HR. As your comment, we referenced to Davis (2018) which shows that models with horizontal grid spacing of one fourth degree or coarser should not produce a realistic number of category 4 and 5 storm in tropical Atlantic. So, we added some discussion about that in "5. Conclusions and discussions".*

L595: How is skillfully defined here?
*Response: rewritten that sentence. The signal of northward propagation is more evident in BCC-CSM2-HR than in BCC-CSM2-MR.*

L603: Provide detail of what observational dataset is used for OLR here.
*Response: Provided.*

L1310: Is it 3 hourly for BCC-CSM2-MR and BCC-CSM2-HR? To make this clear consider removing the "," after "2019."
*Response: Yes, in Figure 9, it is 3 hourly for BCC-CSM2-MR and BCC-CSM2-HR. we have rewritten the caption of Figure 9.*

---

## Author Comment (AC3) · 3 Apr 2021

**Response to comments from Reviewer #3**

*We thank reviewer #3 for his/her constructive and insight comments. We have addressed these comments in the revised manuscript. All our responses are in Italic Font.*

This paper intends to provide a documentation for the latest version of the high-resolution Bejing Climate Center Climate System model (BCC-CSM2-HR). It consists of a succinct overview of what are new in each of the model components and an evaluation of the model performance using a broad set of climate metrics. The evaluation of the BCC-CSM2-HR is presented in parallel with that from a medium resolution model, BCC-CSM2-MR, which differs from the HR model not only in resolution, but also in core components. Improvements of the HR over MR are highlighted throughout the text. BCC-CSM2-MR is used to participate in CMIP6, while BCC-CSM2-HR participating in HiResMIP. The materials presented regarding the models and the simulation skills are expected to be useful for the users of the models and/or the simulated data, and thus suitable for a publication on GMD as an essential reference.

The manuscript however has much room to be improved. The manuscript apparently has writing contributions from different coauthors. The quality of the writing for different parts of the manuscript are quite different. There are many incomplete, inaccurate, inconsistent or unclear statements throughout the text. Many of them are pointed out in the specific comments below, which are not in the order of sereneness of concern, but in the order of their appearances in the originally submitted manuscript. Given the broad evaluation on various model features and climate phenomena, it is understandable coauthors contribute to different sections or subsections with their expertises, and the materials in general are appropriate for a model documentation paper like this. But lack of thorough review of the final writing seems evident.

In addition to addressing the comments below to refine the writing, it is also suggested to provide a separate section to describe the observational data used in the evaluation and if any regriddings (for grid normalization) are involved. In the current form, the description for the reference data are scattered and very limited.

*Response: Following your suggestion, we added a section "3.2 Data used for evaluations" to introduce all of the datasets and their details, and refine the writing for the context about the model description and results.*

*In addition, in order to test the rationality of historical simulations from 1950 to 2014, we added a subsection "4.1 Global mean surface air temperature variations from 1950 to 2014", and added a figure, so all the figures in the first version of manuscript are re-numbered.*

**Specific comments**

Line 31, "participation to", use "participation in" instead.
*Response: Modified.*

Lines 45-45, "double ITCZ : : : is : : : disappeared", just use "disappears".
*Response: Modified.*

Line 65, remove 'but'
*Response: Modified.*

Line 68, higher resolutions are not limited to 50; change "i.e." to "e.g.".
*Response: Modified.*

Line 76, just say "MJO" in the context should be sufficient to convey the same message concerning MJO. Do the authors have other emphasis implied for MJO in its representations in model?
*Response: Modified. "the QBO, the MJO's representation" in line 79 of the revised version is changed to "the MJO"*

Line 78, The meaning of "small-scale" here is very vague, and storms and TCs are not really small-scale. Suggest to change to "weather scale" or "small-scale processes associated with mid-latitude storms and tropical cyclones".
*Response: Modified. That sentence is rewritten as "Some small-scale processes associated with mid-latitude storms and tropical cyclones, and ocean eddies also feedback on the simulated large-scale circulation, climate variability and extremes…" in line 84-86.*

Line 92, change 'within' to 'with' or 'using'
*Response: Modified.*

Line 95, it is costly but certainly more than just a few research centers can perform it. Suggest to remove "and can be realized only at a few : : :.:., or as a way to transition to the follow-up description, emphasize it with " : : : costly effort but a growing number of research centers can exercise it.'
*Response: Modified. That sentence is rewritten as "At present day, performing high-resolution climate simulations with model grid smaller than 50 km in the atmosphere and 0.25 °in the ocean is still a very costly effort but a growing number of research centers can exercise it…" in line 100-103.*

Line 110, use consistent notation for the numbers in the same phrase, either with "2 weeks to 1 year" or "two weeks to one year".
*Response: Modified.*

Line 111-112, I think "the medium resolution version" is more suitable here than "the previous version of medium resolution". Both HR and MR model mentioned are for the same generation (BCC-CSM2) and the 'current' version does not seem to have an MR model that shares the same code base as the HR.
*Response: Modified. That sentence is rewritten as "The purpose of this paper is to evaluate its performance by comparing it with the medium resolution version…" in line 117-119.*

Line 123, to 'a' fine grid, just like describing the other grids.
*Response: Modified.*

Line 124, with an achievement to "deliver all of these model versions".
*Response: Modified.*

Line 125, suggest to put it as " four components – atmosphere, ocean, land and sea ice – interacting with each other".
*Response: Modified.*

Acronyms should be defined when first appeared e.g., AVIM, BCC-AGCM3, BCCAGCM, among others.
*Response: Modified.*

Figure 1, it should be useful to have a separate subplot for the thickness of the model layers in the lowest few kilometers; but it does not have to, given no emphasis on the description of layer thickness of the lower tropospheric levels.
*Response: Following your suggestion, we added a subplot in Figure 1.*

Lines 145 and 187, replace 'dynamic core' with 'dynamical core'.
*Response: Modified.*

Line 170, suggest to change to 'at different spatiotemporal scales and interactions between them'.
*Response: Modified.*

Line 189, to make it clear, please use 'grid spacing dependence'.
*Response: Modified.*

Line 202, suggest to change to '... at the top and the kth layers of the model, respectively'.
*Response: Modified.*

Line 168 and more in the text, suggest to use 'Spatially-varying' in place of 'spatially-variable'. Also change it in Table 1 for the HR column of the dynamical core.
*Response: Modified.*

Line 205, it should be useful to explain what the 'polar instabilities' here refer to and how it increases with damping coefficient near model top.
Line 209, The meaning of the sentence starting from "This is : : :" is not clear. Does this properly convey the meaning: "This is possibly due to much more damping of the meridional waves,"?
*Response: We rewritten this paragraph in line 207-221.*

Line 231-232 on transporting entrained cloud water to its neighboring grids inside the model time step. Does it involve treatments other than what can be expected from typical process splitting? The description seems to imply that as it is said to be only in the HR model. Please elaborate, given that it appears to be a unique feature.

*Response: Along with increasing resolution in BCC-AGCM3-HR, the detrained cloud water can be transported to its adjacent grid boxes, which is accomplished in the dynamical core. Part of the horizontally-transported cloud water is permitted to be transferred downward to lower troposphere and the amount of downward transferred water vapor is determined by the horizontally-transported convective cloud water increment with time. These modifications of the deep convection scheme are only used in BCC-CSM2-HR.*

Along with increasing resolution in BCC-AGCM3-HR, the detrained cloud water can be transported to its adjacent grid boxes, which is accomplished in the dynamical core. Part of the horizontally-transported cloud water is permitted to be transferred downward to lower troposphere and the amount of downward transferred water vapor is determined by the horizontally-transported convective cloud water increment with time

Line 236, use in favor of instead.
*Response: Modified.*

Line 248, BLs, in general, readers should be able to know what it refers to. But as always, please expand it in the first use.
*Response: Modified. BLs denotes boundary layers.*

Line 255, 'svl' does not have 'vl' as subscript.
*Response: Modified.*

Line 248, some description should be helpful on how to detect the presence of inversion layer for determining which eddy diffusivity formulation would be used for a column, especially if the detection of inversion involves some special treatment.
*Response: Our original meaning to present is " In the case of an entrainment layer, at the top of convective boundary layers (BLs), the diffusivity is parameterized with …", not " In the case of an inversion layer at the top of convective". So, we rewrote that sentence.*

Line 290, cite some references that use LTS=17.5K as a threshold criterion for BL stability to factually demonstrate the 'robustness' as written in the text. Furthermore, the Hack scheme is said to be applied to the whole atmosphere column previously. Please elaborate if it remains so after introducing this activation threshold, and if so, justify the use beyond boundary layers.
*Response: We have rewritten the description as "In the light of its robustness, a new lower troposphere stability (LTS) criteria is introduced into the original Hack scheme in BCC-CSM2-HR, which is defined as …. The modified Hack scheme is activated only in the decoupled BL regimes with $LTS < 17.5 \text{ K}$ below 700 hPa to remove adiabatically moist*

*instability, and the original Hack scheme (Hack, 1993) is still retained above 700 hPa to remove any local instability as long as the two adjacent model layers are moist adiabatically unstable.*

Line 306-307, It appears that AVIM2.3 is mentioned abruptly. Is AVIM2.3 used in the HR model while AVIM2.2 in the MR version? Is the difference between v2.2 and v2.3, in sub-grid surface classification, to take advantage of the capability of the HR model? Last, provide a description for the grid configurations used by the land surface model – aware it is indicated in Table 1.
Table 1, please add references for CLM3, SISv1, SISv2.
*Response: Modified the description in the context for Land surface model and be consistent to Table 1.*

Lines 320-323, the text for the neutral diffusion scheme with constant diffusivity here to describe the MOM5 that is used in the HR model appears inconsistent with that in Table 1, which says neutral diffusion scheme is not used.
*Response: We corrected the description as "MOM5 has a complete set of physical processes with advanced parameterization schemes. Effect of mesoscale eddies through the neutral diffusion scheme of Griffies et al. (1998) is not included in this work." in line 350-352 in the context and Table 1*

Line 315, what does it mean by 'comforts of algorithm'?
*Response: The correct expression should be "comforts of algorithm implementation". we rewrote that sentence as "The quasi-horizontal rescaled height coordinate, namely, z\* vertical coordinate is employed to enhance flexibility of model applications, which allows the free surface to fluctuate to values as large as the local ocean depth" in line 343-346.*

Lines near 318, please add some reference for tracer advection scheme MDPPM.
*Response: Modified.*

Line 336-337, add a reference for the Semtner's scheme.
*Response: Modified.*

Line 342, use plural for 'simulations'.
*Response: Modified.*

Line 343, suggest to remove 'from 1971 to 2000', which would otherwise give an incorrect initial impression that the simulations themselves are for that period only. It is clear enough at the beginning of the Results section and it is this period of the historical simulations that are analyzed for the paper.
*Response: Yes, the medium-resolution simulation is a typical CMIP6 historical run, from 1850 to 2014, as defined by the CMIP framework. Our high-resolution simulation is however an ad hoc run, starting from 1950 and terminating in 2014. We rewrote that*

*sentence as "The principal simulation to be analyzed is the CMIP6 historical run (hereafter referred to as historical) with prescribed forcings from 1850 to 2014 for BCC-CSM2-MR and from 1950 to 2014 for BCC-CSM2-HR." in line 373-375.*

Lines 371-372, Only see two models involved. Is 'three models' just a typo? Also, it is dubious to say making a 'right' intercomparison. A 'reasonable' intercomparison would sound more appropriate here. After all, the model components are quite different, so do the initialization procedure and the simulation period.

*Response:We deleted the original expression and added "We choose the same period of 1950-2014 from both BCC-CSM2-MR and BCC-CSM2-HR historical simulations to evaluate their performance against observation-based or reanalysis data" in line 402-404.*

Line 375-376, the meaning of the sentence is ambiguous. suggest to change to "...primary source of data for estimating Earth's energy balance."
*Response: Modified.*

Line 377 and related text: which version of CERES-EBAF data are being used?
*Response: Modified. CERES-EBAF version 4.1 data are used.*

Lines 385-386, the text says "TOA LW and SW components in HR are much closer to CERES-EBAF than the MR". The numbers in Table 2 clearly says the opposite for almost all quantities with the exception of clear sky fluxes.
Lines 395-397 and Table 2. It is not true both the MR and HR have stronger LW cloud radiative forcing than CERES-EBAF data. Only the HR model does. The sentence as is also problematic. Rephrase to reflect that the HR model has near 2 W/m2 of additional warming effect (biases). SW cloud forcing for CERES-EBAF looks like from an earlier version of data. If using the latest version of CERES-EBAF (e.g., v4.1), the model-observation discrepancy could be much larger. Again, please indicate the version of CERES-EBAF data used, and try to use the latest version of data if not yet. Given the several inconsistencies between the text and the Table 2, I am not sure if the correct table is used in the manuscript.

*Response: We apologize for the confusion. Values recorded in Table 2 were correct ones, but we did not update the corresponding text. We revised the inconsistent expressions and rewrote that paragraph in 513-531.*

Lines 406-408, suggest to change to " : : : new treatments for boundary layer processes", because there is a new scheme introduced, as well as new treatment to go with a scheme (Hack scheme) that is also used by MR. Also suggest to remove the 2nd part of the sentence, it could be perceived like the confinement of water vapor only occurs to the HR model over the eastern ocean basins. Moreover, simply attributing to the parameterization scheme could be an understatement because differences in both horizontal and vertical resolutions could also have an impact.
*Response: Suggestions adopted. We rewrote that paragraph as "In mid-latitudes of both the hemispheres, the shortwave cloud radiative forcing from BCC-CSM2-HR is much closer to*

*CERES-EBAF than from BCC-CSM2-MR. But in low latitudes between 30 °S and 30 °N, BCC-CSM2-HR simulates excessive cloud shortwave radiative forcing which mainly comes from evident biases over the eastern tropical Pacific and tropical Atlantic oceans (Figure 4). These biases are possibly attributable to new treatments for boundary layer processes." in line 536-542.*

*At the recent, we test the new model physics with the same horizontal and vertical resolutions to BCC-CSM2-MR, and find out the same biases of cloud shortwave radiative forcing occurred in BCC-CSM2-HR.*

Though CIRA86 appears to indicate the data period is for 1986, it is still necessary to explicitly state the time span of the data, just like for the other data products.
*Response: We rewrote the description for CIRA-86 as "CIRA-86 (available at https://catalogue.ceda.ac.uk/uuid/4996e5b2f53ce0b1f2072adadaeda262) includes a global climatology of zonal atmospheric temperature and velocity extending from pole to pole on a 5-degree latitude grid and 0-120 km approximately at 2 km vertical resolution. It is derived from a combination of satellite, radiosonde and ground-based measurements (Fleming et al., 1990)." in line 437-442.*

Lines 416-421. It is unclear whether the first sentence is to describe the observed vertical structure or the model biases. It is more likely the former but then the wordings are not appropriate. The vertical structure is the well-known Earth's atmospheric stratification, from surface up, troposphere, stratosphere and part of mesosphere in the HR model. The cool layers span the transition from troposphere to stratosphere layer. The description in the text appears like casually picking a pair of cold/warm features without reference to the actual atmosphere. The 'cool layers' as used in the text, the center of which over broader latitudes reflect the location of tropopause, and it is not centered near 300 hPa. The layer centered at 1hPa cannot be 'too warm' by itself.. It marks the top of the stratosphere and the transition to mesosphere. I think the current description is oversimplified and not acceptable, other than that the HR model is capable of capturing the structure of upper stratosphere and the transition to mesosphere while the MR model cannot, and the reversal of polar stratosphere structure from DJF to JJA. The description for Figure 4 needs to be totally rewritten.
*Response: We apologize for the confusion and we rewrote that paragraph as "The observed vertical profile of atmospheric temperature shows a clear structure of stratification, with an evident seasonal transition. In DJF, it is characterized as cool layers over broader latitudes spanning the transition from troposphere to stratosphere over the Northern Hemisphere, and warm layers spanning from the top of the stratosphere to mesosphere over the Southern Hemisphere. Those different vertical structures in both hemispheres during DJF are almost reversed in JJA. BCC-CSM2-HR is capable of capturing the structure of upper stratosphere and the transition to mesosphere while BCC-CSM2-MR cannot." in line 549-557.*

Line 429, The description does not appear to separate well this tropical region of warm biases and the thicker layer of warm biases in broader lower stratosphere over the tropics and

mid-latitude. Note that the warm biases are still situated in lower stratosphere, not 'upper stratosphere'.

*Response: Modified. We have rewritten the description.*

Line 442-443, the reference to QBO here is purely speculative and without basis. These are biases in mean climatology, as a long term mean, even if the models have skill in simulating QBO, the signal would have been largely averaged out in long term mean. Suggest to remove this statement.

*Response: Adopted*

Fig 7, right panel for zonal mean precipitation over land? No words about it? HR should resolve better the precipitation features influenced by orography. If there is no intent to provide a description to highlight the improvements over land in HR, might as well not to include the Figure.

*Response: Following your suggestion, we removed the right panel for zonal mean precipitation.*

Huffman et al. 2019 for IMERG data and algorithm, please provide a URL for the reference.

*Response: Modified.*

Also on IMERG data at line 475, while what is stated in Huffman et al. is that their algorithm can also be used to intercalibrate 'potentially other precipitation estimators', it sounds very odd that the available IMERG data product has combined 'potentially other precipitation estimators' – which would mean the source data for IMERG are not fully clear. The 'potential' apparently is for different context, Re-examine the IMERG document if such 'other' data are included and if so make it explicit. The description in the current form is not appropriate.

*Response: Modified the description.*

Again on IMERG and Figure 8, though it can be assumed in such comparison, it should be explicitly stated that all data are rearranged to have the same grid resolution and time averaging interval. Don't see it in the text. Also, the Figure caption says the data are three hourly, but line 476 says hourly precipitation. Make them consistent.

*Response: Modified the figure 9 captions.*

Furthermore, the description for Figure 8 is less than accurate and over simplified, which could be misleading by following the text alone. The 1mm/hr and 10/hr cutoff may be about fine for the MR model, but substantially off for the HR model. The authors may start with just describing the comparison of MR with IMERG, then highlight the improvement of HR, instead of describing them in the same sentence with the same cutoff value. The last sentence in the paragraph is well suited.

*Response: Paragraph rewritten.*

Line 486-488, can the authors provide an explanation why selecting EN4 and CRU data as references? There could be many other choices. Some description on the benefit of selecting

these reference data should add credentials to the evaluation process.

*Response: We added more descriptions about data used in section "3.2 Data used for evaluations" including CRU data. EN4 SST is replaced by HadISST in order to consistence to HadISST Sea ice data.*

Line 495, it should help by explicitly referring to Fig. 3b in addition to the text regarding strong SW cloud radiative forcing.

*Response: modified.*

Figure 11, The plot should be for spatial distribution of sea ice concentration rather than overall sea ice extent. The description uses the correct term.

*Response: modified.*

Line 527, "mostly smaller"? It does not read well with "almost smaller", the meaning of which would also be ambiguous.

*Response: modified.*

Line 531, add 'by' to have "overestimated sea ice by about : : :".

*Response: modified.*

Paragraph starting line 542, TC criteria, please explain why using a relative vorticity criterion differing by 15 times for the HR and MR models, how sensitive are the results to this criterion and how it impacts the interpretation of the derived results.

*Response: In the first version of the manuscript, the threshold of relative vorticity at 850 hPa to detect TC for BCC-CSM2-MR and BCC-CSM2-HR were wrongly set to different values, which created difficulties for our purpose of inter-comparison, although that practice was perfectly valid to detect TC in individual models. We unified now the threshold (to a unique value of $15 \times 10^{-5}$ $s^{-1}$ in Figure 14). We also added some discussions on the influence of different thresholds: "If this threshold gets looser to $5 \times 10^{-5}$ $s^{-1}$, the averaged total global TC numbers per year in BCC-CSM2-MR and BCC-CSM2-HR will enhance to 55.9 and 101.1, respectively" in line 684-686. The following figure shows the global distribution, but omitted in the main text of the revised manuscript.*

[Figure]

*Figure S1. The global distribution of tropical cyclone (TC) densities (number per year) averaged for BCC-CSM2-MR (left column), and BCC-CSM2-HR (right column). The thresholds of the relative vorticity for tracking the TC are $5\times10^{-5} s^{-1}$, $10\times10^{-5} s^{-1}$, $15\times10^{-5} s^{-1}$, respectively. The value on the upper-right corner denotes the total number of global TCs on $5°\times5°$ grid box.*

Line 542, Citing a single work of Murakami (2014) does not appear consistent with a plural form of 'previous studies'. Suggest to cite the works that first used the criteria that are listed in the paragraph.

*Response: modified. We used the multiple criteria given in Murakami (2014) to detect TCs for 6-hour interval outputs: instantaneous values for BCC-CSM2-HR and accumulated ones for BCC-CSM2-MR.*

Line 546, please elaborate how the air temperature deviation is defined or cite references that provide clear description for the calculations. The threshold of 0.8 K as the sum of the deviations at three levels also appears to be small, compared to thresholds used in the definitions of TC warm core in other TC trackers (e.g., the warm core criterion in GFDL TSTORMS tracker is at least 1K warmer than the surrounding local mean (Zhao et al., doi: 10.1175/2009JCLI3049.1) for an averaged temperature over a depth, not even a sum).

*Response: We have rewritten the description as*
*"We use the multiple criteria reported by Murakami (2014) to detect TCs with 6-hourly outputs from models (instantaneous values from BCC-CSM2-HR, but accumulated values from BCC-CSM2-MR). (1) The maximum of relative vorticity of a TC-like vortex at 850 hPa exceeds $15\times10^{-5} s^{-1}$ (a threshold that can vary from $1\times10^{-5} s^{-1}$ to $15\times10^{-5} s^{-1}$ in function of resolution (Murakami, 2014). (2) The warm-core above the TC-like vortex, which is presented as the sum of the air temperature deviations (subtracting the maximum temperature from the mean temperature within the TC-like vortex center for an area of $10°\times10°$) at 300, 500 and 700 hPa, exceeds 0.8 K, a threshold falling in the range 0.6~1.0K*

*that are recommended in Murakami (2014); (3) The maximum wind speed at 850 hPa is higher than that at 300 hPa; (4) The maximum wind speed at 10 m within the TC-like vortex center for an area of 3 °×3 °grid is higher than 10 m s-1; (5) The genesis position of the TC-like vortex is over the ocean; (6) The duration of the TC-like vortex satisfied above conditions exceeds 48 hours." in line 476-490.*

Lines 548-549: 'within the vortex center 3 deg x 3 deg grid box', suggest to change to " within the vortex center for an area of 3 deg x 3 deg'. Please also indicate which level of wind speed is concerned here.
*Response: modified.*

Line 556-559, the description of global annual mean TC numbers read like it is also higher in the MR model than IBTrACS, which is not true given the numbers in the text and Figure 13. Furthermore on the description for Figure 13, while the speculation of the factors that are possibly relevant for missing Atlantic TC are reasonable, would the authors offer some discussion if the different criteria used play a role?
*Response: We added more discussion in the part "5. Summary and discussions" about insufficient TC activities in the North Atlantic.*

Line 575 about models cannot capture weak storms with maximum wind speeds less than 10 m/s. The statement seems inconsistent. Even IBTrACS does not have it, while actually the MR model has some with max wind speed at or below 10m/s. The description should also indicate that max wind speeds in MR are consistently weaker which is understandable given coarser resolution.
*Response: modified.*

Line 580, the whole paragraph is to describe Figure 14 which is indicated at the beginning. Remove the redundant mentioning of Figure 14 at the end.
*Response: Modified the caption of Figure 15.*

Line 588, should be lag-latitude evolution for the right panels of Figure 15.
*Response: Clarified.*

Figure 16a, which observational/reanalysis data are used to create the observed MJO life cycle, or if it is adapted from a figure in another publication?
*Response: Clarified*

Line 619, use instead 'A good simulation of QBO : : : '
*Response: Modified.*

Line 630, suggest to remove 'in amplitude'. The meaning of the sentence would remain intact. Amplitude typically would account for the extent of the oscillation between both phases.
*Response: Modified.*

Line 637-638, the description sounds like the authors have looked at the comparison of the parameterized convective gravity wave forcing, but the results are less than conclusive, is that so? Otherwise, without any concrete evidence or explanation, how to draw a statement that " : : : seemed enhanced : : :". A statement of the forcing could potentially be enhanced would be a better statement if all by speculation, but that should also need to be justified.
*Response: Modified.*

Line 658 at the beginning of the 2nd sentence, suggest to change to "The phase locking (i.e., the peak variance) : : :" to inform even less familiar readers what the term refers to.
*Response: Modified.*

Line 666-667, suggest to change to "despite over extension into the Western Pacific".
*Response: Modified.*

Line 667-671, after describing the observed expansion of influence to extra-tropics, it should be followed by the description of more equatorially/tropically confined in the models. The focus is not on observational analysis, after all. The descriptions for the models are clearly over-simplified, with a single statement that HR improves over MR. Moreover, for a plot like this and the description to compare models vs obs on both negative and positive correlation, it should be useful to denote in the plot where the correlations are statistically significant, otherwise some of the discussions could be simply irrelevant.

*Response: modified the description. And add the correlations are statistically significant in Figure 20.*

Line 681, use participating in Itemization. In the paragraph starting line 683, use 'First,.... Second, : : :, Third : : :.' instead.
*Response: Modified.*

Lines 702-703, the description would give an impression that the simulations are just for 30 years. Suggest to change to "historical simulations with fully coupled BCC-CSM2-MR and BCC-CSM2-HR are analyzed over a 30 year period from 1971 to 2000.".
*Response: Modified.*

Line 715, lower troposphere temperature biases are relatively small in both models, right? Indicate so.
*Response: Rewritten it as "*Temperature biases in the low- to mid-troposphere below 300 hPa in BCC-CSM2-HR are relatively small, within the range of -1K to 1K.*" in line 873-874.*

Lines 719-720 "do not change at higher resolution" and "insensitive to atmospheric resolution" are apparently redundant.
*Response: Rewritten it as "Although those prominent systematic biases in temperature and wind seem relatively insensitive to changes in atmospheric resolution…." in line 878-879.*

---

## Author Comment (AC4) · 3 Apr 2021

Dear Editor,

We have finished point-by-point response to all referee comments and specify all changes in the revised manuscript. Enclosed the marked-up manuscript version showing the changes made. Thanks.

Best regards,

Tongwen Wu, and Co-authors
Please also note the supplement to this comment:
https://gmd.copernicus.org/preprints/gmd-2020-284/gmd-2020-284-AC4-
supplement.pdf